# Uncovering Self-Emergent Similarity in Deep Vision Networks: A Systematic Framework

## Abstract

Similarity is a key construct in psychology, neuroscience, linguistics and computer vision. Similarity can manifest in various forms, including visual, semantic, and contextual similarity. Among these, semantic similarity is particularly important. Not only it serves as an approximation of how humans categorize objects by capturing connections and hierarchies based on shared functionality, evolutionary traits, and contextual meaning, but also offers practical advantages in computational modeling via the lexical structures such as WordNet. Unlike human polls, WordNet-defined similarity is constant and interpretable, making it an important baseline for evaluation. As in the domain of deep vision models there is still a lack of a clear understanding about the emergence of similarity perception, we introduce Deep Similarity Inspector (DSI). It is a systematic framework to inspect and visualize how deep vision networks develop their similarity perception during training and how it aligns with semantic similarity. Our experiments show that both Convolutional Neural Networks' (CNNs) and Vision Transformers' (ViTs) develop a rich similarity perception during learning with 3 phases (initial similarity surge, refinement, stabilization), while clear differences are found in their dynamics. Both CNNs and ViTs, besides the gradual mistakes elimination, improve the quality of mistakes being made (the mistakes refinement phenomenon).

## 1 Introduction

Similarity is studied in psychology and neuroscience to understand human categorization. The cognitive economy principle (Rosch & Lloyd, 1978) suggests that similarity perception, evolving over time (Medin et al., 1993), minimizes effort and finite cognitive resources. Another principle (Rosch & Lloyd, 1978) highlights that real-world objects have high correlational structure, making similarity a natural and intuitive basis for categorization. Notions of similarity vary, including visual and semantic ones. While semantic similarity is central to linguistics, it also plays a key role in vision and perception, explaining how humans categorize and relate objects through shared attributes, evolutionary traits, appearance, function, or context.

Similarity has also sparked interest in computer vision, mainly to make networks learn representations aligning better with human judgments or semantic similarity (Bertinetto et al., 2020; Bilal et al., 2017; Chen et al., 2020; Caron et al., 2021). While enforcing similarity is common, we argue that it is underexplored how and how well networks naturally develop similarity perception. As the visual world's correlated structure may be a sufficient reinforcement (Rosch & Lloyd, 1978), such evaluation would be an important contribution to the field of model evaluation and explainability.

While similarity/hierarchy-based performance evaluation initially sparked some interest, the attention shifted to basic accuracy in pursuit of performance gains (Russakovsky et al., 2015; Bertinetto et al., 2020). Recent limited studies (Huang et al., 2021; Bilal et al., 2017; Bertinetto et al., 2020) revisited using confusion patterns and templates from large-scale datasets to analyze how networks infer similarity. Nevertheless, they only presented a very limited view by examining only Convolutional Neural Networks (CNNs) and ignoring Vision Transformers (ViTs). Aside from minor qualitative assessments (Huang et al., 2021), no quantitative and systematic similarity-based model evaluation framework for during training analysis exists. None of the works simultaneously uses two complementary (Medin et al., 1993) approaches to similarity estimation (direct, indirect). Lastly,

they are based only on a functional approach to similarity estimation (using large-scale datasets), rather than a possible structural approach (Mopuri et al., 2020; Filus & Domanska, 2023).

Motivated by these significant gaps and the crucial role of similarity in categorization, we propose a framework called ***Deep Similarity Inspector (DSI)*** for inspection and visualization of deep vision networks' training from the perspective of similarity perception and numerical measures. It represents a significant departure from the available, predominantly qualitative, approaches, offering a more systematic and scalable solution. We use it to provide novel insights on how networks learn to recognize objects and how their perception changes during training, contributing to a better understanding of their knowledge and representations development. This contributes significantly to the domain of model evaluation. The fact that our framework enables better understanding of the learning processes, makes it important to the field of explainable artificial intelligence. Our key contributions are: **(1)** Definition and implementation of a systematic framework for comprehensive vision networks' training examination from the similarity perception perspective. **(2)** Formulation of convenient numerical metrics to examine how network's perception is developed during training, and aligns with semantic similarity. Using all the direct, indirect, functional and structural approaches to similarity measurements. **(3)** Thorough examination and comparison of the training process of CNNs and ViTs from the perspective of the perceived similarity and its alignment with semantic similarity. We provide our implementation and reproducibility details in Supp. Mat. and our partial results at `https://zenodo.org/records/13860285` (GitHub upon acceptance).

## 2 RELATED WORK

Similarity is crucial to human categorization and computer vision (Tang et al., 2017; Nayak et al., 2019; Muttenthaler et al., 2023; Chen et al., 2020), and is a broad concept with different notions Veit et al. (2017), e.g. visual similarity, feature similarity (comparing data points representations learned by networks, e.g. (Kornblith et al., 2019)), contextual similarity (it accounts e.g. for spatial relationships (Shi et al., 2019)), adversarial similarity (similarity under adversarial setup (Elsayed et al., 2018; Filus & Domańska, 2024)) and semantic similarity (Pedersen et al., 2004). Vision research focuses mainly on creating training schemes that incorporate different kinds of similarity and hierarchy into networks, e.g. via hierarchy enforcement (Bertinetto et al., 2020; Bilal et al., 2017) or contrastive representation learning (Chen et al., 2020; Caron et al., 2021; He et al., 2020). Different sources of similarity are used as a reference for the neural representations alignment: e.g. human judgments (Muttenthaler et al., 2023; Roads & Love, 2021) or semantic similarity (Bilal et al., 2017). While these efforts undeniably try to explicitly force neural networks to better align similarity, we focus on examining "*to what extent perception of deep vision networks trained with standard procedures aligns with semantic similarity naturally (with NO enforcement) during training?*". We use semantic similarity as a reference, because human-based judgments are very data-dependent (unique high-volume judgements needed for every dataset), limiting their applicability. By this, we enabled creation of a more general similarity-focused framework that can operate with numerous datasets and problems, while still providing an important reference of how similarity can be perceived (semantic similarity is based on shared functionalities, appearance, evolutionary traits, or contextual usage - all impacting also visual similarity). Answering the stated questions is essential to reveal how important similarity is to the operation of networks due to the natural relationships between visual and semantic similarities (Deselaers & Ferrari, 2011). The works that explicitly examine whether the perception of networks aligns with semantic similarity are (Bilal et al., 2017) and (Huang et al., 2021). They both performed the analysis only for CNNs, therefore the analysis and comparison of similarity perception with Vision Transformers remain absent. They focused mainly on examining trained networks (Huang et al., 2021) based on confusion matrices or extracted features (we propose to use an image-free approach). Although researchers acknowledge that confusion patterns connect somehow to similarity (Deng et al., 2010; Bilal et al., 2017; Jere et al., 2019; Filus & Domanska, 2023) (also under the adversarial setup (Elsayed et al., 2018; Mopuri et al., 2020)), they do not thoroughly examined this connection. Also, available works focus on qualitative methods and not quantitative ones. While they draw attention to the topic, they only touch its very surface and do not show how modern networks develop similarity perception and how it affects their predictions. No adequate and systematic framework for similarity-based testing is available up to this day that could be efficiently used during training. That is why, we introduced such a framework and provide observations on how networks perceive and develop similarity perception.

## 3 DEEP SIMILARITY INSPECTOR FRAMEWORK

**Deep Similarity Inspector (DSI)** is a framework proposed to enable effective inspection and visualization of the learning process of deep vision networks for object recognition from the perspective of similarity perception. Its unique characteristics are that (1) it uses simultaneously the direct and indirect, as well functional and structural approaches to similarity estimation, (2) its focus is on numerical analysis methods to quantitatively describe the similarity perception of deep vision networks. It considers its alignment with reference similarity (WordNet's semantic similarity). It also includes qualitative methods. Due to the efficiency and the compliance with the existing training procedures, the proposed framework and its provided implementation, can be used to investigate the networks both during and post training. Below, we define the core data structures and methods of the framework (see Appendix L for more in depth description).

**Data structures**    A core framework's data structure is a Class Similarity Matrix (CSM). Let $C = \{c_1, c_2, \ldots, c_N\}$ be the set of classes in a given dataset $\mathcal{D}$ with $N$ classes on which a given model $\mathcal{M}$ is being trained. $CSM$ is an $N \times N$ matrix that stores the pairwise similarities between classes. Each element $CSM_{ij}$ of the matrix quantifies the similarity between classes $c_i$ and $c_j$ ($CSM_{ii} = 1$, in our analysis, diagonal is excluded). Using this basic notion of a CSM, we can define its 3 variants:

- **Network Class Similarity Matrices (NCSMs)** - it represents a *direct* and *structural* approach to similarity estimation from the perspective of $\mathcal{M}$. To obtain this matrix, we use the learned class representations from a final classifier of a deep network $\mathcal{M}$ (Mopuri et al., 2020; Nayak et al., 2019) (previously used for knowledge distillation and adversarial testing). Each neuron $c$ of the classification layer corresponds to one of the classes - $c$. A vector $w_c$ of weights connecting neuron $c$ to the penultimate layer can be treated as a class template (learned representation in this layer's feature space) of class $c$. $w_c$ is represented as $w_c = [w_{c1}, w_{c2}, \ldots, w_{cn}]$, where each $w_{ci}$ corresponds to the weight connecting the $c$-th neuron to the $i$-th neuron in the penultimate layer. Similarity between templates of classes $i$ and $j$ can be computed with cosine similarity (CS): $\text{CS}(i, j) = \frac{w_i^T w_j}{||w_i|| ||w_j||}$. Computing the similarities between all $N$ class pairs and scaling them to range $\langle 0, 1 \rangle$ results in the **NCSM**. We scale the matrices for numerical comparisons, while we use raw values for visualization. In contrast to more standard similarity estimation approaches, *we do not need any test data* to compute similarity, making the method significantly faster than the approach based on data samples (Huang et al., 2021) or confusion matrices (Bilal et al., 2017).

- **Confusion-based Class Similarity Matrices (CCSMs)** - it represents an *indirect* or *functional* approach to similarity estimation from the perspective of $\mathcal{M}$. Let **CM** be the confusion matrix, where $\mathbf{CM}_{ij}$ is the number of instances of class $c_i$ that are classified by $\mathcal{M}$ as class $c_j$. **CM** is an $N \times N$ matrix. To create a **CSM** from **CM**, we first normalize each row of **CM** (each row sums to 1), and then fill the diagonal with value 1. This results in a **CCSM**. **CCSM** is the only **CSM** used in our experiments that can be asymmetric, meaning that $\mathbf{CSM}_{ij} \neq \mathbf{CSM}_{ji}$ in general. The reason is that the confusion between class $c_i$ being classified as class $c_j$ may differ from class $c_j$ being mistaken as class $c_i$.

- **Semantic Class Similarity Matrices (SCSMs)** - it is a *similarity reference* used by our framework. Semantic relations approximate collective human similarity judgments, as they capture structured connections that humans intuitively recognize (common functionality, appearance, evolutionary traits). Semantic similarity is a relation between terms with a similar meaning (Kolb, 2009). Semantic similarity can be measured e.g. via WordNet (Miller, 1998) similarity measures (Pedersen et al., 2004; Leacock & Chodorow, 1998; Wu & Palmer, 1994). We use path similarity in our study, as according to (Kolb, 2009), it outperforms other measures in terms of correlation with human judgment of semantic relatedness. We use WordNet's rich semantic relations over human judgments. Its large lexical database better supports generalization across different datasets, increasing the chance of finding relevant representations. The advantages of WordNet path similarity are also its clear formulation, a consistent score due to derivation from a fixed lexical database and an objective and systematic approach to defining relationships. By computing the similarities between all categories in $\mathcal{D}$ (expressed via WordNet nodes), we obtain the **WordNet CSM (WNCSM)**, which is a special case of a SCSM (see computation method in Appendix B).

**Core methods**    Using different CSMs, we construct a set of key methods of our framework.

- **Visualization of CSMs** - a qualitative method for graphical inspection of how the similarity perception and hierarchy look. While SCSMs are constant in time, both the NCSMs and CCSMs measured after different training steps. It can be used to visualize how the perception changes with time and how is developed from the beginning of the training to its end. In the main paper, we present some chosen matrices (measured after epochs 1, 5, 25 and 200), however the matrices for all epochs can be used to create a full animation of the similarity development during training.

- **Similarity Alignment Index (SAI) Curves** - Similarity Alignment Index (SAI) measures to what degree 2 Class Similarity Matrices are similar, thus how two different similarity perceptions are aligned. Let $\mathbf{CSM}_1$ and $\mathbf{CSM}_2$ be 2 matrices of size $N \times N$. The first step is to exclude the diagonal elements from both matrices: $\mathbf{CSM}'_1 = \mathbf{CSM}_1 - \text{diag}(\mathbf{CSM}_1), \mathbf{CSM}'_2 = \mathbf{CSM}_2 - \text{diag}(\mathbf{CSM}_2)$. The second step is to Normalize the remaining elements of both matrices to the range $\langle 0, 1 \rangle$. After applying these modifications, the two matrices can be compared with a chosen similarity measure to obtain $\mathbf{SAI}(\mathbf{CSM}_1, \mathbf{CSM}_2)$. We chose Cosine Similarity due to its frequent usage for high-dimensional data. We name the plots of SAI as a function of a number of epoch as **Similarity Alignment Index (SAI) Curves**. They allow to observe how the similarity perception of a given network changes during training. Below, we define possible SAI variants:

  - **SAI(NCSM, SCSM)** - measures the alignment between the direct similarity perception of a network with the semantic similarity. Due to an expected dense structure of Network- and WordNet CSM variants, we also use Structural Similarity Index (SSIM) (Wang et al., 2004). It focuses on capturing structural information (changes in luminance, contrast, and correlation) to model human visual comparison. Cosine similarity and SSIM are better matched to asses visual quality than distance measures (Mean Squared Error, Mean Absolute Error) (Wang & Bovik, 2009), but we also provide their results in Appendix C. **SAI(NCSM, SCSM)** *examines how well a network's similarity perception corresponds to semantic similarity, indicating if its understanding of relationships aligns with semantic reasoning*. Low **SAI(NCSM, SCSM)** values suggest that a network's perception does not align with semantics. Its high values imply they are well aligned. Changes in SAI indicate whether a network's perception converge with semantic similarity during learning or diverge from it, suggesting the use of more machine-like reasoning for categorization.
  - **SAI(NCSM, CCSM)** - measures to what extent the direct similarity perception of a network aligns with its indirect perception. **SAI(NCSM, CCSM)** *evaluates how well a network's similarity perception aligns with its own mistake patterns. The aim is to determine whether the errors are predictable and stemming from perceived similarity*. Indirect and direct perception should be as similar as possible to obtain predictable mistakes (predictable means mistakes between highly similar classes).
  - **SAI(CCSM, SCSM)** - measures to what extent the indirect similarity perception of a network aligns with semantic similarity. *It assesses how closely the network's mistake patterns align with semantic similarity, indicating whether its errors make sense from a semantic perspective.* Better alignment means more reasonable errors.

- **Inverse Dissimilarity Metric (IDM) Curves** - Dissimilarity Metric is a metric introduced in work (Filus & Domanska, 2023) primarily for the assessment of the damage caused by adversarial attacks. It can be interpreted as the mean similarity shift between the ground truth label and the post attack label. In this work, we generalize it to a variant that computes the mean similarity shift between the ground truth label and the predicted label. We also propose to use an inverse version of this metric as an extension of accuracy assessment (the metric takes higher values when accuracy increases or/and when the mistakes are closer in a given Class Similarity Matrix to the ground truth). The metric can be computed as follows. To generate the standard DM values, a given **CSM** is used. Each row is sorted in a descending order and a matrix with ids of the classes belonging to the particular similarity values is obtained - **Sorted Class Similarity Matrix (SoCSM)**. Each $c$-th row stores classes with the decreasing similarity values for the $c$-th class. For each sample, we take the ground truth label $i$ and the prediction $j$. We check in the SoCSM at which index label $j$ is placed in the $i$-th row. The larger the index, the more dissimilar the label is to the ground

truth. As our target is to use DM for accuracy inspection, we transform it to obtain the Inverse variant: $IDM = 1 - DM$. Along with IDM, we use a variant that considers only the cases in which a given network returned an incorrect prediction (**IDM's errors only variant**). This can be approximated as $\frac{DM}{1-accuracy}$. Plotting the IDM value as a function of the epoch number shows how the accuracy and the mistakes being made change in relation to similarity. Depending on which **CSM** is used to obtain the **SoCSM**, we define:

- **Network-based IDM (NIDM)** - The inverse version of the original DM metric proposed in (Filus & Domanska, 2023). It uses Network CSM to measure the inter-class similarities. It measures how accurate is the network in terms of the perceived similarity (the indicator of predictability of mistakes). The errors-only variant allows to measure whether the network's similarity perception and its mistakes are related, and to what extent this perception impacts the mistakes. This is a more local approach than SAI(NCSM,CCSM). Increasing NIDM suggests that the network starts to make mistakes between categories it perceives as increasingly similar.
  - **WordNet-based IDM (WIDM)** - it is our original modification of DM. Instead of NCSM, it uses WCSM, therefore it can be treated as a semantic version of IDM. It can be interpreted as follows: how accurate is the network (even if it makes mistakes) in terms of semantic similarity. The errors-only variant allows to focus on the semantic similarity of the mistakes made. Increasing WIDM suggests that the network starts to make mistakes between categories that are increasingly semantically similar.

- **Weights Similarity Index (WSI) Curves** - Weights Similarity Index (WSI) is a mean of some specific elements of an NCSM. Depending on which values are used to calculate it, the index describes a different aspect of the network's similarity perception. These can be computed after each training step and then used to plot the WSI Curves. Below, we formulate and interpret different variants of WSI:
  - **Mean WSI** - The Mean WSI $WSI_\mu$ is computed as the mean value of all elements in the upper triangle of the similarity matrix: $WSI_\mu = \frac{2}{N(N-1)} \sum_{i=1}^{N-1} \sum_{j=i+1}^{N} \mathbf{NCSM}_{ij}$. Its curves show how the similarity of weights changes overall. If it increases, it means that the representations of classes are pulled towards each other. In the opposite case - they are pushed away from each other.
  - **Max WSI** - it represents the mean maximum similarity of classes. For each class $i$, let $\mathcal{S}_{\geq Q_i(0.95)}(i)$ represent the set of similarities that are larger or equal than the quantile $0.95$, excluding self-similarity ($S_{ii}$). Max WSI is the mean value of the averaged similarities for each class $i$: $WSI_{max} = \frac{1}{N} \sum_{i=1}^{N} \frac{1}{|\mathcal{S}_{\geq Q_i(0.95)}(i)|} \sum_{j \in \mathcal{S}_{\geq Q_i(0.95)}(i)} S_{ij}$. This index's curves represent how the similarity of classes perceived as the most similar by a given network changes. It can be treated as an index of changes in the local similarity of classes. Increases suggest discovering highly similar classes (their representations are pulled towards each other).
  - **Min WSI** - it represents the mean similarity of the most dissimilar elements for each class. For each class $i$, let $\mathcal{S}_{\leq Q_i(0.95)}(i)$ represent the set of similarities that are less or equal than the quantile $0.05$. Min WSI is the mean value of the averaged similarities for each class $i$: $WSI_{max} = \frac{1}{N} \sum_{i=1}^{N} \frac{1}{|\mathcal{S}_{\leq Q_i(0.95)}(i)|} \sum_{j \in \mathcal{S}_{\leq Q_i(0.95)}(i)} S_{ij}$. This index and its curves represent how the networks learns that some classes are highly dissimilar (their representations are pushed away).

## 4 EXPERIMENTS

In our experiments, we examine the similarity perception of 2 standard CNNs (ResNet18 (He et al., 2016), MobileNetV2 (Sandler et al., 2018)), 1 CNN modernized with ViT-inspired techniques (ConvNeXt-T (Liu et al., 2022b)), 2 ViTs (ViTB (Dosovitskiy, 2020), SwinV2T (Liu et al., 2022a)) and 1 hybrid model (attention blended with convolutions - MaxViTT (Tu et al., 2022)). They represent older and more recent state-of-the-art models. In the main body of the paper, we experiment with the Mini-ImageNet (Vinyals et al., 2016) dataset. It is a version of an original ImageNet (Russakovsky et al., 2015) with 100 classes consisting of leaf-only categories. In Appendix A, we describe these elements in more detail. We also present further numerical results obtained on

Mini-ImageNet (App. C), CIFAR100 (App. D) and some additional qualitative results obtained for CIFAR10, ImageNet-1k and COCO 2017 as a proof of generalizability of our findings (App. F). **The aim of our experiments is to answer the following research questions:**

**RQ1:** How does the network's direct similarity perception change during training?

**RQ2:** Does the network similarity perception align with semantic similarity during training?

**RQ3:** Do the confusion patterns follow the network's similarity perception during training?

**RQ4:** Do the confusion patterns follow semantic similarity during training?

## 4.1 HOW DOES THE NETWORK'S SIMILARITY PERCEPTION CHANGE DURING TRAINING?

We examine the Weights Similarity Index (WSI) Curves to see how inter-class similarities change during training (Fig. 1). For the majority of networks, mean similarity slightly increases at the beginning of training to quickly decrease toward the negative cosine similarity. The minimum values are close to 0, suggesting the pursuit to achieve orthogonality. 2 standard CNNs behave significantly different from other models (ResNet18, MobileNetV2) with Mean WSIs growing in a logarithmic fashion with values close to 0 (see the results for 2 more CNNs in App. G - they support the finding that this interesting shape of Mean WSI is characteristic for CNNs and differs them from ViT and hybrid models). For Max WSI, all networks show an increase at the beginning of training. It reaches a peak at app. 15 epochs for CNNs, and 40 for ViTs). The curves are mirrored with respect to the x-axis for Min WSI. The results for the hybrid models are closer in terms of the curve shape to the ones of CNNs and in terms of the final values - to ViTs. It shows that hybrid models combine dynamics of both architectures. The results suggest the existence of a *phase of network's rapid discovery of the most similar categories and the most dissimilar categories*, showing an effort to push the first ones to, and pull the latter from each other. It occurs during the early epochs, when networks obtain low classification effectiveness. After these initial gains/drops for the Max/Min variants respectively, the similarities start to decrease/grow, suggesting a 2nd phase, in which *differences are discovered between similar classes, and similarities between dissimilar classes* (suggesting a more fine-grained analysis, a deeper data understanding and pursuit of vector orthogonality). After some epochs, the perception reaches a relative stability (we call it the *stability phase*).

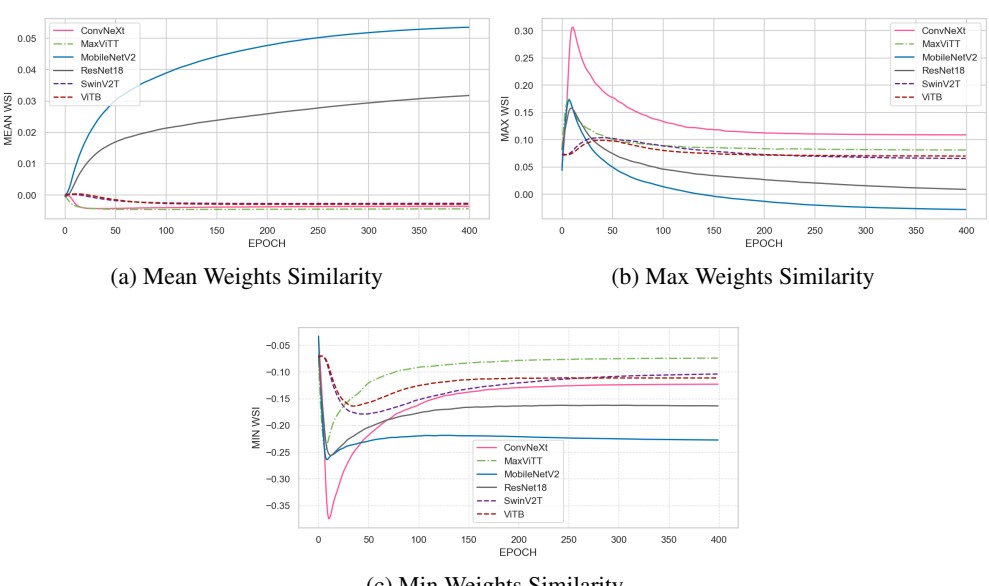

(a) Mean Weights Similarity             (b) Max Weights Similarity

(c) Min Weights Similarity

Figure 1: Mini-ImageNet: Weights Similarity Index (WSI) Curves (descr. inter-class similarities)).

## 4.2 DOES THE NETWORK SIMILARITY PERCEPTION ALIGN WITH SEMANTIC SIMILARITY DURING TRAINING?

We now aim to find out whether the similarity perception of CNNs and ViTs aligns with semantic similarity and how it changes during training. In Fig. 2 we present the Cosine- and Structure-based **SAI(NCSM, SCSM)**. For both variants, a rapid increase in the alignment between the network and semantic similarity can be observed for all the examined networks. This increase is faster for CNNs/the hybrid than for ViTs. This matches the 1st training phase observed while analyzing the WSI plots. It suggests that the dynamic learning of inter-class similarities is due to the actual semantic similarities and highly correlated structure of the world. These are discovered and learned to understand, which is in line with the categorization principles from cognitive psychology presented in the introduction. Again, after this initial growth, the alignment slightly decreases (suggesting the similarity perception 'refinement') with visible 'bumps' in the curve to practically stabilize in the later epochs (slightly earlier for CNNs/the hybrid than for ViTs) (see Appendix I). After reaching its plateau, the alignment persists to be higher for ViTs than for CNNs/the hybrid. We support these numerical results with visualizations of NCSMs for the 1st, 5th, 25th and 200th training epochs for 2 models in Fig. 3. We also present the reference WordNet matrix in Fig. B.1 in Appendix B. It is again visible that while the presented CNN (ResNet18) has already developed a clear hierarchical structure matching the one of WordNet after only 5 epochs, this structure is still barely visible for SwinV2 at that time. Nevertheless, after 25 epochs, both models present it.

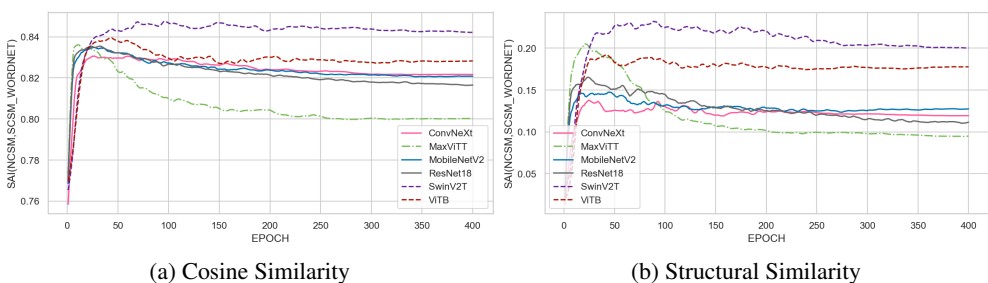

(a) Cosine Similarity          (b) Structural Similarity

Figure 2: Mini-ImageNet: **SAI(NCSM, SCSM)** Curves (network-semantic similarity alignment).

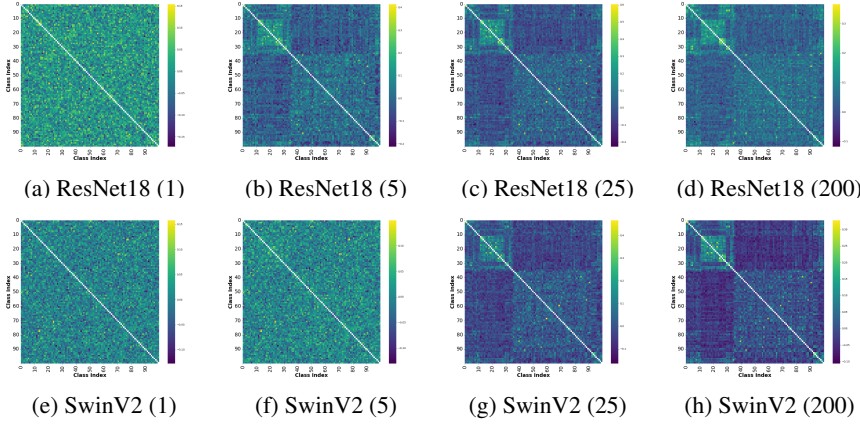

(a) ResNet18 (1)   (b) ResNet18 (5)   (c) ResNet18 (25)   (d) ResNet18 (200)

(e) SwinV2 (1)   (f) SwinV2 (5)   (g) SwinV2 (25)   (h) SwinV2 (200)

Figure 3: Mini-ImageNet: NCSMs of ResNet18 and SwinV2 (epoch number in brackets).

It can be noted here, that the highest gains in accuracy and loss optimizations happen in the first 50-100 epochs (accuracy and loss curves in Fig. B.2 of Appendix B), thus this phase is longer than the phase of the highest gains in the similarity optimization (app. 25 epochs). This suggests that the network's discovery of the inter-class similarities is responsible for the initial high gains in the performance, while the *similarity refinement* phase corresponds to the second part of the highest

accuracy/loss gains. The drops in the SAI curve, imply the emergence of other semantic relations in networks not captured well in WordNet (physical proximity, meronymy, containment, etc.) proven to exist in the trained networks (Bilal et al., 2017). This drop can be also caused by pulling the most similar classes' templates from each other to minimize the mistakes between them by focusing on their differences (visible in Fig. 1) and maximizing the task-specific performance.

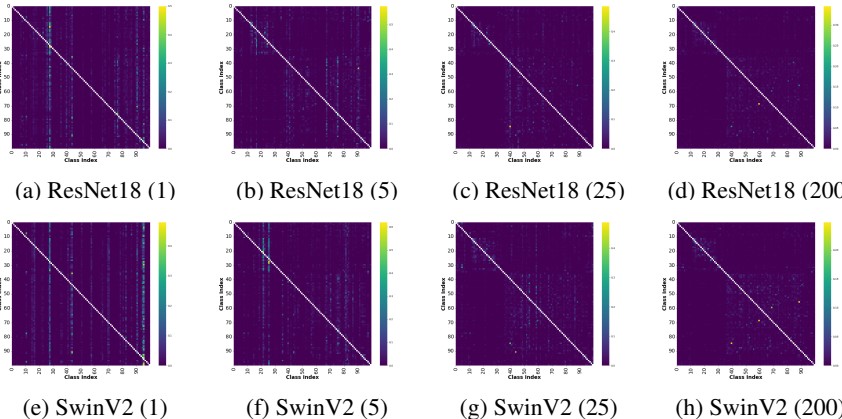

(a) ResNet18 (1)  (b) ResNet18 (5)  (c) ResNet18 (25)  (d) ResNet18 (200)

(e) SwinV2 (1)  (f) SwinV2 (5)  (g) SwinV2 (25)  (h) SwinV2 (200)

Figure 4: Mini-ImageNet: CCSMs of ResNet18 and SwinV2 (epoch number in brackets).

### 4.3 DOES THE CONFUSION MATCH NETWORK SIMILARITY PERCEPTION DURING TRAINING?

To answer this question, we start with the visualizations of Confusion-based CSMs (see Fig. 4). After some epochs, CCSMs start to reflect a similar box-diagonal structure as NCSMs shown in Fig. 3. This structure is less dense than the one of NCSMs. It also needs more epochs to be clearly developed, e.g. for the 5th epoch of ResNet18, the confusion (thus the indirect similarity estimation) can be observed even out of the basic-level categories (see the off-diagonal 'noise' and visible vertical 'stripes' in the CCSMs). It suggests that while the networks quickly learn the similarities between categories, they need more epochs to align their mistakes with their similarity judgments, e.g. via improving on more 'atypical' or 'difficult' samples.

To further examine how the errors align with the network's perception, we present the **SAI(NCSM, CCSM)** curves in Fig. 5. The plot shows that both similarity perception estimates (direct - NCSM, indirect - CCSMs) align closely, peaking around the 25th epoch. The fact that confusion-based CSMs have a sparser structure than NCSMs results in lower SAI values (visible when we compare CCSMs in Fig. 4 and NCSMs in Fig. 3). The behavior of the ViT-CNN hybrid and the ViT-inspired CNN lie very close to each other. The curves of CNNs are initially steeper than for ViTs. Then, an alignment decrease occurs and stabilization.

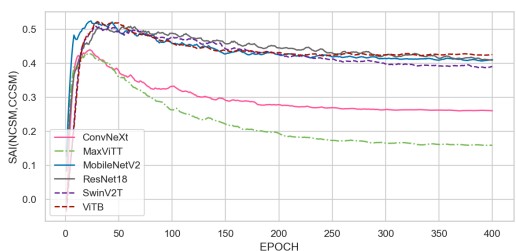

Figure 5: Mini-ImageNet: **SAI(NCSM, CCSM)** (network's similarity-mistakes alignment).

The decline is caused by the smaller number of mistakes (we now examine further this drop).

In Fig. 6, we provide the plots of the Network-based IDM Curves. These plots can be used to analyze the mistakes being made by networks more locally (how distant are the predictions being made in the perceived class space from the ground truth). The basic IDM plot confirms the previous results and shows that the network quickly starts to predict classes from the closest perceived neighborhood of a given class (thus its decisions are guided by its similarity perception). The plots reach their peak and align after around 100 epochs for all the examined networks. Surprisingly, the errors-only IDM reveals that after reaching its peak, the IDM values slightly drop, indicating the network is making

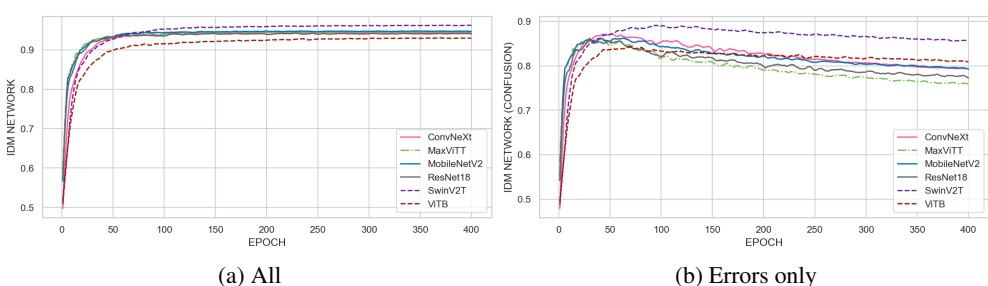

(a) All                                    (b) Errors only

Figure 6: Mini-ImageNet: Network-based IDM (Inverse Dissimilarity Metric).

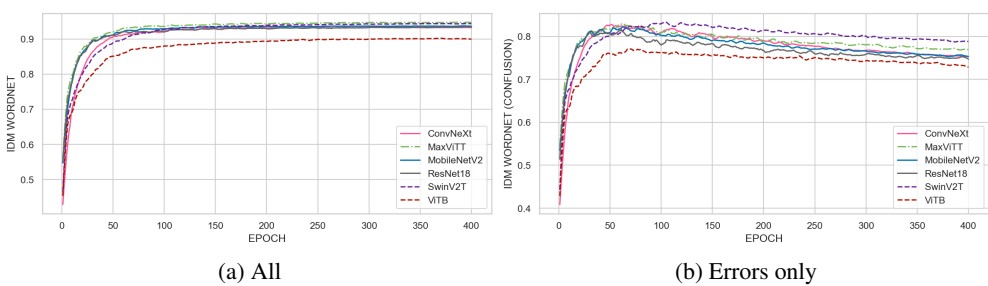

(a) All                                    (b) Errors only

Figure 8: Mini-ImageNet: WordNet-based IDM (Inverse Dissimilarity Metric).

errors between categories it perceives as less similar. The reason can be the following. The network's accuracy is already high then, having eliminated more 'obvious' mistakes. It is now tackling more challenging and less typical samples, or even potential noise from mislabeled data (we elaborate on it in App. H). Despite the drop, networks still confuse classes perceived as relatively similar.

### 4.4 DO THE CONFUSION PATTERNS MATCH SEMANTIC SIMILARITY DURING TRAINING?

To answer this question, we use the Similarity Alignment Index Curve for the Confusion-based similarity and the WordNet-based similarity - **SAI(CCSM, SCSM)** as a more global measure, and WordNet-based IDM as a more local measure. In Fig. 7, we present the SAI(CCSM, SCSM) curves. The figure shows that not only do the confusion patterns follow the network's similarity perception, but also the semantic similarity. The rapid growth in the alignment also includes the first epochs of training, peaking around the 25th epoch. Then, it either decreases and stabilizes or immediately reaches stabilization. Furthermore, the initial

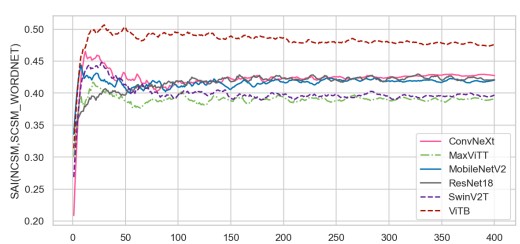

Figure 7: Mini-ImageNet: **SAI(CCSM, SCSM)** (mistakes-based&semantic similarity alignment).

rapid growth of both the Network-based and WordNet-based SAI variants is yet another, now indirect, proof that the network's similarity perception partially aligns with the semantic similarity in the very beginning of the training. The results of the WordNet-based IDM (see Fig. 8) also align with these results. For the general variant, a rapid increase in the early epochs can be noticed, followed by a quick stabilization at a high level (app. 0.9 or more). For the errors-only variant, it is visible that the mistakes initially tend to become semantically related to the ground truth, peaking before slightly declining (to a relatively high level of app. 0.75). The results also show that although the TOP1 accuracy is low in the early training, the networks are already very semantically close to the ground truth prediction. It results in a relatively good quality of a classifier (semantically 'reasonable' decisions).

## 5 DISCUSSION, LIMITATIONS, CONCLUSIONS

**Discussion** The analysis performed in this study allowed us to observe some interesting phenomena regarding the learning of vision networks. The results show that CNNs and ViTs develop a rich and hierarchical similarity perception during the course of training for standard object recognition on natural datasets. Similarity perception emerges very early in the training. This process is significantly more dynamic for CNNs/hybrid models than for ViTs. CNNs develop similarity perception rapidly and then refine it more dynamically after reaching their peak. ViTs take longer to mature their similarity perception, but once achieved, it is maintained more consistently. Using hybrid models results in the blending of both architecture features. Nevertheless, the overall dynamics of the changes in the similarity perception are close for both architectures and 3 phases of similarity development during training can be drawn: **(1)** *Initial Similarity Surge* - models rapidly discover similarities and dissimilarities between categories.; **(2)** *Similarity Refinement Phase* - models discover dissimilarities between similar classes and push them from each other, while finding relations between less similar categories.; **(3)** *Similarity Stabilization Phase* - the similarity perception becomes steady and networks focus on further elimination of mistakes Our results also show that both CNNs and ViTs make improvements in the quality of their mistakes, which we call the *mistakes refinement phenomenon*. It is particularly evident in the first phase of similarity perception development, in which the mistakes tend to show greater similarity to the ground truth.

**Limitations** Although our framework uses both direct and indirect estimations, CCSMs provide only simplified similarity structures. While this can be enough for some tasks, for tasks that require denser class similarity matrices, the network-based (structural) approach can be used instead (rich representation, no samples). Different implications of similarity could be also examined apart from the examined ones, e.g. connected to adversarial robustness (we present some quantitative and qualitative results regarding this aspect in App. K, we also comment there on the possibility of using adversarial samples as an extension of our framework). We focused on object recognition networks trained with a standard supervised procedure on ImageNet. This could limit the generalizability of our findings. To mitigate it, we provided additional results obtained for different datasets and networks trained on different supervised tasks in Appendices D and F. We also described how to extend our framework to consider networks trained on self-supervised tasks and text-image data in App. J. We also presented there some qualitative and quantitative results for DINOv2 Oquab et al. (2023) and CLIP Radford et al. (2021). We focused on relatively accurate networks, and ignored networks struggling to learn. To mitigate it, and provide additional insights, we compare a "good" and a "bad" ViT in Appendix E. Although differences in their perception occur, the results show that similarity is important even for models performing poorly in their task. While we did not analyze the impact of network capacity, initialization, or training nuances, the consistent results observed across different networks, training paradigms, and datasets demonstrate the robustness of our findings.

**Conclusions** We introduced a systematic framework for inspection of the training process of deep vision networks from different similarity-focused perspectives and numerous qualitative and quantitative methods. The results highlight the crucial importance of similarity for the categorization of deep vision networks. They support the categorization principles from cognitive psychology, which suggest that the similarity perception evolves over time, revealing the structured and correlated nature of real-world similarities. The emergence of this perception reflects a natural and optimal/suboptimal use of available finite resources of a system. Our framework and the gathered insights provide valuable contributions to the field of explainable artificial intelligence by enhancing the understanding of model decision-making and learning processes. They also contribute to the area of model evaluation with possible practical implications. The results suggest that similarity perception impacts the learning performance, mistakes being made and even adversarial robustness. It can be possibly used in other fields of deep learning outside of vision as well. We anticipate that comparable similarity structures can also be observed and used in the training of language models, audio processing models, and, in the next step, also in the training of classification models for use in e.g. autonomous vehicles. We hope that our study will prompt the community to dig deeper into these topics and use our insights to propose e.g. new training similarity-related schemes (e.g. loss functions based on similarity metrics). We will also focus on it in our future work. Only by understanding the operation of deep neural networks, we will be able to fully use their potential.

**Reproducibility Statement**  To ensure reproducibility, we provide the implementation of our framework at `https://zenodo.org/records/13860285` and also in the Supplementary Materials for this submission. We also attach there the configuration files of the models used in our study (they contain the necessary parameters, the defined architectures, the random seeds etc.). We also attach the partial results used to generate the final results (plots, matrices etc.) for the study. We will release all of these materials as a GitHub repository upon acceptance. We also provide some more details for reproducibility in the Appendices.

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

## A DATASETS AND MODELS USED IN THE STUDY

### A.1 DATASETS USED IN THE STUDY

In the main part of our paper, we use **Mini-ImageNet** (Vinyals et al., 2016). It is a version of ImageNet (Russakovsky et al., 2015) with 100 classes randomly chosen from the original dataset. ImageNet and its versions are well suited for the similarity related research, as they were created based on the semantic hierarchy of WordNet (Miller, 1998). While original ImageNet included both the internal and leaf WordNet nodes, ImageNet-1k (Russakovsky et al., 2015) and Mini-ImageNet (Vinyals et al., 2016) consist of only leaves. This results in ImageNet-1k and Mini-ImageNet not including any built-in hierarchy due to their labels being at one hierarchy level, and not different hierarchy levels (which occurs in the original version). Therefore, smaller ImageNet versions are suitable for studying how vision networks represent complex information hierarchies and the similarity between their concepts. Their direct connection with WordNet eliminates any ambiguities caused by the ambiguity of text labels of other datasets, for which the WordNet IDs are not given. Moreover, they are both current and important benchmarks in computer vision.

**CIFAR100** (Krizhevsky et al., 2009) is a dataset of a similar size as Mini-ImageNet, however with significantly smaller original size of the images (only 32x32 pixels). With its relatively high number of classes, it is a good alternative to Mini-ImageNet, therefore we use it to examine whether our observations based on the Mini-ImageNet generalize to other datasets for object recognition.

Additionally, we also use some example models trained on **CIFAR10** (Krizhevsky et al., 2009) and **COCO 2017** (Lin et al., 2014). The first dataset is a small version of CIFAR with only 10 classes. Its classes are also at a more basic-level of the semantic hierarchy than the ones of CIFAR100. COCO 2017, on the other hand, is a dataset that includes the labels for object detection and segmentation, representing a dataset used for other tasks. It includes 80 different natural categories. We use the models trained on these datasets to further (qualitatively) examine the generalizability of our findings (whether the hierarchical similarity structure develops in their NCSMs).

### A.2 MODELS USED IN THE STUDY

In our main experiments, we use the following state-of-the-art models: 2 standard CNNs (ResNet18 (He et al., 2016), MobileNetV2 (Sandler et al., 2018)), 1 CNN 'modernized' with the techniques from the ViT domain - ConvNeXt-T (Liu et al., 2022b), 2 ViTs (ViTB (Dosovitskiy, 2020), SwinV2T (Liu et al., 2022a) and 1 hybrid model (MaxViTT (Tu et al., 2022) - it uses an attention model blended with convolutions). We use their implementations provided via torchvision python library. They represent older and more recent CNNs and ViTs (and models that use techniques borrowed from the contrasting architecture). We train all the models from scratch to examine how they perception of similarity changes during training under the assumption that no knowledge was present in a given classifier before training. We do not use any techniques to enforce the development of similarity perception nor hierarchy, to examine whether these phenomena do and how they self-emerge in networks trained with standard training procedures. We use standard data augmentation techniques suitable for a given model architecture: for CNNs, we use Random Resized Crop, Random Horizontal Flip, Random Rotation, Gaussian Blur, Color Jitter, Random Perspective Transformation and Random Affine Transformation. For ViTs, we also use Cutmix (Yun et al., 2019). We use a scheduler with a linear warmup and Reduce On Plateau (reproducibility: see our GitHub repository - supplementary materials and Zenodo during the revision stage - for the specific values of their parameters and random seeds). We also use some additional models in the additional experiments in the appendices. We provide their names and links to access them in the sections regarding experiments with these particular models for clarity.

Our experiments were performed on a Linux-based system within a high-powered computer center computation grid, with 2x GPGPU NVIDIA A100 and 80 GB RAM per task. The average training time per experiment was approximately 26 hours for 400 epochs. The additional complexity of computing our new metrics accounted for approximately 0.29% of epoch time computation for MobileNetV2, and it stayed consistent for other networks as well (see Tab. A.1 with the results for 3 other models), making it an insignificant addition to the total training time.

Table A.1: Computational time of our methods as a percentage of the epoch time computation for Different Networks

| Network | Epoch Time Percentage (%) |
| --- | --- |
| MobileNetV2 | 0.29 |
| DenseNet121 | 0.19 |
| EfficientNetV2B0 | 0.18 |
| SwinV2T | 0.11 |

# B  WORDNET REFERENCE MATRICES AND PERFORMANCE METRICS FOR MINI-IMAGENET AND CIFAR100

In this appendix, we provide the generated WordNet semantic similarity matrices obtained for Mini-ImageNet (which we use to experiment with in the main paper) and CIFAR100 (used in additional experiments in Appendix D). We also provide the test accuracy and loss plots for these two datasets to enable establishing the connection between the similarity and standard performance metrics.

## B.1  SEMANTIC CLASS SIMILARITY MATRICES OBTAINED FOR WORDNET AND THE EXAMINED DATASETS

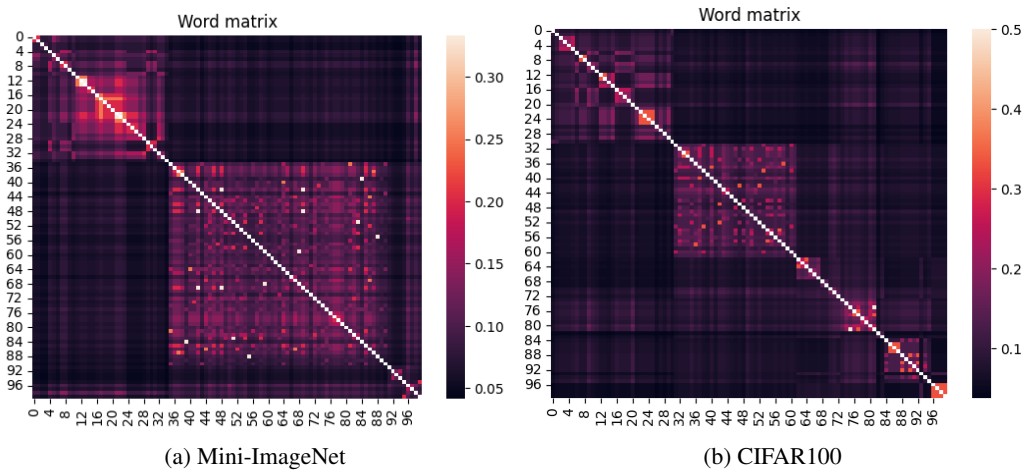

(a) Mini-ImageNet                    (b) CIFAR100

Figure B.1: Semantic Class Similarity Matrices (SCSM), precisely the WordNet Class Similarity Matrices (WCSM) obtained for Mini-ImageNet and CIFAR100. A clear hierarchical structure is visible for both datasets.

For computing semantic similarity and obtaining Semantic Class Similarity Matrices, we use the NLTK framework (Bird et al., 2009) along with WordNet linguistic taxonomy. The semantic similarity is computed for each pair of the classes in the dataset as an inverse of the distance of the shortest path connecting them in the linguistic taxonomy (path connecting two nodes in the Word-Net taxonomy tree). This metric can take the values from 0 to 1, where higher values indicate semantically closer concepts.

In Figure B.1, we present the computed Semantic Class Similarity Matrices (SCSM), precisely the WordNet Class Similarity Matrices (WCSM) obtained for Mini-ImageNet and CIFAR100. There are two main basic-level semantic groups in WordNet: the first one (in the left upper corner of the matrix) contains different animals, and the second one - artificial objects. In the case of CIFAR100, the square in the left upper corner also corresponds to different living organisms, while the subsequent squares to different subgroups of the artificial objects and formations. Both heatmaps show the hierarchical nature of the semantic relations in the two examined datasets.

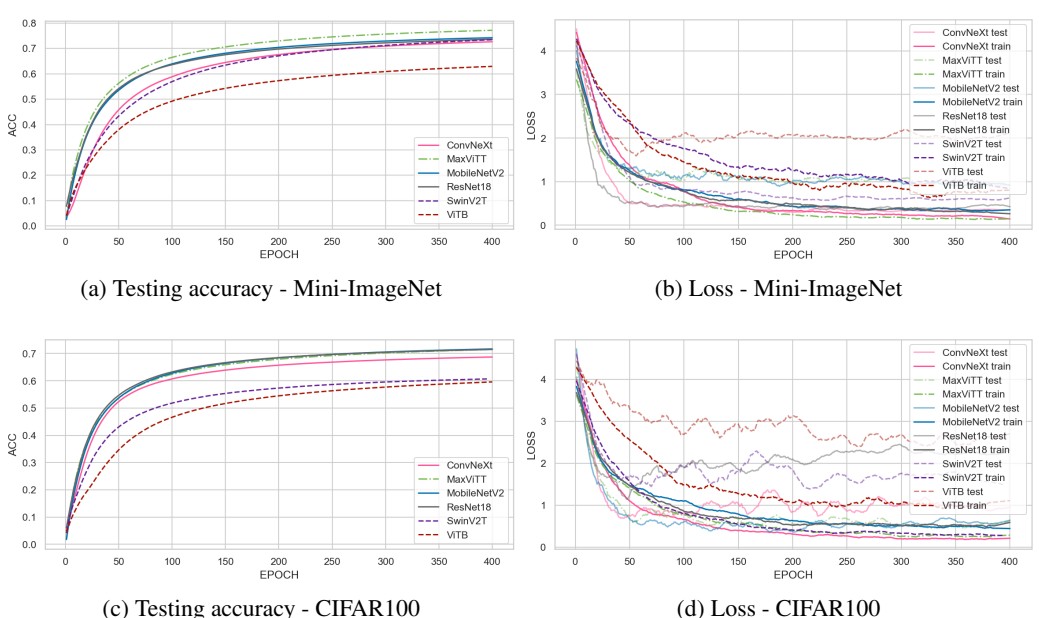

(a) Testing accuracy - Mini-ImageNet  (b) Loss - Mini-ImageNet

(c) Testing accuracy - CIFAR100  (d) Loss - CIFAR100

Figure B.2: Testing accuracy and Train/Test loss value curves for the two examined datasets: Mini-ImageNet and CIFAR100.

## B.2 THE COURSE OF THE NETWORKS TRAINING FROM THE PERSPECTIVE OF STANDARD METRICS

In Fig. B.2, we provide the plots of testing accuracy and train/test loss curves for both the examined datasets - Mini-ImageNet and CIFAR100. We used the same hyperparameters to train the models on CIFAR100 that we initially used for Mini-ImageNet for the comparison. It is visible that while the testing accuracy of all networks is rather similar for the two datasets (one exception is the SwinV2T model on CIFAR100), the loss plots show higher overfitting of the models trained on CIFAR100 compared to Mini-ImageNet.

## C ADDITIONAL NUMERICAL RESULTS OBTAINED ON MINI-IMAGENET

In this appendix, we provide the remaining results obtained for the Mini-ImageNet. In Fig. C.1, we present the obtained **SAI(NCSM, SCSM)** for the distance measures: Mean Squared Error (MSE) and Mean Absolute Error (MAE). The results show that these measure also reflect the changes in the similarity alignment between the Network's and the Semantic perception of similarity.

In Fig. C.2, we present the Network Class Similarity Matrices for the remaining models. Similarly to the qualitative results obtained in the main body of the paper for the ResNet18 and the SwinV2T models, the matrices show that a clear hierarchical similarity structure is developed faster for CNNs than for ViTs. It is also visible that in later epochs of training, ConvNeXt, similarly to ResNet18 discovers more similarities between the classes that do not belong to the main semantic groups (the 'off-diagonal noise') than the ViT model. It is also visible that ConvNeXt, similarly to ViTs (from which it incorporates some architectural features), needs more time to develop a clear similarity structure than standard CNNs. MaXViT (a hybrid model), on the other hand, needs less epochs than other ViTs to develop such a structure, further suggesting that the introduction of techniques from one architecture to another results in the intermingling of behavioral features of both architectures.

In Fig. C.3, we present the Confusion-based Class Similarity Matrices for the remaining models (ConvNeXt, ViTB, MaXViTT, MobileNetV2). Similarly to the qualitative results obtained in the main body of the paper for the ResNet18 and the SwinV2T models, the matrices show that the confusion patterns after app. 25 epochs of training reveal a hierarchical similarity structure. Again, faster for CNNs/the hybrid model than for ViTs. Especially for the categories from the animals

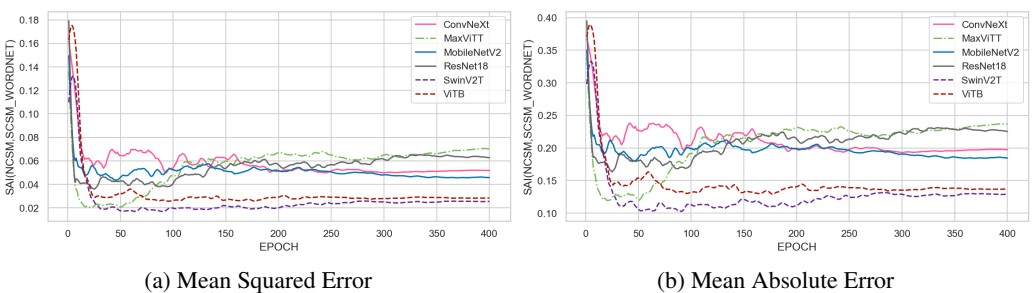

(a) Mean Squared Error
(b) Mean Absolute Error

Figure C.1: Mini-ImageNet: Similarity Alignment Index Curves based on distance measures for Network and WordNet similarity perception - **SAI(NCSM, SCSM)**. Both measures show that networks quickly develop a similarity perception that largely aligns with semantic relations. Excluding some minor drops, this alignment persists as training continues.

basic-level category, it can be observed that the mistakes are made mainly within narrower semantic categories in the later epochs of training.

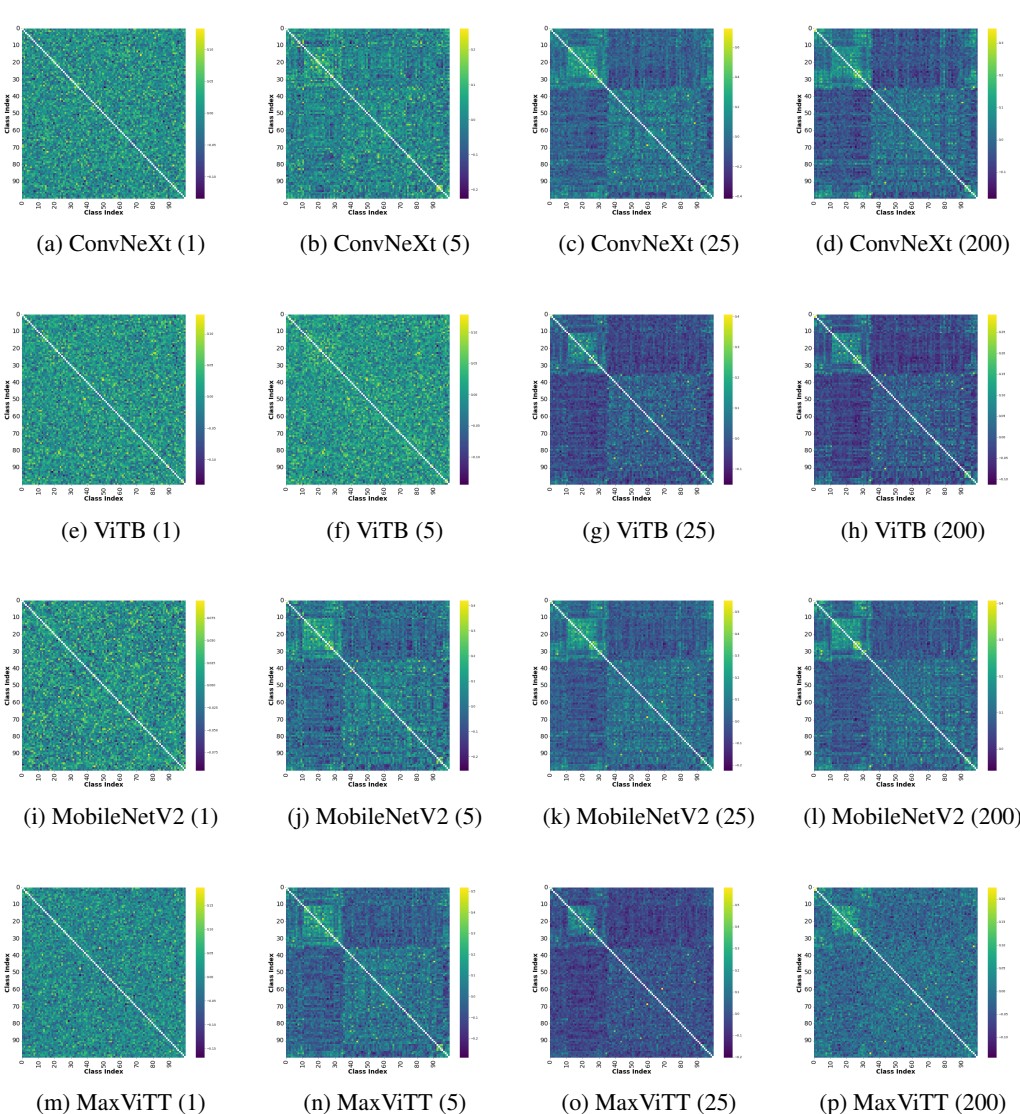

Figure C.2: Mini-ImageNet: Network Class Similarity Matrices - Network-based similarity of the remaining models for at different epochs (number - in brackets). Networks develop the hierarchical similarity perception in the early epochs (ResNet earlier than Swin). While the example ViT eliminates less significant similarities in the later epochs, more semantically unrelated categories emerge as more similar for ResNet18 (visible as off-diagonal 'noise').

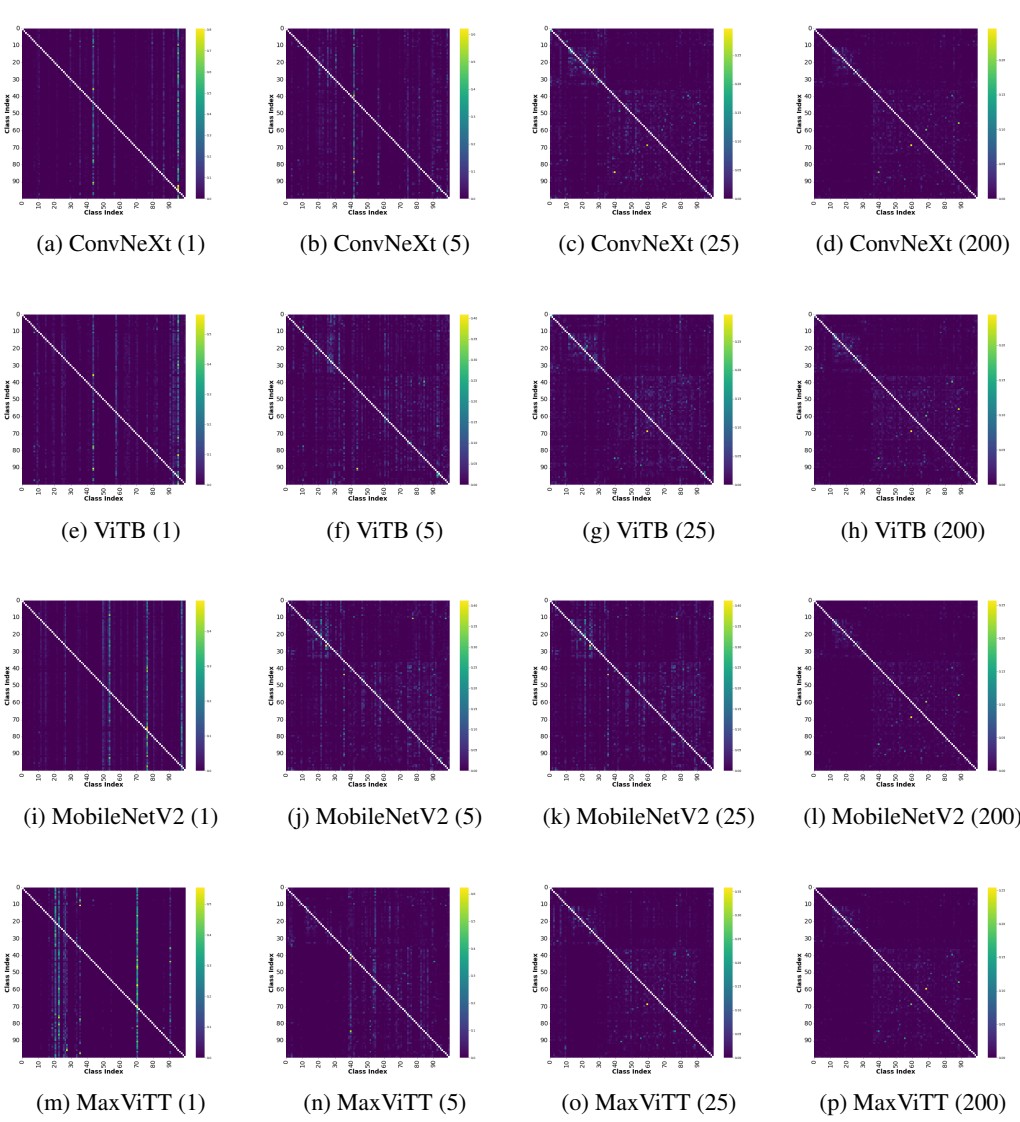

(a) ConvNeXt (1)  (b) ConvNeXt (5)  (c) ConvNeXt (25)  (d) ConvNeXt (200)

(e) ViTB (1)  (f) ViTB (5)  (g) ViTB (25)  (h) ViTB (200)

(i) MobileNetV2 (1)  (j) MobileNetV2 (5)  (k) MobileNetV2 (25)  (l) MobileNetV2 (200)

(m) MaxViTT (1)  (n) MaxViTT (5)  (o) MaxViTT (25)  (p) MaxViTT (200)

Figure C.3: Mini-ImageNet: Confusion-based similarity of the remaining models at different epochs (number - in brackets). At the beginning, both networks targets only a few distinct classes as a confusion result. They initially cover the whole space, and then smaller and smaller groups of hierarchy, making the mistakes more distributed and as a result - clearly showing the hierarchy (SwinV2 needs more epochs than ResNet18 to achieve this).

## D  EXPERIMENTS WITH CIFAR100

To increase the generalizability of our findings, we conducted experiments analogous to those performed on Mini-ImageNet on the CIFAR100 dataset. In order to focus on the impact of the training data on the behavior of the network and exclude other factors, we decided to use the same hyperparameters of the models as for Mini-ImageNet (reproducibility: the configuration files of the models used in the experiments can be found in our GitHub repository - supplementary materials and Zenodo during the revision stage). We also train our models for 400 epochs and inspect the models during the training procedure with the implementation of our Deep Similarity Inspector Framework and save the result for presentation. In this appendix, we present analogical results for CIFAR100 to those obtained on Mini-ImageNet and discuss them shortly.

### D.1  HOW DOES THE NETWORK'S SIMILARITY PERCEPTION CHANGE THROUGHOUT THE TRAINING PROCESS FOR CNNS AND VITS? IS IT IN LINE WITH SEMANTIC SIMILARITY?

In Fig. D.1, we present the results of different variants of the Weights Similarity Index (WSI) Curves obtained for the CIFAR100-trained models. The results are in line with the results obtained for Mini-ImageNet. It shows that the chosen network architecture and its hyperparameters impact the behavior of the network more than the chosen dataset. Again, for the majority of networks, Mean WSI drops with training for ViTs, a hybrid model and a ViT-inspired CNN. Standard CNNs, on the other hand, are characterized with Mean WSI increases, but with values still close to 0. Also, the Max/Min WSI variants behave in the same manner: the examined CNNs are characterized by a steep increase/decrease followed by a steep decrease/increase respectively, while the ViTs behave more steadily with their changes in similarity perception of templates. The hybrid model and the ViT-inspired CNN firstly behave similarly to CNNs, to get closer to the ViT behavior (via values) in the later stages of training.

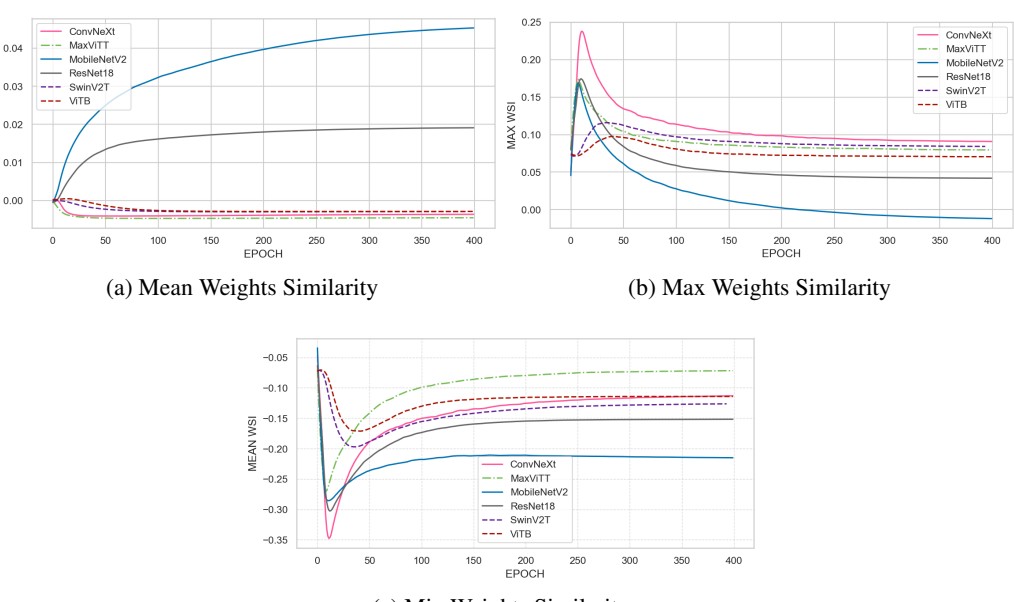

(a) Mean Weights Similarity

(b) Max Weights Similarity

(c) Min Weights Similarity

Figure D.1: CIFAR100: Weights Similarity Index (WSI) Curves. The min and max variants maintain an approximately inverse relationship. The variants also show similarities within the network families (ViTs, CNNs) in terms of the changes in the perception of the most/least similar categories.

The results of different variants of Similarity Alignment Index Curves for Network and WordNet similarity perception - **SAI(NCSM, SCSM)** presented in Fig. D.2 are also in line with those obtained for Mini-ImageNet. The alignment grows quickly in the first app. 25 epochs, to later drop slightly and stabilize. The drop is the most visible for MaxViTT and structural similarity/distance measures.

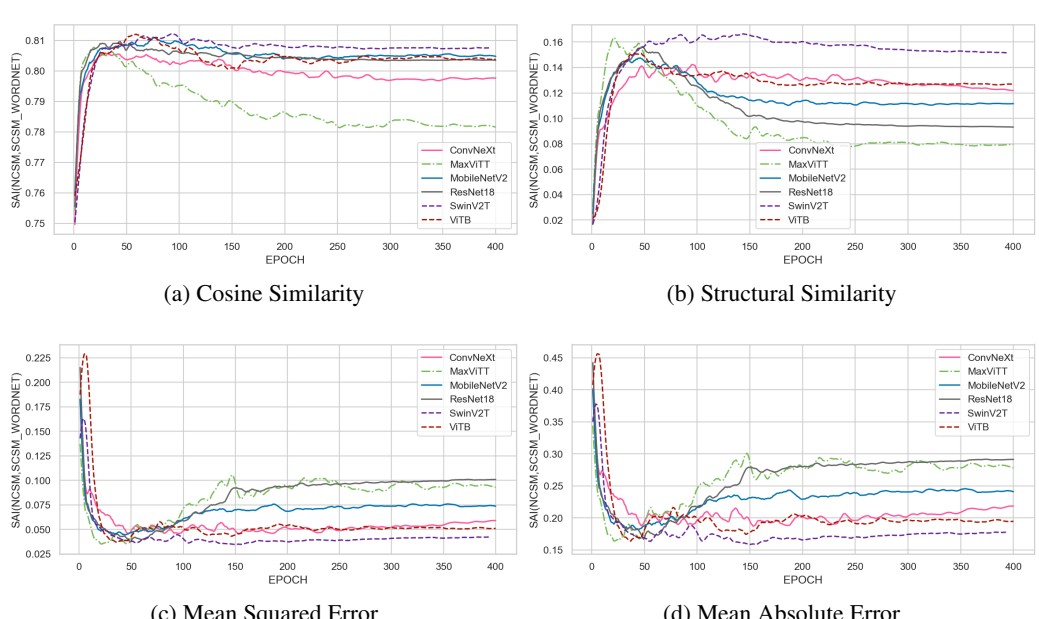

(a) Cosine Similarity

(b) Structural Similarity

(c) Mean Squared Error

(d) Mean Absolute Error

Figure D.2: CIFAR100: Similarity Alignment Index Curves for Network and WordNet similarity perception - **SAI(NCSM, SCSM)** for all possible similarity/distance measures. Both measures show that networks quickly develop a similarity perception that largely aligns with semantic relations. Excluding some minor drops, this alignment persists as training continues.

The qualitative analysis with Network Class Similarity Matrices presented in Fig. D.7 also shows similar results to the ones obtained for Mini-ImageNet. The examined CNNs quicker reveal a clear hierarchical similarity structure than ViTs (also at the very beginning of training - after app. 5 epochs).

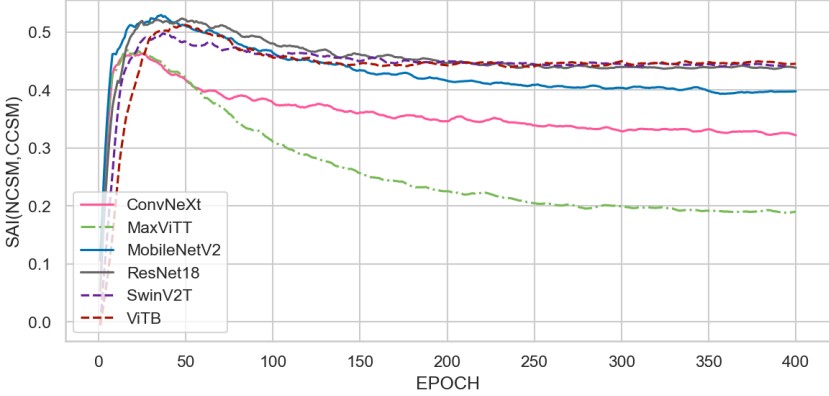

Figure D.3: CIFAR100: Similarity Alignment Index Curve between the Confusion-based similarity and the Network-based similarity - **SAI(NCSM, CCSM)**. The index rapidly grows in the very first epochs of training, reaches its maximum, then drops slightly with time.

## D.2 DO THE CONFUSION PATTERNS OF CNNS AND VITS MATCH THEIR SIMILARITY PERCEPTION THROUGHOUT THE TRAINING?

Also in this case, the results for CIFAR100 regarding the alignment of the direct and indirect similarity perception of networks confirms the observations from the main body of the paper obtained for Mini-ImageNet. All the examined networks behave practically the same as for Mini-ImageNet

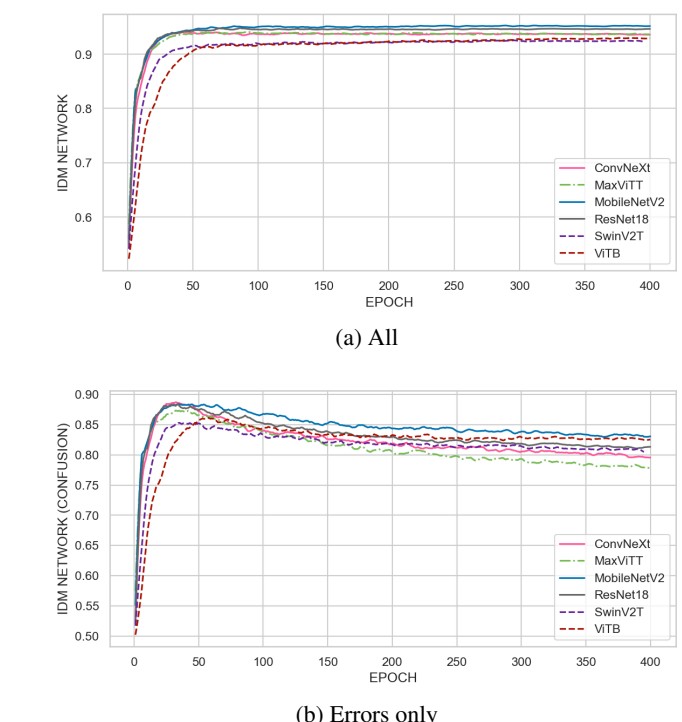

(a) All

(b) Errors only

Figure D.4: CIFAR100: Network-based IDM. The plots show that all networks quickly start to make mistakes between categories they perceive similar. After the initial gains, IDM stabilizes. Surprisingly, the errors only variant shows that with time, the networks start to make mistakes that are perceived as less similar (balanced by the increasing accuracy in the basic variant).

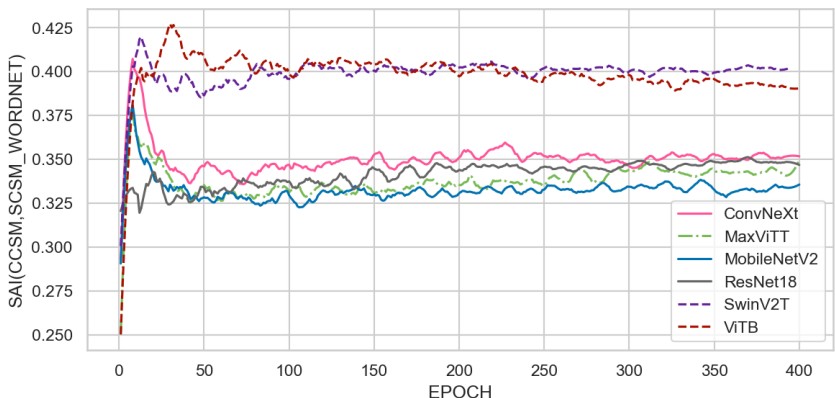

Figure D.5: CIFAR100: Similarity Alignment Index Curve for the Confusion-based similarity and the WordNet-based similarity - **SAI(CCSM, SCSM)**. The index rapidly grows in the very first epochs of training, reaches its maximum, then drops slightly with time and stabilizes.

with slightly higher values of the Similarity Alignment Index Curve between the Confusion-based similarity and the Network-based similarity - **SAI(NCSM, CCSM)** (see Fig. D.3). The hybrid model and the ViT-inspired CNN, again, can be easily distinguished from other models with their very similar to each other behavior. Similarly, the Network-based IDM Curves presented in Fig. D.4 show a very close behavior to the curves obtained for Mini-ImageNet. The small difference is that in the case of CIFAR100, the examined ViTB and SwinV2T models obtained slightly lower results for the basic variant than CNNs. Also in the case of CIFAR100, it is visible that CNNs faster align their confusions with their perception of similarity than ViTs. This observation is also supported by the

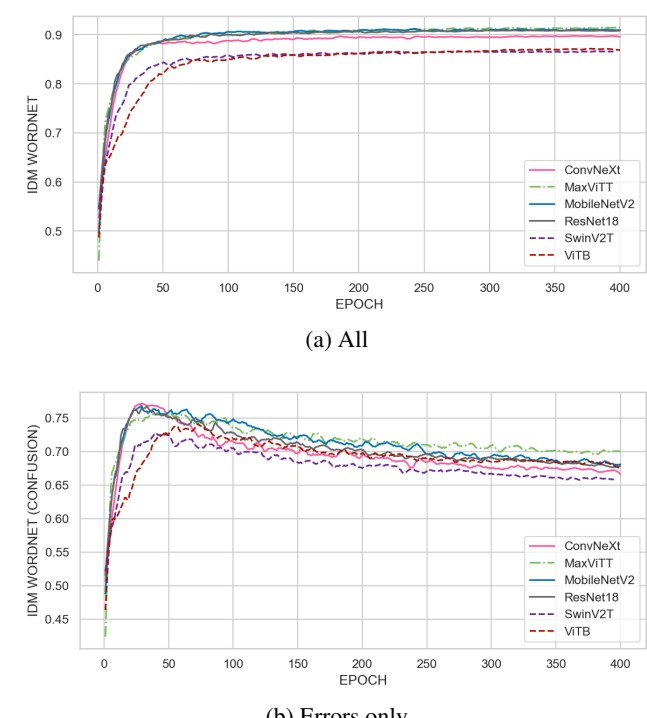

(a) All

(b) Errors only

Figure D.6: CIFAR100: WordNet-based IDM. The plots show that all networks quickly start to make semantically-related mistakes. After the initial gains, IDM stabilizes. The errors only variant shows that with time, the networks start to make mistakes that are more distant in the WordNet hierarchy (balanced by the increasing accuracy in the basic variant).

qualitative results based on Confusion-based Class Similarity Matrices (CCSMs) presented in Fig. D.8. While for the ResNet18 and ConvNeXt models, the indirect similarity perception is revealed as soon as after app. 25 epochs, more time is needed for ViTs used in the experiments. Again, it can be observed that the indirect similarity measurements via confusion matrices results in much less dense similarity matrices with only an approximate structure of similarity perception.

### D.3 DO THE CONFUSION PATTERNS OF CNNS AND VITS ALIGN WITH SEMANTIC SIMILARITY THROUGHOUT THE TRAINING?

The results obtained on CIFAR100 support our results from the main body of the paper. For this dataset, the indirect similarity patterns derived from the confusion matrices also partially align with semantic similarity. It is, first of all, visible via the visualization of the CCSM presented in Fig. D.8 and similarity to WCSM (SCSM) obtained for CIFAR100 (especially of the last CCSMs obtained for the 200th epoch). In Fig. D.5, we present the Similarity Alignment Index Curve for the Confusion-based similarity and the WordNet-based similarity - **SAI(CCSM, SCSM)**, which numerically prove this partial alignment. Again, the SAI(CCSM, SCSM) values (indirect, functional similarity assessment) are significantly lower than the SAI(NCSM, SCSM) values (direct, structural similarity assessment), which is caused by less dense structure of the CSM created based on the confusions. Nevertheless, also in this case a rapid increase in the SAI value can be noticed in the first epochs of training, followed by the similarity perception refinement and stabilization. This behavior is also reflected in the plots of WordNet-based IDM presented in Fig. D.6. The obtained values are slightly higher than the ones obtained for Mini-ImageNet, showing that in the case of CIFAR100, all the trained networks make mistakes from a narrower semantic neighborhood of the ground truth classes.

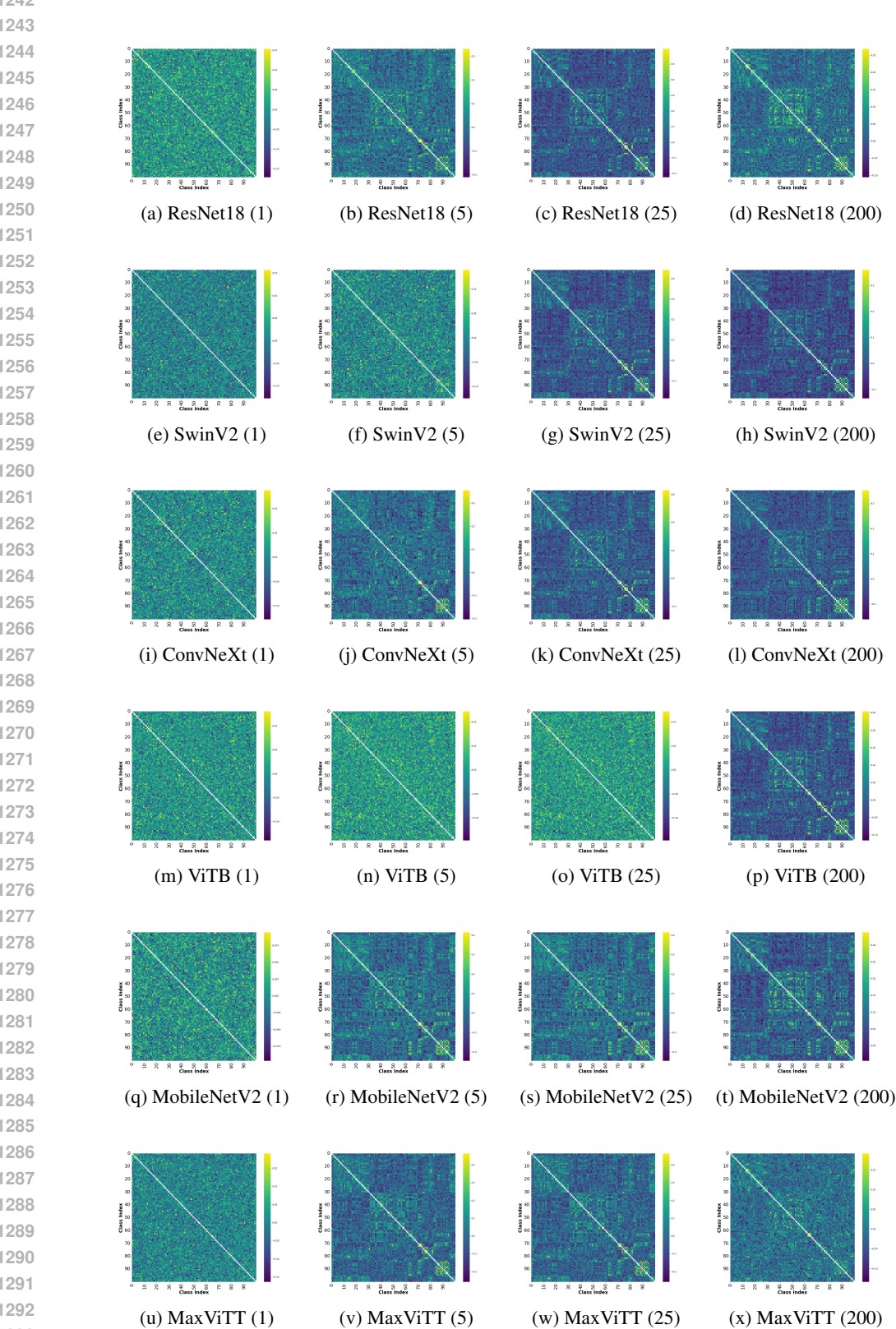

Figure D.7: CIFAR100: Network-based similarity of all models for at different epochs (number - in brackets).

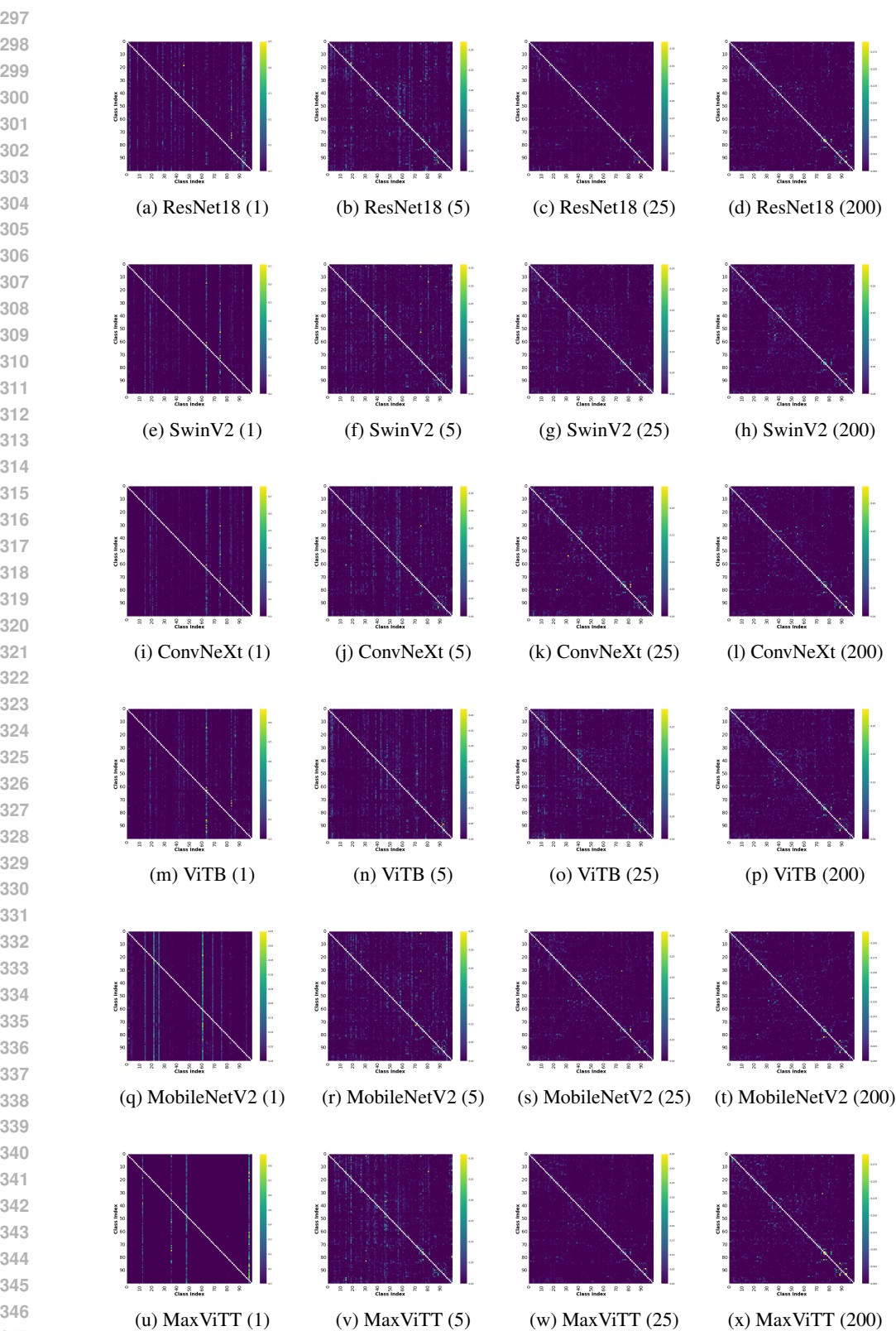

Figure D.8: CIFAR100: Confusion-based similarity of all models at different epochs (number - in brackets).

# E    DOES SIMILARITY PERCEPTION EMERGE IN "BAD" NETWORKS?

We also decided to examine a case, in which network cannot reach an acceptable accuracy in a task that it is trained for. We name such a network a "bad" network, in contrast to "good" networks used in our experiments, that achieve relatively good accuracies. We choose the same ViT as the one used in our experiments but with significantly higher learning rate, due to which the network cannot learn effectively. It is our "bad" network. For a "good" network, we take the same ViT as in our original experiments. In Fig. E.1, one can notice that "bad" ViT achieves a very poor accuracy and almost no optimization is visible for it loss function.

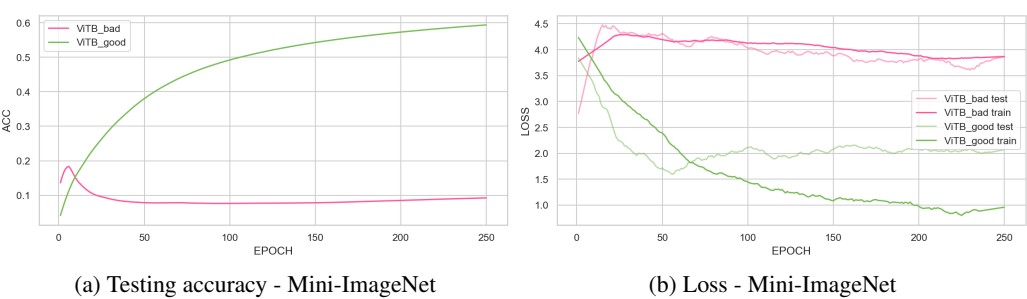

(a) Testing accuracy - Mini-ImageNet              (b) Loss - Mini-ImageNet

Figure E.1: Testing accuracy and Train/Test loss value curves for the Mini-ImageNet for a "good" and a "bad" ViT.

When it comes to the behaviour of weights and its analysis with different WSI variants (see Fig. E.2), surprisingly, the curves of a "bad" network are quite similar to the ones of a "good" network, especially the Mean variant. For the Mean variant, a main difference is that the plot is not that smooth as the one of a "good" one, but they mostly overlap. The Min and Max variants are similar in terms of the curves' shape, however they can be perceived as the "scaled" versions (they obtain significantly smaller/larger values respectively).

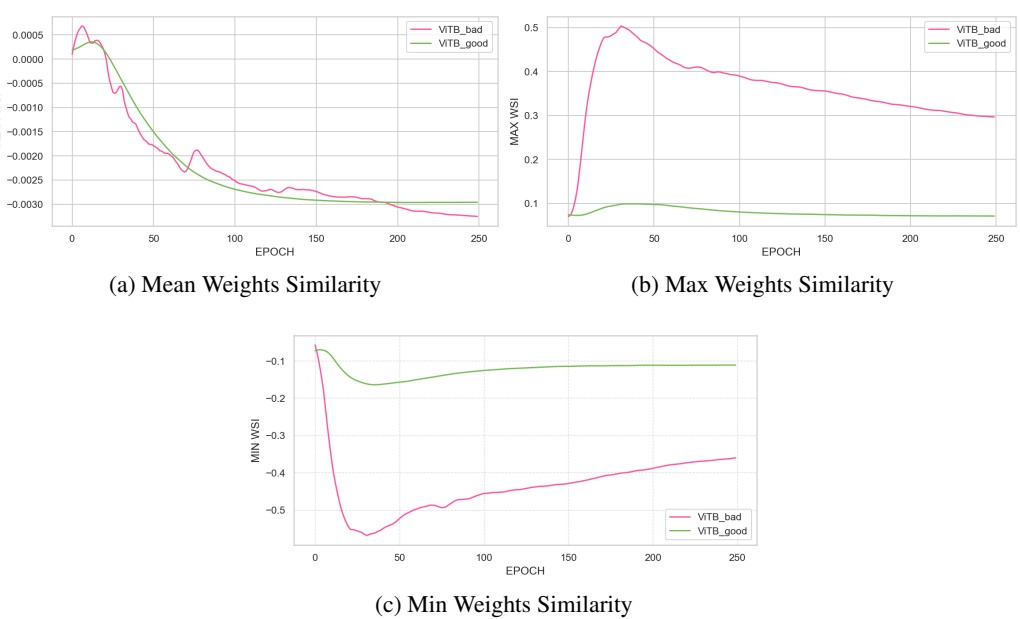

(a) Mean Weights Similarity              (b) Max Weights Similarity

(c) Min Weights Similarity

Figure E.2: Mini-ImageNet: Weights Similarity Index (WSI) Curves for a "good" and a "bad" ViT.

In Fig. E.3, we provide the SAI Curves obtained for all SAI variants for the "good"-"bad" network pair. It is visible, that all "bad" network's SAI curves behave significantly different than their "good"

network counterparts. First of all, as expected, SAI(NCSM, SCSM) and SAI(NCSM, CCSM) obtain significantly lower values. Also, the shape of the curve is different. Instead of a harmonic (increase-refinement-stabilization) curves, one can notice a sudden increase (with lower maximum peak value than for a "good" network), a sudden drop and a slow increase until the end of training. This final phase shows that even "bad" networks tend to partially improve their similarity perception and incorporate it into their operation. It further suggests that the similarity emergence is due to a highly correlated structure of the real-world objects and that it is natural for a categorization system to discover these correlations via similarities. Surprisingly, for the SAI(NCSM, SCSM), values obtained at the end of the training are higher than the ones obtained for the "good" network. It may be caused, by the fact aforementioned in the main body of the paper, that in the later epochs, "good" networks make significantly less mistakes than "bad" networks, therefore their CCSMs are very sparse. That is why, to measure the relationships between CCSMs and other CSMs, it is good to also include in the analysis the exploration of different IDM variants (which focus on a more functional and local approach to errors), which we do in the next part of this section.

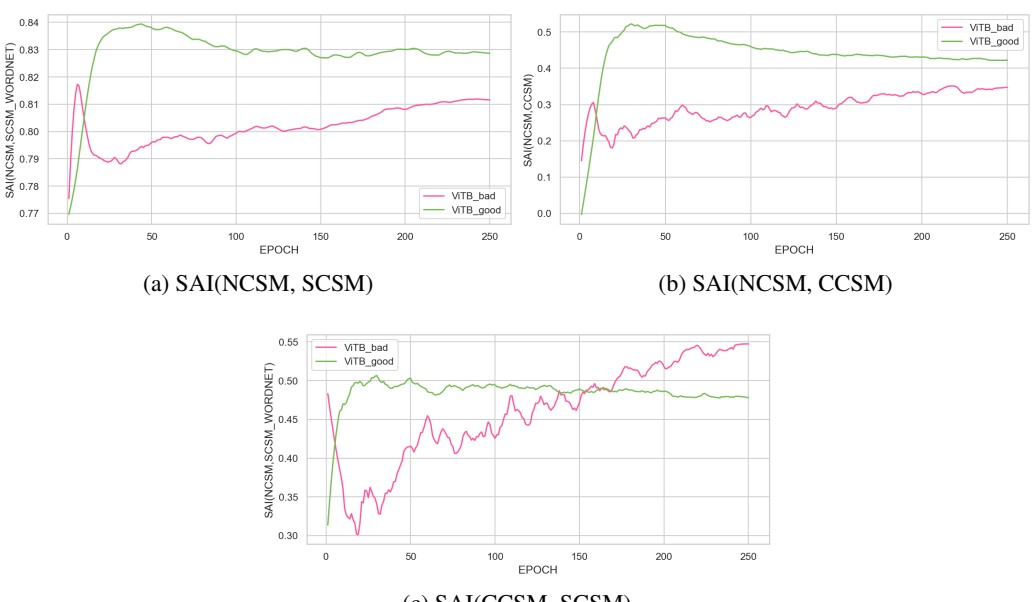

(a) SAI(NCSM, SCSM)

(b) SAI(NCSM, CCSM)

(c) SAI(CCSM, SCSM)

Figure E.3: Mini-ImageNet: All SAI Curves for a "good" and a "bad" ViT.

Now, let us focus on the quality of a "bad" network's predictions and more in-detail analysis of its mistakes. In Fig. E.4, we present all DM variants (both the network-based and the WordNet-based ones). It is visible here, that the quality of the "bad" network's prediction is much lower than the one of a "good" network, although some improvement with time can be noticed (which i barely visible in the test accuracy plot), showing that the similarity perception optimization does introduce some performance improvements in the latter training stages (the *error refinement phenomenon* is visible also for "bad" networks, but happens later in the training). This improvement (although not visible in the accuracy plot) is also visible as the decrease in the loss plot, therefore the IDM plots can be used as an explanation of this decrease. The errors-only variant shows, that although the improvements undoubtedly happen, the mistakes still are on average placed around the half of the class space (both for the network-based and the WordNet-based variants). It indicates high randomness of the mistakes being made. It is also visible in CCSMs obtained for our "bad" network in Fig. E.5. While for the 5th epoch, a hierarchy is visible, it is much less prominent for the later epochs (although some hierarchical groups are slightly visible, e.g. close to the animal classes).

To summarize, also for bad networks, some similarity phases can be defined, however they are significantly different than for "good" networks. Here, we define 2 most prominent phases (we discard some initial gains of the network at the beginning of the training, as they are most probably related to the initial accuracy/loss improvement of the network). These phases are: (1) *initial similarity drop* and (2) *stable similarity growth*.

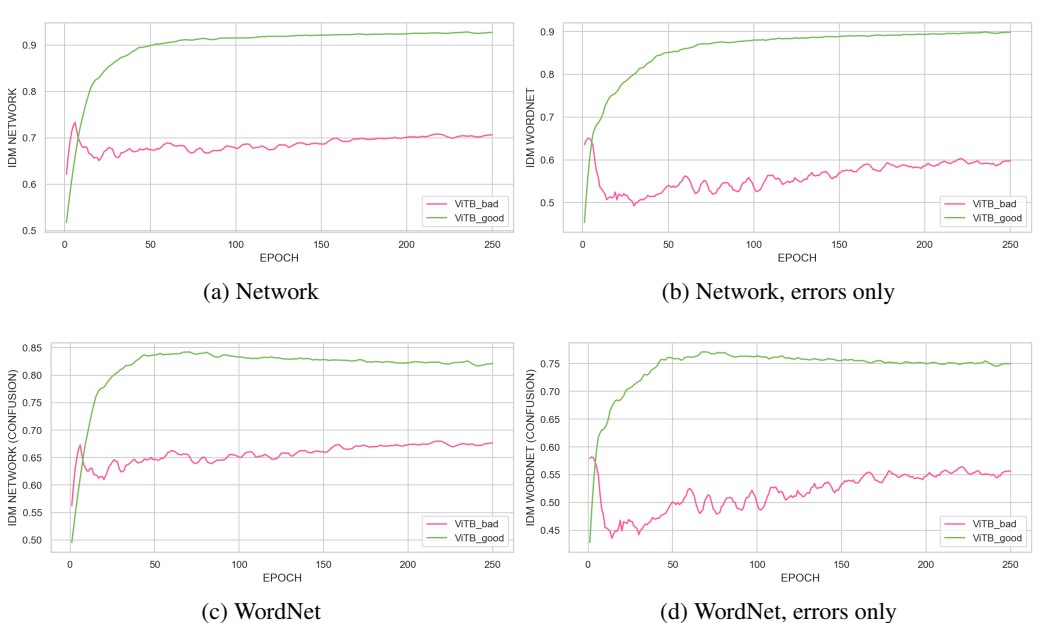

(a) Network

(b) Network, errors only

(c) WordNet

(d) WordNet, errors only

Figure E.4: Mini-ImageNet: All DM Curves for a "good" and a "bad" ViT.

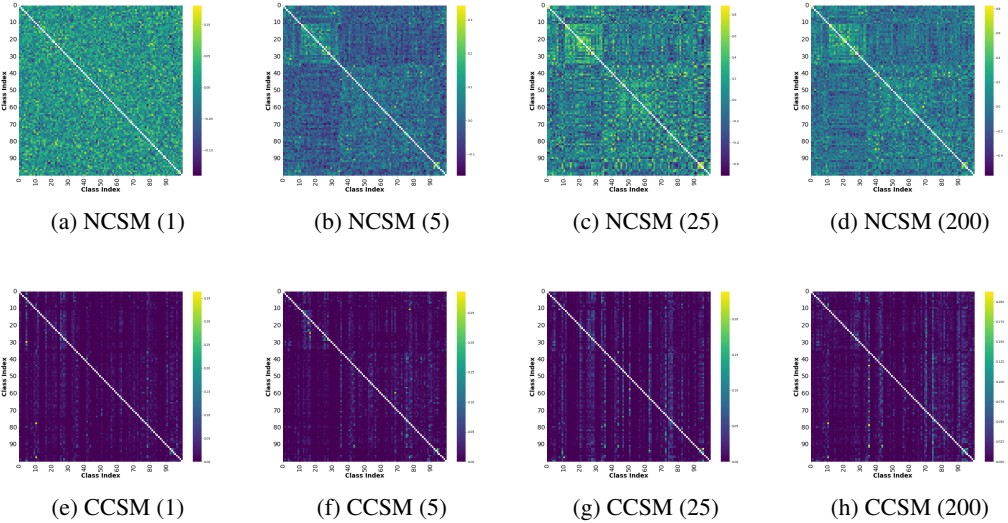

(a) NCSM (1)

(b) NCSM (5)

(c) NCSM (25)

(d) NCSM (200)

(e) CCSM (1)

(f) CCSM (5)

(g) CCSM (25)

(h) CCSM (200)

Figure E.5: Mini-ImageNet: Network Class Similarity Matrices and Confusion-based Class Similarity Matrices for a "bad" ViTB for at different epochs (number in brackets).

Our last finding can be further developed into a method of assessing the progress of the network's training and monitoring its potential overfitting (or as in "good" and "bad" networks example, as an early indicator of overall model performance). With using our metrics, different phases of the network training can be distinguished as the training progresses, and can help with early stopping, checkpointing or managing learning rate during training. It can also be further developed as a loss function component for added regularization – both of the use cases will be considered by us in our future works.

## F    IS SIMILARITY PERCEPTION EMERGENCE TIED SOLELY TO OBJECT RECOGNITION AND LARGE DATASETS?

After the initial experiments with the object recognition model trained on Mini-ImageNet (the main body of the paper) and CIFAR100, we decided to performs some qualitative experiments with models trained on different datasets and tasks to see whether our results have a potential to generalize to (1) smaller/larger datasets than the medium-sized datasets used in the study and (2) models for object detection/scene segmentation. For the purpose of (1), we generated additional NCSMs for example models trained on ImageNet-1k (a larger dataset with 1000 classes) and CIFAR10 (a smaller dataset with only 10 classes, at a higher level of abstraction than CIFAR100 and Mini-ImageNet).

In the case of ImageNet-1k, we used a trained ConvNeXt-S model from `https://huggingface.co/facebook/convnext-small-224`. We present its NCSM in Fig. F.1 along with the NCSM obtained for ConvNeXt-T from our experiments (at epoch 200). It is visible that a similar characteristic structure of the semantic categories has been developed by two models. While the ranges of the matrices values differ, the overall structure stays the same, suggesting that our results generalize also to larger datasets.

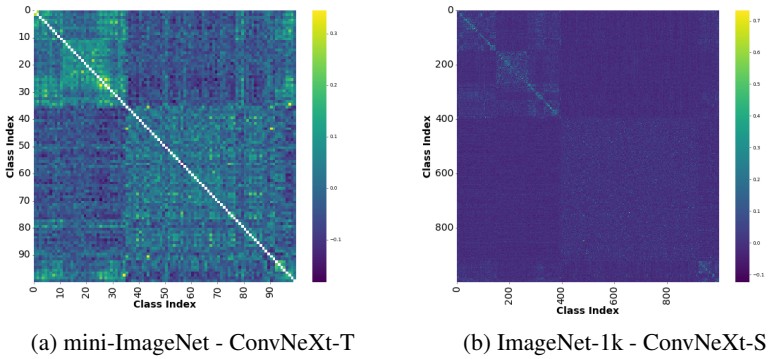

(a) mini-ImageNet - ConvNeXt-T          (b) ImageNet-1k - ConvNeXt-S

Figure F.1: Network Class Similarity Matrices obtained for 2 ImageNet versions.

In the case of CIFAR10, we trained an example small network (a simple, sequential model) that obtained app. 85% accuracy on the test set (we provide the structure of this network and the seed used for weights initialization in our GitHub repository for reproducibility - supplementary materials and Zenodo during the revision stage). Although CIFAR10 has a significantly shallower semantic hierarchy of concepts than CIFAR100 and Mini-ImageNet, also in this case clear semantic groups can be distinguished (in the left upper corners - animals: 'bird', 'frog', 'dog', 'cat', 'horse', 'deer' and the 2nd group - vehicles: 'aircraft', 'car', 'boat', 'truck'). This also supports the theories from the cognitive psychology than the similarities in the world are revealing and the visual structure of the world is highly correlated (Rosch & Lloyd, 1978; Medin et al., 1993).

We also performed a qualitative experiment with networks trained on different tasks on COCO 2017 dataset. We use the following available networks from HuggingFace: DEtection TRansformer (DETR) model with ResNet-50 backbone for object detection (`https://huggingface.co/facebook/detr-resnet-50`), YOLOS-t for object detection (`https://huggingface.co/hustvl/yolos-tiny`), MaskFormer model for COCO instance segmentation (`https://huggingface.co/facebook/maskformer-swin-tiny-coco`), Mask2Former model for COCO panoptic segmentation (`https://huggingface.co/facebook/mask2former-swin-base-coco-panoptic`) and DETR for COCO 2017 panoptic segmentation (`https://huggingface.co/facebook/detr-resnet-50-panoptic`). We provide the NCSMs generated with our framework in Fig. F.3 (it is worth noting that this method is compatible with all models that include a standard label classifier, therefore also the one present in the classifier of the object detection/segmentation networks). We also provide the WCSM for COCO in this figure. The results show that all networks used in this experiment managed to develop a clear hierarchical structure of similarity. In many parts (boxes) is is very similar to

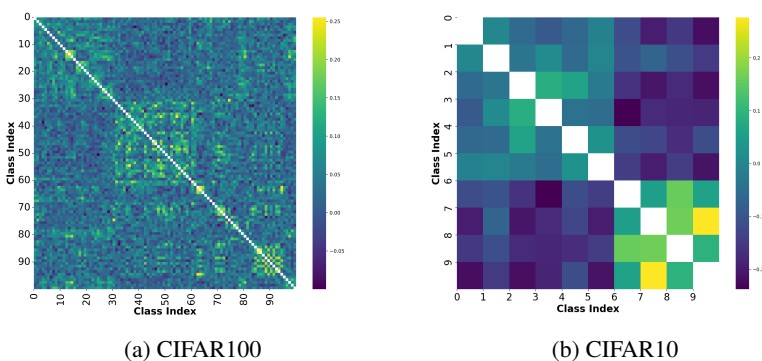

(a) CIFAR100       (b) CIFAR10

Figure F.2: Network Class Similarity Matrices obtained for 2 available versions of CIFAR.

the structure of the generated WCSMs, but when we analyze the NCSMs closer, it is visible that networks rely on more semantic relations than the ones reflected by WordNet (e.g. look how 'fire hydrant' and 'road sign' are placed together in the first square along with different vehicle types, suggesting the importance of context and co-occurence of objects in the process of similarity perception development).

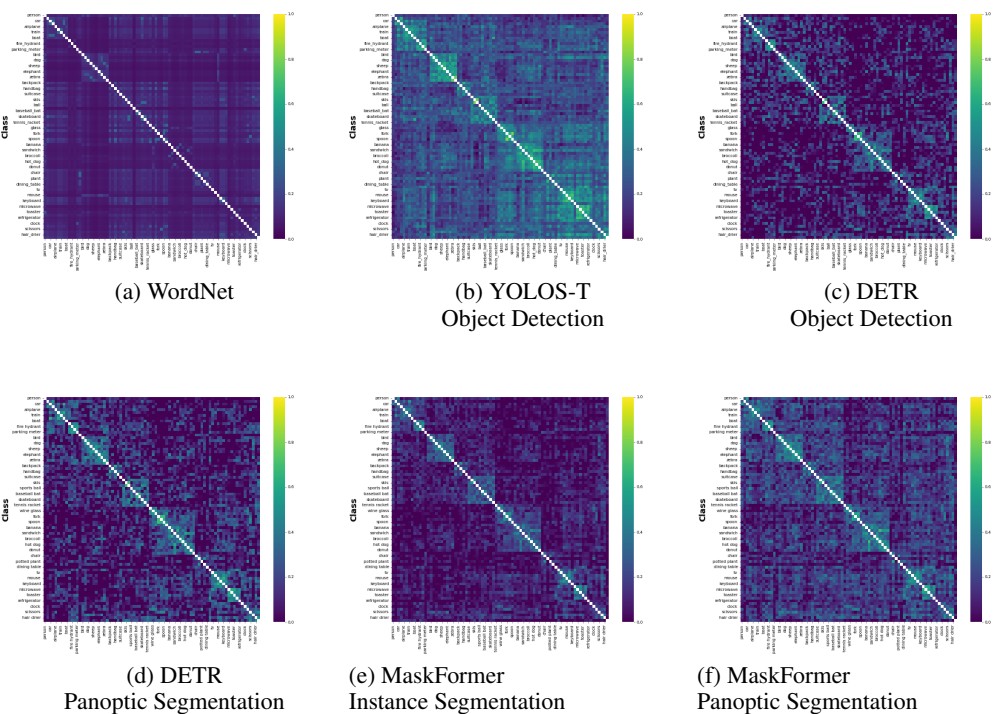

(a) WordNet     (b) YOLOS-T Object Detection     (c) DETR Object Detection

(d) DETR Panoptic Segmentation     (e) MaskFormer Instance Segmentation     (f) MaskFormer Panoptic Segmentation

Figure F.3: Network Class Similarity Matrices obtained for COCO classes and different object detection and segmentation networks. In all cases a clear hierarchy is visible. Relations, nevertheless, are richer that the WordNet-based ones. E.g. they rely a lot on context - see how 'fire hydrant' and 'road sign' are placed together in the first square along different vehicles.

# G CLOSER INSPECTION ON THE WSI CURVES OF CNNs

In Figure 1 in Section 4.1 of the main paper we observed a clearly visible difference in the mean Weight Similarity Index (mean WSI, describing the mean similarity of weight templates within a network) curves for ResNet18 and MobileNetV2 and other models (ViTs, hybrids), which was surprising. It prompted us to better explore this phenomenon. As both of these models represent convolutional neural networks, our hypothesis was that such a mean WSI curve is characteristic for CNNs. To better prove it, we chose two additional CNNs, namely DenseNet121 and EfficientNetB0. We trained them on mini-ImageNet. In Figure G.1, we present the results obtained for these models and different WSI variants. The results support our hypothesis - also these new models result in a similar Mean WSI curve, which further highlights the impact of neural network architecture on the similarity perception.

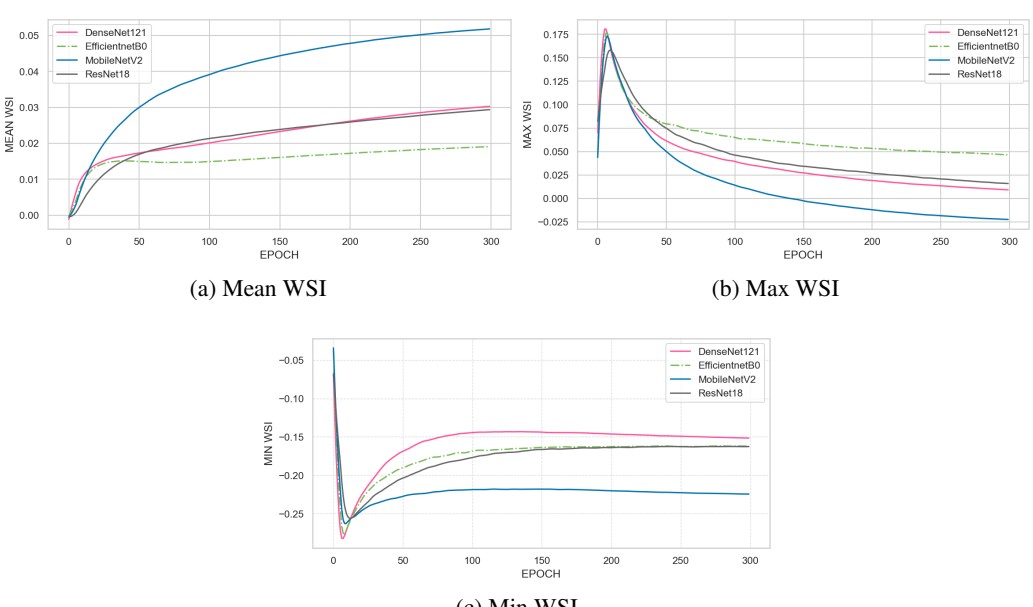

(a) Mean WSI

(b) Max WSI

(c) Min WSI

Figure G.1: Closer inspection on the WSI curves for different CNNs on Mini-ImageNet dataset.

# H CLOSER INSPECTION OF NETWORK'S MISTAKES DURING TRAINING

In Section 4.3 of the paper, we analyzed whether the confusion patterns of CNNs and ViTs match their similarity perception throughout the training. For this purpose, we used, among others, our proposed method, the IDM curves. The results showed us an interesting phenomenon: the errors-only IDM revealed that after reaching the peak, its values slightly drop. It indicates that the network is making errors between categories it perceives as less similar. We hypothesized that the reason can be that the network's accuracy is already high then, and more 'obvious' mistakes have been eliminated. The network is now tackling more challenging and less typical samples, or even potential noise from mislabeled data. We now take a closer look on the network's mistakes. First of all, we examine the difference in the similarity perception of correctly and incorrectly classified samples by an example network used in our analysis - MobileNetV2. To do this, we first extract the templates for all samples from the training and testing datasets. After averaging, we obtain 2 matrices storing the dataset-level templates. We use them as a reference. Then, we create the average template matrices, but separately for the correctly and incorrectly classified images. For the two splits of the dataset, we obtain 4 matrices of templates in total. We then compare the average templates for the correctly and incorrectly classified samples with the overall template via cosine similarity (for the whole dataset). Note that this comparison is done for particular classes, so we compare with each other the templates of the same class. We present the results in Table H.1. The results show that in the later epochs, the samples that are misclassified are more distant (less similar) in the feature space from those

Table H.1: Average cosine similarity between averaged templates obtained for the correctly and incorrectly classified samples after different training epochs.

| Data split | Epoch | | | | | | | | | |
|---|---|---|---|---|---|---|---|---|---|---|
| | 7 | 10 | 12 | 20 | 25 | 35 | 50 | 100 | 200 | 299 |
| Train-True | 0.978 | **0.981** | **0.984** | **0.991** | **0.994** | **0.996** | **0.998** | **1.** | **1.** | **1.** |
| Train-False | **0.984** | **0.981** | 0.980 | 0.972 | 0.964 | 0.951 | 0.950 | 0.930 | 0.893 | 0.859 |
| Test-True | 0.977 | 0.980 | **0.982** | **0.989** | **0.993** | **0.994** | **0.995** | **0.997** | **0.997** | **0.997** |
| Test-False | **0.983** | **0.983** | 0.981 | 0.973 | 0.966 | 0.957 | 0.955 | 0.949 | 0.952 | 0.954 |

classified correctly. This also suggests that the network makes more informed mistakes at this point, and that the drop in the IDM is due to less typical/difficult examples, or even mislabeling issues. In Fig. H.1, we also included the plot of the number of mistakes as a function of the epoch number. The plot shows that the number of mistakes decreases in a logarithmic fashion (which is expected, as the network learns its task). As the number of errors decreases, the remaining errors caused by the less typical/difficult etc. samples become more prominent within the set of incorrectly classified samples (also visible in the second plot of H.1, presenting the histograms of cosine similarity values between the incorrectly classified samples and their templates for 4 example epochs). That is why a slight decrease in errors-only IDM occurs, but overall (as IDM is still high) mistakes are still driven by the similarity perception of the network. These observations align with our earlier findings. To examine this even more comprehensively (image-level), we take MobileNetV2 and manually analyze the images, for which the network made mistakes to show in a practical and straight-forward way what samples cause errors.

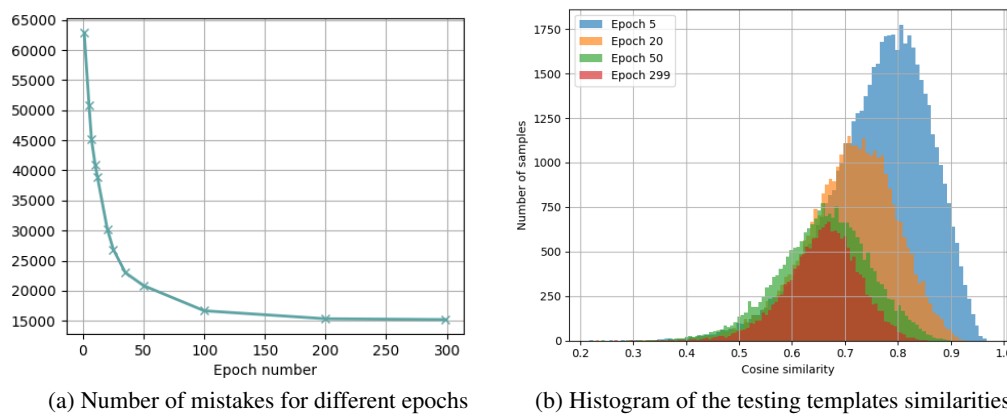

(a) Number of mistakes for different epochs      (b) Histogram of the testing templates similarities

Figure H.1: Analysis of the changing number of mistakes at different epochs.

In Figures H.2, H.2 and H.2, we present the samples images, for which the network made mistakes after the 5th, 50th and 299th epoch respectively. Figure H.2 shows that in the early epochs of training, the mistakes of the network are not very reasonable (e.g. a missile is named a dalmatian - mistakes between very unrelated objects), which can be expected, but at that point it is also visible that at least some similarities are learned by the network (e.g. it mistakes a golden retriever with another dog breed). Close the the IDM peak (see Figure H.3), the network mostly make mistakes between similar classes (which is shown by the high IDM values in Section 4.3). E.g., it makes mistakes between similar dog breeds, between very similar classes (e.g. parallel and horizontal bar). The errors are also cause by the mislabeling issues (e.g. wok/frying pan), therefore they are not truly the mistakes sometimes. In Fig. H.4, we can see that in later epochs (when the IDM errors only variant values drop), the network makes mistakes on the less typical images (e.g. a dog hidden in a plastic box), difficult images (e.g. blurry images, very small target objects), due to mislabeling issues (e.g. a frying pan named a wok) or due to a coocurrence of objects in the same picture. These examples support the hypothesis made earlier.

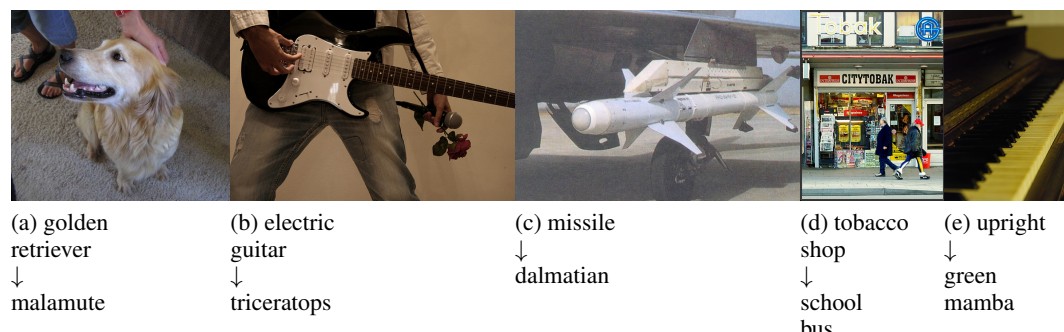

(a) golden
retriever
↓
malamute

(b) electric
guitar
↓
triceratops

(c) missile
↓
dalmatian

(d) tobacco
shop
↓
school
bus

(e) upright
↓
green
mamba

Figure H.2: Inspection of MobileNetV2's mistakes on the test set at epoch **5** and Mini-ImageNet. (a) a rather typical image, mistaken with a similar class (another breed), (b), (c), (d), (e) a very clear, typical image, yet mistaken with a very dissimilar class.

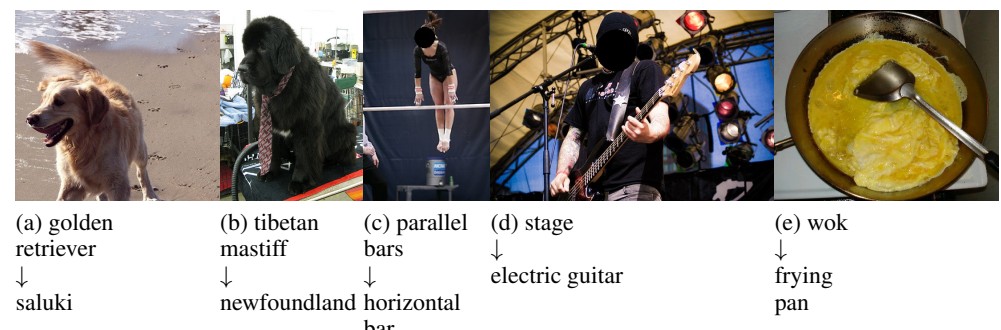

(a) golden
retriever
↓
saluki

(b) tibetan
mastiff
↓
newfoundland

(c) parallel
bars
↓
horizontal
bar

(d) stage
↓
electric guitar

(e) wok
↓
frying
pan

Figure H.3: Inspection of MobileNetV2's mistakes on the test set at epoch **50** and Mini-ImageNet. (a) A rather typical image of a dog mistaken with a similar class (another, similar breed of dog), (b) A rather typical image of a dog mistaken with a similar class (another, similar breed of dog), (c) A mistake due to the occurrence in the class space of 2 highly similar classes, (d) Both labels are true (cooccurence), (e) It is a normal pan, not a wok - mislabeling issues.

## I    INSPECTION OF THE CYCLICAL NATURE OF THE BUMPS ON THE NETWORK SEMANTIC SIMILARITY ALIGNMENT CURVE

In Section 4.2 of the main paper, we aimed to examine how does the network's similarity perception change during training for CNNs and ViTs and whether it is in line with semantic similarity. In Figure 2, we presented the SAI(NCSM, SCSM) curve (the alignment between the Network Class Similarity Matrices and Semantic Similarity Matrix obtained via WordNet). In this figure, visible bumps can be observed (while the alignment slightly decreases). We named this the similarity perception 'refinement'.

In this section, we provide some additional plots in Fig. I.1 showing the figures of the train loss and learning rate presented in the same plots together with this SAI variant (for an example network trained for our additional CNN-focused experiments - EfficientNetV2B0). These bumps occur in the phase that the network still dynamically learns (see the train loss plots in Fig. I.1). It is visible that the loss and SAI curves are similar but do not present the same thing. While they both steeply increase/decrease in the first stage, in the second stage the loss curve still decreases dynamically. The SAI curve is characterized by a stable trend (however with visible bumps) or slight decreases (and not increase this time). It shows that at this point, it is not the similarity perception learning of the classes, but rather their differentiation. In the loss curve, also some bumps are visible in this phase. These bumps occur in the period, in which the learning rate scheduler makes the learning rate constant and then utilizes learning rate decay. These abrupt changes can cause the loss and the

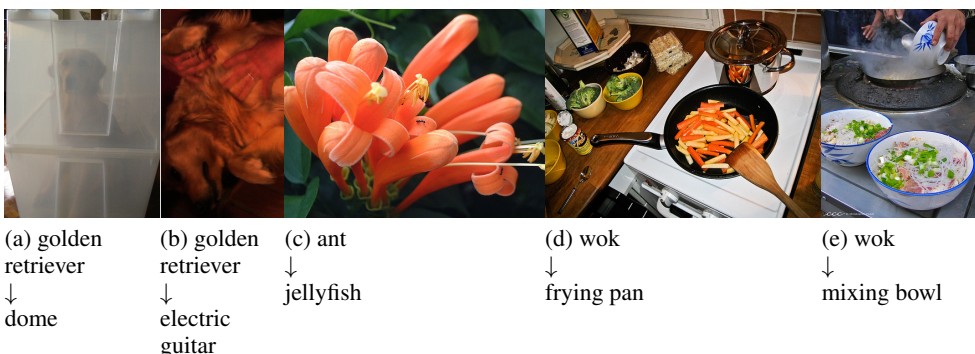

(a) golden retriever
↓
dome

(b) golden retriever
↓
electric guitar

(c) ant
↓
jellyfish

(d) wok
↓
frying pan

(e) wok
↓
mixing bowl

Figure H.4: Inspection of MobileNetV2's mistakes on the test set at epoch **299** and Mini-ImageNet. (a) represents a less typical picture of a dog (it is less visible due to it being closed in a box), (b) a more difficult to categorize picture of a dog due to it being blurry, also - the hands can be connected with a guitar, (c) difficult image, an object is very small and surrounded by different objects, also the flower is very colorful and overwhelms the picture, (d) it is a normal pan, not a wok - mislabeling issues, (e) both labels are true (co-occurence), bowls are more visible.

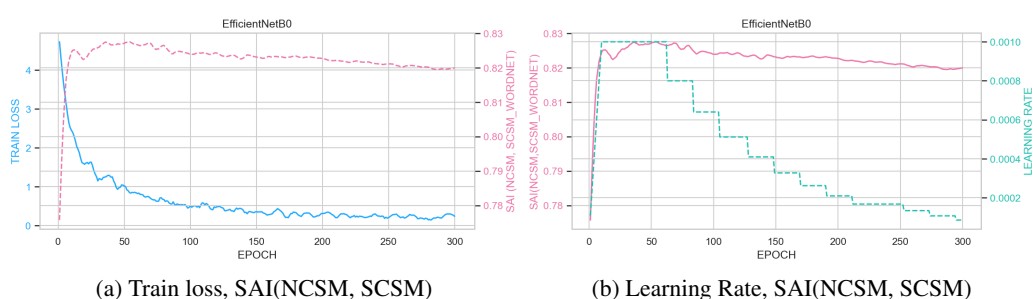

(a) Train loss, SAI(NCSM, SCSM)

(b) Learning Rate, SAI(NCSM, SCSM)

Figure I.1: Inspection the cyclical nature of the bumps on the network semantic similarity alignment curve for EfficientNetV2B0 trained on Mini-ImageNet.

SAI curve to temporarily spike as the optimizer adjusts. It is visible that the SAI curve in Figure I.1 stabilizes when learning slows down and the changes in the learning rate become less abrupt. Also, techniques such as dropout add noise during training, so this noise can also cause some temporary instability in the loss/SAI curves.

## J   SIMILARITY INSPECTION BEYOND THE OBJECT CATEGORIZATION

In the main paper, we focused on the similarity inspection of different object recognition networks. In Section F, we examined additional networks trained on object detection and segmentation showing that the similarity perception also emerges in such networks in a similar way as it occurs in the object recognition models. Nevertheless, all of these models have a lot of commonalities. E.g. even though the object detection and segmentation networks differ in task from the object recognition, they often use the backbones pretrained on object recognition (usually ImageNet). Also, in all of these networks some kind of an object-level classifier occurs (which makes it possible to use our weight-based similarity computation method). They are all trained with the supervised approach and the optimization process of these networks is also similar.

Both from the perspective of computer vision and deep learning, there are many other training objectives for visual learning and understanding, e.g. self-supervised and contrastive learning approaches (Oquab et al., 2023; Chen et al., 2020; Margalit et al., 2024). On the other hand, also models trained on joint text-image objectives are available (Radford et al., 2021). Such networks learn strong semantically meaningful representations and can produce strong candidate models of the visual processing.

Table J.1: Mini-ImageNet: Similarity Alignment Index for different versions of direct similarity estimation measures for networks (based on the MobileNetV2 and different epochs) - rounded to 2 decimal places (values 1.0 stand for values very close to 1.0, but not exact 1). While NCSM can be obtained for the networks with classifiers, TCSM can be used as its dataset-dependent approximation. We compute two possible variants of TNCSMs for the train-test split (named in the table as -tr and -tst).

| Data split | Epoch | | | | | | | | | |
|---|---|---|---|---|---|---|---|---|---|---|
| | 7 | 10 | 12 | 20 | 25 | 35 | 50 | 100 | 200 | 299 |
| SAI(NCSM, TNCSM-tr) | 0.95 | 0.95 | 0.95 | 0.95 | 0.95 | 0.95 | 0.95 | 0.94 | 0.94 | 0.93 |
| SAI(NCSM, TNCSM-tst) | 0.95 | 0.94 | 0.95 | 0.95 | 0.95 | 0.94 | 0.94 | 0.93 | 0.93 | 0.92 |
| SAI(TNCSM-tst, TNCSM-tr) | 1.00 | 1.00 | 1.00 | 1.00 | 1.00 | 1.00 | 1.00 | 1.00 | 1.00 | 1.00 |

In this section, we present some additional results for two models trained with self-supervised objectives (not for object recognition): DINOv2 (Oquab et al., 2023) and CLIP (Radford et al., 2021). DINOv2 is a model trained with a self-supervised learning framework designed specifically to produce high-quality image representations. CLIP, on the other hand, learns a common representation space for images and text simultaneously, which enables cross-modal tasks and the understanding not only of visual, but also textual semantic relations. As those models, during pre-training do not include a traditional classifier, the classifier's weights cannot be used to produce the Class Similarity Matrix. Nevertheless, even with those models we can enable a similarity-based analysis. To do this, we need the annotated dataset (E.g. Mini-ImageNet). In the evaluation step of the training, a dataset is used to extract templates from the network. These templates are aggregated for each class as it is done for object recognition networks in e.g. work (Huang et al., 2021). This step requires significantly more computational resources, however it can be treated as an alternative for our approach in the cases, in which we cannot use the weights of the classifier, making our method suitable also for self-supervised approaches. We can name this NCSM (Network Class Similarity Matrix) variant Templates-based Network Class Similarity Matrix (TNCSM).

TNCSMs can be obtained also for traditional classifier, however, as mentioned before, they require much more computational resources and are dataset-dependent, therefore are an approximation of the network's similarity perception. In Table J.1, we present the values of Similarity Alignment Index between the example network's (MobileNetV2's) TNCSM generated for the training and testing set of Mini-ImageNet at different epochs of training (we use the network's feature extractor to to this). We compare the TNCSM-train and TNCSM-test matrices with each other and with the NCSM (based on weights). The results show that the alignment between TNCSM-train and TNCSM-test is very high (almost 1.0) for all epochs. It is also very high, but slightly lower between template-based and network-based matrices (as TNCSM is a dataset-dependent approximation of NCSM). Nevertheless, this alignment is still high, therefore if not possible TNCSMs can be used as a good enough (however more costly) alternative to weights-based CSM. Moreover, it is visible that the values of SAI(NCSM, TNCSM-test) and SAI(NCSM, TNCSM-train) are closer to each other in the earlier epochs of training than in the later epochs revealing the slight overfitting of the network. It suggests the possibility of using the training and testing variants of similarity-based metrics to reveal phenomena such as overfitting in networks.

In Figure J.1, we present the Templates-based Network Class Similarity Matrices obtained for DINOV2 and CLIP for the Mini-ImageNet testing set (the models with weights were taken from https://huggingface.co/docs/transformers/model_doc/clip and https://huggingface.co/docs/transformers/model_doc/dinov2. It is visible that the overall structure of the matrices is the same as for all the considered networks analyzed in the main body of the paper (trained for object recognition). The main difference between these two matrices is that the one for CLIP includes more similarities between less related categories (out of the main similarity groups: artifacts and animals), showing the impact of textual semantics on the similarities (e.g. in sentences, dogs can occur in the sentence frequently close to some home objects). We also computed the numerical values of the semantic similarity alignment between the TNCSM obtained for DINOv2 and CLIP and Wordnet CSM (SAI(TNCSM, SCSM)). It is **0.85** for DINOv2 and **0.82** for CLIP. The value obtained for DINOv2 is slightly higher than the maximum values obtained for networks and the value obtained for CLIP is similar to the majority of traditional object recognition

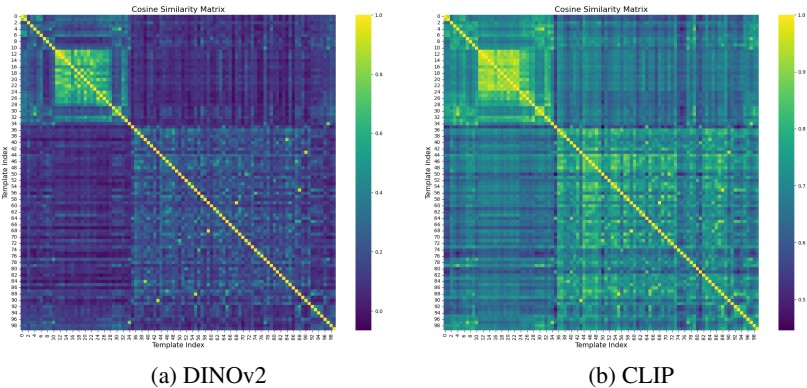

(a) DINOv2        (b) CLIP

Figure J.1: Templates-based Network Class Similarity Matrices obtained for DINOV2 and CLIP - networks trained with representations-focused objectives.

networks used in our experiments (the highest value was app. 0.84, all higher than 0.8). This surprisingly shows that although these networks use similarity-focused techniques, the final results do not diverge very much from those obtained via traditional training schemes. In our future work, we will more deeply analyze this fascinating finding.

# K SIMILARITY AND ADVERSARIAL ROBUSTNESS

As presented in this work, similarity has a crucial role for the learning of deep vision networks. It possibly has many more implications than the ones presented in this paper. An interesting aspect is the relationship between the similarity perception and adversarial robustness. To provide some insights on this aspect, we performed some additional experiments to examine the impact of the alignment of Network similarity perception with semantic perception on adversarial robustness of networks. To do this, we take two different networks from our experiments (1 ViT: SwinV2T and 1 CNN: MobileNetV2) and use them to generate perturbations with one of the common adversarial attacks - Projected Gradient Descent (PGD) (Madry et al., 2017) and its cleverhans (Papernot et al., 2018) implementation (with parameters eps=0.05, eps_iter=0.001, nb_iter=2). We generate the attack for the model checkpoints at different epochs and plot the Fooling Rate results along with the Semantic Similarity Alignment (SAI(NCSM, SCSM)) - see Fig. K.1. We also show some confusion-based CSMs for different epochs obtained under the adversarial setup.

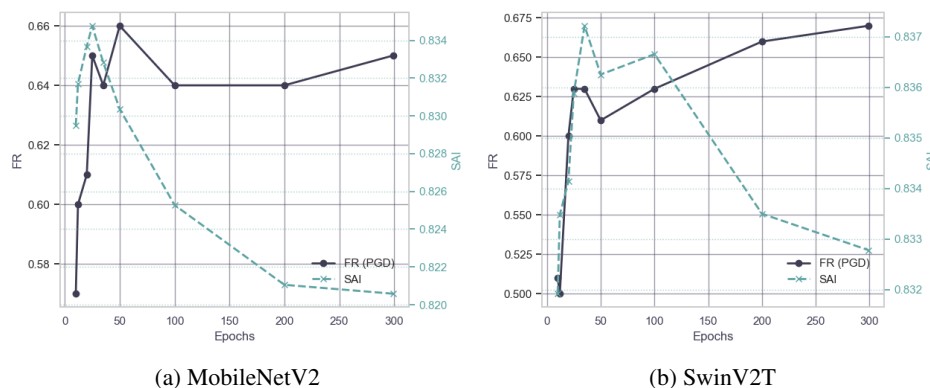

(a) MobileNetV2        (b) SwinV2T

Figure K.1: Mini-ImageNet: Relation between the semantic similarity alignment of the network perception of similarity and the adversarial robustness (expressed via the Fooling Rate for the PGD result) at different epochs.

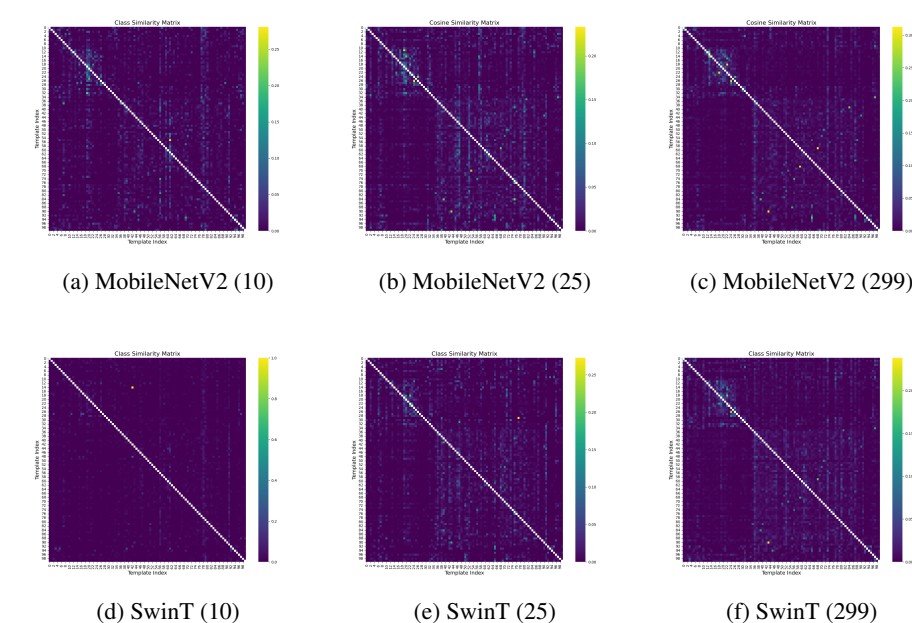

| (a) MobileNetV2 (10) | (b) MobileNetV2 (25) | (c) MobileNetV2 (299) |

| (d) SwinT (10) | (e) SwinT (25) | (f) SwinT (299) |

Figure K.2: Mini-ImageNet: Confusion-based Class Similarity Matrices under the adversarial setting for the Projected Gradient Descent attack.

In the early epochs, both for the examined ViT (SwinV2Tiny) and CNN (MobileNetV2), robustness decreases with the increase in accuracy (e.g. at epoch 10 the networks are good at generalization, but they are not very accurate, so they can be considered too general and in some ways robust to adversarial attacks) - see Fig. K.1. Nevertheless, when one looks at the confusion-based Class Similarity Matrices for the adversarial case for these networks presented in Fig. K.2, it is visible that the mistakes under the adversarial setup also become more and more similarity-dependent (networks more often make mistakes between similar classes - which can be observed as a clear hierarchical structure in their CSMs). After this initial grow, a slight drop occurs. In later epochs, the robustness drops (fooling rate increases) with the drop in the decreasing value of the alignment, suggesting that our metric (Semantic Alignment Index for Network CSM and Wordnet CSM) reflects the overfitting of the network (visible as divergence of the perception from the semantic similarity). Although this aspect requires a more in-depth analysis, the presented results show the potential of using the proposed framework also in the area of adversarial robustness improvement (e.g. via using our metric as a component of the loss function or for a regularization). The results also show that the confusion-based CSMs under the adversarial setup can be used as an alternative for our Confusion-based Class Similarity Matrices obtained on clear data and an extension of our framework. Nevertheless, it is worth remembering that generating adversarial perturbations introduces significant costs, particularly with iterative attacks (therefore testing can be done e.g. not each epoch, but e.g. 1 in every 5 epochs or so). Moreover, the attack choice is highly variable introducing additional complexity.

# L    DICTIONARY OF TERMS USED

As our framework includes many metrics, which results in the presence of numerous abbreviations in the main paper, we decided to provide a short description and additional figures to better explain the main terms, metrics and their variants used in the paper.

**Class Similarity Matrices (CSMs)**

- **NCSM** – Network Class Similarity Matrix (based on network weights, image-free). It is created based on weights of the final classifier of the network (data-independent, direct similarity measurements).

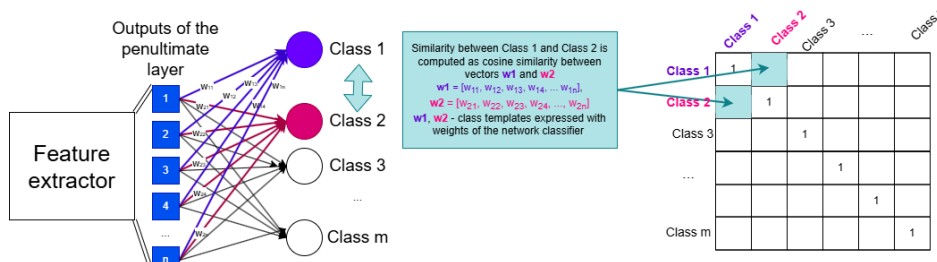

Figure L.1: NCSM matrix calculation scheme.

- **CCSM** – Confusion matrix-based Class Similarity Matrix (possible train and test variants). It is created as a transformation of the confusion matrix obtained for a given network and a given dataset (data-dependent, indirect similarity measurements).

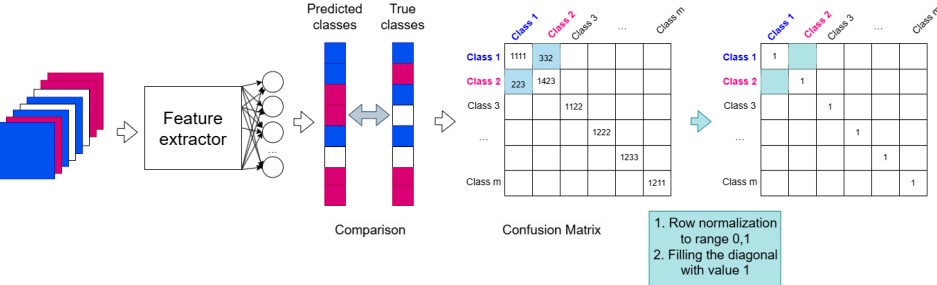

Figure L.2: CCSM matrix calculation scheme.

- **TNCSM** – Image templates-based Network Class Similarity Matrix (possible train and test variants). It is created based on the features extracted by a given neural network. Features obtained for a particular class are averaged to obtain a general representation of a given class dependent on this dataset (data-dependent, direct similarity measurements).

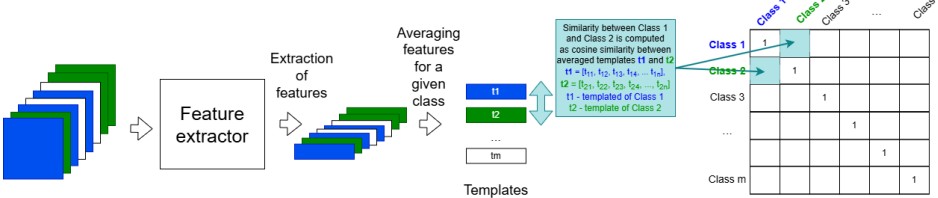

Figure L.3: TNCSM matrix calculation scheme.

- **SCSM/WCSM** – Semantic/WordNet Class Similarity Matrix (As a reference, Semantic Similarity source is used). It is created based on the WordNet structure and WordNet's

similarity measure – path (see Appendix B for details of path calculation). It is computed between two concepts in the WordNet tree, thus reflecting the distance between them.

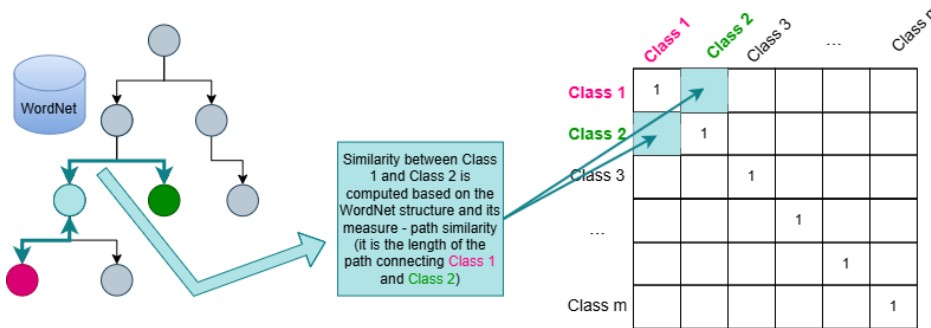

Figure L.4: TCSM matrix calculation scheme.

Note that **CCSM** and **TNCSM**, as they are the only data-dependent CSMs, are computed per dataset, therefore if train-test split is used, 2 different matrices per each CSM variant can be obtained (i.e. **CCSM-test**, **CCSM-train** and **TNCSM-train**, **TNCSM-test**).

**Possible (Semantic Alignment Index) SAI variants**

SAI is computed as a comparison (numerical) between two similarity matrices (see Fig L.5 below). As their format is the same regardless of the data source used to create them, all CSM variants can be technically used for comparison.

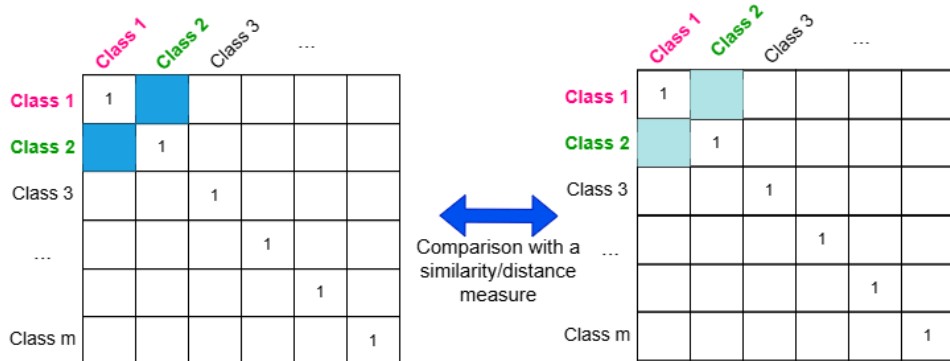

Figure L.5: SAI calculation scheme.

**SAI variants used in paper:**

- **SAI(NCSM, SCSM)** – Similarity Alignment Index (Network Class Similarity Matrix, Semantic Class Similarity Matrix) – it determines how well the Network-perceived similarity (based on weights) aligns with semantic similarity.
- **SAI(CCSM, SCSM)** – Similarity Alignment Index (Confusion-based Class Similarity Matrix, Semantic Class Similarity Matrix) – it determines how well the Network-perceived similarity (measured indirectly based on confusion on the testing set) aligns with semantic similarity.
- **SAI(CCSM, NCSM)** – Similarity Alignment Index (Confusion-based Class Similarity Matrix, Network Class Similarity Matrix) – it determines how well the Network-perceived similarity (based on Confusion on test) aligns with the Network's direct similarity perception (based on weights).

- **SAI(TNCSM-train, TNCSM-test)** – Similarity Alignment Index (Templates-based Network Class Similarity Matrix for train dataset, Templates-based Network Class Similarity Matrix for test dataset) – it determines how similar are the Templates-based Network CSMs for the training dataset and testing dataset (measured directly based on templates extracted with a feature extractor).
- **SAI(TNCSM-train, NCSM)** – Similarity Alignment Index (Templates-based Network Class Similarity Matrix for train dataset, Network Class Similarity Matrix) – it determines how well the Network-perceived similarity (based on templates generated with the train dataset) and the Network-perceived similarity (based on weights) align. To see whether we can use them interchangeably
- **SAI(TNCSM-test, NCSM)** – Similarity Alignment Index (Templates-based Network Class Similarity Matrix for test dataset, Network Class Similarity Matrix) – it determines how well the Network-perceived similarity (based on templates generated with the test dataset) and the Network-perceived similarity (based on weights) align.

**Inverse Dissimilarity Metric (IDM)**   It is a metric that measures how far in the space defined by an NCSM's similarity (in terms of the normalized number of classes in the matrix sorted with increasing similarity to a given class - per row) for this network are the predictions of the network from their ground truth labels. It is based on the Dissimilarity Metric (DM) introduced in Filus & Domanska (2023). The original DM can be computed in the following way. We take the ground truth label $i$ and the post-attack prediction $j$ for each image in the dataset. We check the index of $j$ in row $i$ of the Sorted Class Similarity Matrix (SoCSM). After gathering predictions on a dataset (with perturbations), DM quantifies the harmfulness of the attack. For each image, let $i$ represent the ground truth label, and $j$ the post-attack prediction (in our reformulation - it is the prediction on the clear dataset). We determine the rank $r_{ij}$ of $j$ in row $i$ from the Sorted Class Similarity Matrix. A higher $r_{ij}$ indicates greater damage, as the post-attack prediction $j$ is more dissimilar to the ground truth label $i$ (in our variant, these higher value of this rank, means less reasonable predictions, therefore we need an additional computational step at the very end - $1 - DM$ to obtain higher values for better semantic accuracy). The computational steps of DM are as follows:

1. Compute the rank $r_{ij}$ for all images in the dataset.
2. Calculate the mean rank value:

$$\text{Mean Rank} = \frac{1}{N} \sum_{k=1}^{N} r_{ij}^{(k)}$$

   where $N$ is the number of images in the dataset, and $r_{ij}^{(k)}$ is the rank for the $k$-th image.
3. Normalize the mean rank by dividing by the total number of classes minus one:

$$\text{DM} = \frac{\text{Mean Rank}}{C - 1}$$

   where $C$ is the total number of classes. This normalization ensures DM being in range $\langle 0, 1 \rangle$.

The original's DM values can be interpreted as (in terms of the harmfulness of the adversarial attack):

$$\begin{cases} 1 & \textit{max harmfulness} \Rightarrow \textit{accuracy} = 0\% \\ 0 \leq DM \leq 1 & \textit{the higher the more harmful attack} \\ 0 & \textit{min harmfulness} \Longleftrightarrow \textit{accuracy} = 100\% \end{cases}$$

As we compute the *inverse* of the DM, we transform its values as $IDM = 1 - DM$. Therefore, the IDM's values can be interpreted as (in terms of the accuracy extension):

$$\begin{cases} 1 & \textit{max semantic accuracy} \Longleftrightarrow \textit{accuracy} = 100\% \\ 0 \leq DM \leq 1 & \textit{the higher the more resonable the mistakes} \\ 0 & \textit{min semantic accuracy} \Rightarrow \textit{accuracy} = 0\% \end{cases}$$

We also compute the errors-only variant of IDM, in which we only consider samples, for which a network made a mistake. We also propose to include a **new variant** of IDM - WordNet-based IDM, which uses semantic similarity instead of Network-based similarity to compute ranks.

