# OpenReview forum: "Uncovering Self-Emergent Similarity in Deep Vision Networks: A Systematic Framework"
_ICLR.cc/2025/Conference — Submitted to ICLR 2025_

### Official Review · Reviewer_295U · 2024-10-19

**Soundness:** 2
**Presentation:** 2
**Contribution:** 2
**Rating:** 5
**Confidence:** 4

**Summary:**

This work performs a quantitative analysis in an attempt to understand when semantic similarity emerges in different vision networks. The focus is mostly on models that are trained to perform classification and similarity is defined based on different ways of assessing class confusion. The authors outline different ways of generating class similarity/confusion matrices and different metrics for comparing models based on these matrices. Results are present across different CNN and ViT models trained on datasets such as mini ImageNet.

**Strengths:**

[S1] The work is exploring a very interesting set of questions, and multiple experiments are conducted to investigate them.

[S2] Some of the observations and insights wrt to when similarity emerges in different models are interesting.

**Weaknesses:**

[W1] Evaluation
* The evaluation hinges on the assumption that category level labels are a good proxy for visual similarity. As noted on line 379, there can be many other semantic relationships present that are not captured by this. In addition, there can also be fine-grained sub-category level differences, e.g., the female of one bird species might look more like the female of another, but the males could be very different. This restriction to category labels and the hierarchy of WordNet is a big limitation as it may not be a good proxy for visual similarity, or at least visual similarity as perceived by humans.
* There are also other factors that will impact how similarity is encoded in a network that are not explored, e.g., network capacity, how the weights are initialized, or nuances of how the networks are trained.

[W2] Missing explanations
* While the experiments are extensive, there are cases where the analysis is lacking. For example, ResNet18 and MobileNetV2 in Fig 1 (a) are outliers but this is not explained or discussed sufficiently.

[W3] Missing Related Literature
* There is a body of work that seeks to develop methods for comparing the representations learned by networks, e.g., CKA by Kornblith et al., ICML 2019.
* Line 91 states that the authors are only aware of two papers that “examine whether the similarity perception of deep object recognition networks aligns with human judgement”. There is in fact a lot more uncited work on this topic, e.g., see the dataset collected by Roads and Love CVPR 2021 that enriches ImageNet with human similarity labels. There is also work on trying to predict human similarity judgements from pre-trained vision networks, e.g., Attarian et al. NeurIPS Workshops 2021, and work using generated images for similarity assessment DreamSim Fu et al. NeurIPS 2023.
* Finally, there is also work in computer vision that addresses the fact that there might be more than one “notion” of similarity used to compare image instances, e.g., CSN from Veit et al. CVPR 2017.

[W4] Paper Readability and Structure
* There are a lot of acronyms that require the reader to revisit earlier sections of the paper to understand what is being presented, e.g., Fig 2 caption notes “SAI(NCSM, CCSM)”. The captions of the figures could be improved to make parsing these details easier.
* Research question 1 (L273), as worded, seems to be two different questions combined? It also overlaps with research question 3. The text of these questions could possibly be refined to make them clearer.

Minor additional comments that do not require a response during the rebuttal:
* The paper would benefit from a clear definition of “similarity” in the introduction
* Use \citep{} instead of \cite
* L618 this paper is incorrectly cited as NeurIPS 2024, and not 2023
* L1299 and elsewhere use ``bad’’ instead of ”bad” in latex

**Questions:**

None

---

> ### Author Response · Authors · 2024-11-24
>
> Thank you for providing us with numerous helpful comments and suggestions, and especially for providing us with concrete works we can add to our related works section to improve it. We are really grateful that the reviewer spent their time to provide us with these bib items! We are also glad that the reviewer perceives the merits of our work: that the set of questions we are exploring is very interesting and multiple experiments performed. We appreciate highlighting that our observations and insights are interesting.
>
> Below, we answer the specific weaknesses/questions. We did our best to incorporate all the changes requested by the reviewer to improve our paper. We marked all the changes in pink in the revised version. If the changes and responses meet the reviewer’s expectations, we would kindly ask the reviewer to reconsider the given score. Thank you very much for your time and effort.
>
> >W1 The evaluation hinges on the assumption that category level labels are a good proxy for visual similarity.
>
> While we theorize that semantic similarity of labels can serve as a good initial insight to how visual similarities are developed during network training, we also provide other metrics that align with our assumptions, that do not rely on linguistic taxonomy. With metric SAI(NCSM, CCSM) we provide network’s similarity-mistakes alignment that doesn’t make use of category labels. Also, in **Appendix H** we provide additional experiments on how similarities develop during the network training on the basis of the features extracted by the network. In both experiments, the development of internal similarities can be observed.
>
> As for the use of category labels themselves - we understand that category labels could be ambiguous or too broad for the category itself, however as we prove in the paper, they can still serve as a valuable insight to how the model is trained. Furthermore, we intended our method to perform well without any additional human input, and that (apart from our additional experiments with semi-supervised networks in **Appendix J**) directed us to use category labels. We also intend to discard their use in our further works, for e.g. industrial-grade coarse classifiers' development, but at this point they serve as a cornerstone for further method development.
>
> >W2 There are also other factors that will impact how similarity is encoded in a network that are not explored, e.g., network capacity, how the weights are initialized, or nuances of how the networks are trained.
>
> Thank you for your valuable insight - that is entirely correct, as networks tend to behave differently when different sets of hyperparameters/initialization etc. are provided. However, in order to perform our analysis in a consistent fashion, we decided to focus on architecture types to differentiate them, and not hyperparameter sets. In order to remain consistent, we trained all the networks with a similar recipe (the only difference being the inclusion of CutMix for transformer-type networks). We agree that the verification of the influence of model initialization, hyperparameters, scheduling etc. would give an extended context to how similarities are developed during model training (and we provide a shortened version of such analysis in **Appendix E**), but due to page limitation we decided to focus on various architectures, rather than on specific training recipes for single architecture type.
>
> TBC in the next comment...

---

> ### Author Response · Authors · 2024-11-24
>
> >W3 While the experiments are extensive, there are cases where the analysis is lacking. For example, ResNet18 and MobileNetV2 in Fig 1 (a) are outliers but this is not explained or discussed sufficiently.
>
> We understand that CNNs differ in similarity metric score from transformer-type networks, and we acknowledged it in our paper. To delve more deeply into the differences between the networks, due to the limited number of pages in the main body of the text, we have added an extended analysis of this phenomenon in the **Appendix G**. In order to further prove our finding, we included additional experiments with two other CNN networks - DenseNet and EfficientNe that align with our findings.
>
> >W4 Missing literature.
>
> We would like to thank the reviewer for providing us with many concrete references that we could use to broaden our related works section. We initially aimed to describe the works most similar to ours, however we agree that such an extension allows for a better placement of our work in the related literature and draws the domain better. We used the provided references (and added some more) to show that different notions of similarity can be used in computer vision. We also used them to show that different reference similarities can be used as a similarity reference (for alignment) and highlighted better why we decided to use semantic similarity and not human judgments based on large polls (generalization to different classes and datasets, consistency of similarity estimation and practical computational modeling of semantic similarity). However, we also highlighted that for some limited datasets, human judgment-based labels exist in the literature.
>
> >W5 There are a lot of acronyms that require the reader to revisit earlier sections of the paper to understand what is being presented.
>
> To improve this aspect, where possible (due to a strict page limit), we included some additional short interpretations of the acronyms, e.g. we modified the description of Fig. 2 to highlight the purpose and meaning of SAI(NCSM, SCSM): Mini-ImageNet: SAI(NCSM, SCSM) Curves (network-semantic similarity alignment).
>
> >W6 Research questions could be refined.
>
> We refined the research questions to make that better to read and to highlight the fact that we focus our research on semantic similarity used as a reference.

---

> > ### Comment · Area_Chair_8Mnd · 2024-11-25
> >
> > Dear Reviewer,
> >
> > The authors have provided their responses. Could you please review them and share your feedback?
> >
> > Thank you!

---

> > > ### Comment · Reviewer_295U · 2024-11-25
> > >
> > > We thank the authors for their responses and for the changes in the paper. I have no further comments.

---

> > > > ### Author Response · Authors · 2024-11-26
> > > >
> > > > We would like to thank the reviewer again for their thoughtful feedback and for acknowledging the changes we made to the manuscript during the rebuttal. In response to the initial comments, we conducted additional experiments, extended the analysis, and revised the text to strengthen the paper. If the changes we made satisfactorily address the reviewer's concerns, we would kindly ask the reviewer to reconsider their evaluation. If there are any aspects in the paper we could improve, or if the reviewer has any other suggestions/questions, we would appreciate further guidance (taking advantage of the rebuttal). We will be more than happy to make additional adjustments if needed.

---

> > > > ### Author Response · Authors · 2024-11-27
> > > > **additional refinements**
> > > >
> > > > We would like to add that during our another round of editing and fixing the text issues, we further eliminated the issues mentioned by the reviewer as [W4]. We added an additional appendix with the dictionary of abbreviations (to make the usage of acronyms more comfortable) and images that better explain the formulation of our methods. We also further rewrote the research questions. As suggested, we separated the RQ1 to 2 separate questions (and the corresponding results to 2 separate sections). Also, we highlighted better that RQ3 (old RQ2) refers to the examination of the relation between the mistakes and the *network's perception of similarity*, while RQ3 (old RQ2) refers to the relation between the mistakes and *semantic similarity*. Thank you for these suggestions! We hope that these changes improve the presentation and quality of our work and satisfy the standards of the reviewer.

---

> ### Comment · Reviewer_295U · 2024-12-01
>
> Greetings authors,
>
> Please find responses to your follow up comments below.
>
> [W3] Related work.
> The authors state that L052 "no quantitative and systematic similarity-based model evaluation framework for during training analysis exists." There are uncited works such as Marcos et al., "Insights on representational similarity in neural networks with canonical correlation", NeurIPS 2018 where the authors use Canonical Correlation Analysis (CCA) is explore how representations, not necessarily similarity directly, changes over time during training which should be referenced.
>
> [W2] Missing explanations.
> The new results in Fig G.1 are helpful - thanks! However, it would have been nice to have some understanding as to why MobileNetV2 is such an outlier.
>
> [W4] Paper Readability and Structure.
> The dictionary in Appendix L is a great addition - thanks! Also the restructuring of the research questions in Section 4 makes things clearer. However, there are some places where the current text is unclear or could be reworded to make it easier to understand, e.g.
> * L052 "no quantitative and systematic similarity-based model evaluation framework for during training analysis exists" -> fix for grammar
> * L078 "feature similarity (comparing data points representations learned by networks, e.g. (Kornblith et al., 2019))"  -> fix for grammar
> * L153 "Semantic similarity can be measured e.g. via WordNet" -> it would be useful here to briefly mention the limitations of such an approach as noted in my original review (i.e. class label not always being a good proxy for semantic similarity and the issues with the WordNet hierarchy).
> * Figs 1, 2, 3, etc.. In general, the captions are short and the paper would be easier to read if they were more detailed and self contained, e.g. describe the main observations and make the captions clearly indicate the metrics being evaluated (using the appropriate abbreviations for the metric names).
> * Intro/Appendix. It would be helpful to clarify that main focus of the analysis is on image classification tasks.
>
> In light of the responses from the authors I've updated my recommendation from a 3 to a 5.

---

> > ### Author Response · Authors · 2024-12-02
> >
> > Dear Reviewer! Thank you very much for these additional remarks. We are grateful for all the support and help to make our paper better! Below, we answer all the specific comments the reviewer provided us with. As for now we cannot upload a new version of the paper (November 27th was the last day that authors could upload a revised PDF), we comment on the changes that we will include in the final version of the paper - we did our best to include all the suggested changes. We hope that all the additional changes will satisfy the reviewer and make our paper suitable for the conference! If yes, we kindly ask the reviewer to reconsider their initial score. Once again, thank you for your time, help and support (and all very specific suggestions that we could easily use to improve our paper)!
> >
> > > [W3] Related work. The authors state that L052 "no quantitative and systematic similarity-based model evaluation framework for during training analysis exists." There are uncited works such as Marcos et al., "Insights on representational similarity in neural networks with canonical correlation", NeurIPS 2018 where the authors use Canonical Correlation Analysis (CCA) is explore how representations, not necessarily similarity directly, changes over time during training which should be referenced.
> >
> > Thank you for this additional reference! We will add it to our Related Work section. As the reviewer suggested, we also searched for additional works with representations-focused numerical measures for during training analysis. Please note, that these metrics (as the reviewer mentioned themselves regarding Marcos et al.) serve different purposes than our framework, however we agree that it will be useful to include such an extended analysis of different perspectives from which the training can be inspected besides the standard accuracy/loss metrics. We believe that this will additionally strengthen our paper as it shows that the works aiming at in-depth training inspection are valued in the literature. Below, we provide the additional reference we found and aim to add to the final paper besides the provided reference:
> >
> > Merlin, Gabriele, et al. "What happens during finetuning of vision transformers: An invariance based investigation." Conference on Lifelong Learning Agents. PMLR, 2023. - the authors present the metrics (for fine-tuning inspection) that investigate the degree to which invariances learned by a pretrained model are retained/forgotten during additional training (fine-tuning).
> >
> > > [W2] Missing explanations. The new results in Fig G.1 are helpful - thanks! However, it would have been nice to have some understanding as to why MobileNetV2 is such an outlier.
> >
> > We are glad that the reviewer likes the addition of this new plot! Thank you for your insightful comment regarding the slightly higher similarity of class representations in MobileNetV2 compared to other CNNs! We appreciate this observation and agree that commenting on the underlying reasons would add value to the analysis. We suspect that it can be due to the fact that MobileNetV2 is the smallest model in our analysis in terms of parameters (with only 3.5M parameters, other CNNs: DenseNet121 - 8M, ResNet18 - 11.7M, EfficientNetV2B0 - 5.3M). This reduced capacity may limit the model’s ability to learn highly differentiated class representations, potentially resulting in the observed slightly higher mean similarity. However, it is important to highlight that the range of mean similarity across all the tested models remains very small (0.02 to 0.05 cosine similarity), and while MobileNetV2 shows a visibly higher value (amplified by the scale of the plot), the difference may not be practically significant in the context of the broader analysis. We believe that the most important takeaway is that CNNs and ViTs (as in many other aspects) differ significantly also in this respect. We will add this discussion to the paper. We hope that this answer satisfies the reviewer and adds more depth to our analysis.

---

> ### Author Response · Authors · 2024-12-02
>
> [W4] Paper Readability and Structure. The dictionary in Appendix L is a great addition - thanks! Also the restructuring of the research questions in Section 4 makes things clearer.
>
> Thank you very much for acknowledging the worth of this additional appendix and the changes in the research questions! We believe that these changes improved the clarity of the paper, and we are really grateful for these helpful suggestions! We are also grateful for the additional comments of the reviewer (we will answer to the specific comments below)! As mentioned at the beginning, we cannot upload a new version of the paper now, but we promise to include all the described changes in the final version of the paper!
>
> > L052 "no quantitative and systematic similarity-based model evaluation framework for during training analysis exists" -> fix for grammar,  L078 "feature similarity (comparing data points representations learned by networks, e.g. (Kornblith et al., 2019))" -> fix for grammar
>
> Thank you for noticing these errors! We fixed them in the final version to improve the readability and flow of these sentences.
>
> > L153 "Semantic similarity can be measured e.g. via WordNet" -> it would be useful here to briefly mention the limitations of such an approach as noted in my original review (i.e. class label not always being a good proxy for semantic similarity and the issues with the WordNet hierarchy).
>
> Thanks for this suggestion! We appreciate your detailed feedback regarding the potential limitations of using category-level labels and the WordNet hierarchy as proxies for visual similarity. We will include this discussion in the limitations (in the Conclusions) to make it more visible for transparency (however, if the reviewer thinks it is better to add it close to line 153, we can add it there as well!). We will add:
>
> *Another limitation can be the restriction to category labels and the WordNet hierarchy, because they may not always be a good proxy for visual similarity (e.g. there can be many other semantic relationships present that are not captured by these). While this approach may not perfectly represent similarity in all its forms, it provides a practical and scalable computational framework for analysis, overweighting its disadvantages.*
>
> We hope that this will make the paper even more transparent, while providing a clear rationale for the approach!
>
> > Figs 1, 2, 3, etc.. In general, the captions are short and the paper would be easier to read if they were more detailed and self contained, e.g. describe the main observations and make the captions clearly indicate the metrics being evaluated (using the appropriate abbreviations for the metric names).
>
> Thank you for this suggestion. In the final version of the paper, we will include the main findings described in the Experimental results section (1 or 2 sentences) also in the captions of the figures to make them more self-contained! E.g. in Fig. 1 we will highlight the differences in the dynamics of CNNs and ViTs in terms of the similarity of class representations, in Fig. 2, we will mention the 3 phases of similarity development, In Figs. 3 and 4, we will mention that the structure of the class similarity matrices reveals that networks learn to perceive similarity etc. In general, we will reuse some of the fragments from the captions in the appendix (they are longer due to the fact that there are no page limits in the appendix, however we will shorten the paper to include the longer captions for the figures also in the main body of the paper).
>
> > Intro/Appendix. It would be helpful to clarify that main focus of the analysis is on image classification tasks.
>
> According to the suggestion, we added a sentence ‘Our main focus is on the image classification tasks.’ to the introduction and also discussed it in more detail in Appendix A (we highlighted there, that we focus on image classification, however as other tasks are also important, we also provided some additional results for different tasks to improve the generalizability of our findings).
>
> Once again, thank you very much for all your help and support! We did our best to solve all the issues reported by the reviewer. If the reviewer has any remaining concerns, we will be more than happy to answer.

---

### Official Review · Reviewer_7tX6 · 2024-10-27

**Soundness:** 3
**Presentation:** 2
**Contribution:** 3
**Rating:** 6
**Confidence:** 3

**Summary:**

This paper explores how vision models learn the similarity structure within the data over the course of training. The authors focus on image classification tasks. To achieve this, the authors propose a set of measurements specifically designed to quantify the similarity between different classes in the data as perceived by the network. The basic structure of the measurements is comparing two NxN similarity matrix over the classes. The similarity matrices can come from different stages of training or different ways to measure similarity.  They apply these measurements over the course of training for several models to identify different phases of similarity learning. The authors also provide a similarity measure to compare model perceived class similarity with WordNet (as a proxy for human-perceived class similarity). The authors find some interesting properties from analyzing several models, these will be detailed in the strengths section.

**Strengths:**

I find that the works major strengths are in question selection, experimental design and results.

The authors ask an interesting question about how class similarity perception evolves during training and how it relates to task performance.

Through studying this question, the authors discover some interesting properties, specifically, they show that networks go through three phases while learning the similarity between classes. In the first stage, models discover the coarse structure of similarity between classes, followed by a stage in which models start to perceive differences between similar groups, and finally a stage where the where similarity is roughly static and the model has stabilized.

Additionally, the authors discover that model's initially make errors according to the perceived similarity structure, but over time model mistakes diverge (they make mistakes between less similar classes) and do not follow the similarity structure as closely. Finally, the authors discover an interesting pattern of "bumps" in the alignment between WordNet similarity and model similarity.

**Weaknesses:**

The primary contributions of this work are in the question and experimentation, since the methods are derived from previously developed methods (as cited by the authors themselves).

Therefore, my assessment of the work is focused on the experimental design and results. I will first summarize my primary concerns and then provide more detailed criticisms. While I find the results provided by the authors to be interesting, I believe the phenomena discovered to be under-explained, and I would like the authors to go one step deeper to understand the some of the properties discovered in the paper.

(W1) In particular, this paper could be significantly strengthened by providing an experiment to concretely explain the decrease in errors only IDM (Fig. 6b). The authors hypothesize that this decrease is due to less typical samples or noise from mislabelled data, but provide no experiments to test these hypotheses. See Q2.

(W2) I am not convinced of the value of the WordNet experiments, see Q1. However, provided there is value, a closer inspection of the cyclical nature of the bumps would be interesting. I prioritize this less than (W1).

(W3) The difference in curves between mean weights similarity for ResNet18 and MobileNetV2 and the other models is surprising and poorly explained (Fig 1a). I prioritize this less than (W1).

(W4) I was not able to find a place where the authors clearly stated whether the experiments were conducted with training data or test data. Assuming the results are using test data, the results are missing the same plots for the training data. Is some of the phenomena we see due to overfitting on the training data?

(W5) Please provide a detailed description of how WordNet similarity is computed in the appendix.

(W5) Additionally, I also find some significant weaknesses in writing style.
Line 053 - "This makes similarity barely explored to this day" is too strong of a statement.
Line 202-215 - Dissimilarity Metric (DM) is not precisely defined please provide an equation.
There is a tendency to anthropomorphize models, for example the word "strive" is used as if the model has a goal (Line 419, 503, 516). Also, "stick to it" (Line 505).

(W6) Minor weaknesses in writing:
Line 223-224: took two passes to understand.
Line 199 - should be indirect similarity?
Line 268 - "WordNet" leaf
Line 211 - SCSM used again, use a different term

**Questions:**

(Q1) I am not sure of the value of semantic similarity comparisons using WordNet, considering language based similarity and visual similarity should have significant differences. Can the authors speak more about the value of this experiment and how it can be interpreted?

(Q2) Related to (1), I am also surprised that the authors do not measure similarity using euclidean/cosine distance over each sample which by my understanding would be a direct and functional method of measuring similarity. Why is this not included? Wouldn't this metric allow you to inspect which samples specifically deviate from the model's perception of similarity.

---

> ### Author Response · Authors · 2024-11-24
>
> Thank you for spending a significant time and effort to provide us with many useful comments and suggestions. Thank you for appreciating our question selection, experimental design and results. We are glad that the reviewer perceives our work, our findings and the discovered phenomena interesting.
>
> Below, we answer the specific weaknesses/questions. We did our best to incorporate all the changes requested by the reviewer to improve our paper. We marked all the changes in pink in the revised version. If the changes and responses meet the reviewer’s expectations, we would kindly ask the reviewer to reconsider the given score. Thank you very much for your time and effort.
>
> >W0 The methods are derived from previously developed methods (as cited by the authors themselves).
>
> The methods are derived from previously developed models ONLY on a very low level. E.g., the cited way to extract class templates via weights was used for adversarial attack evaluation/knowledge distillation, and not for creation of similarity-based evaluation frameworks. Also, only one of the framework's metrics (IDM) is adopted from a different work, in which it was used for a completely different purpose (also attack evaluation). Therefore, this is novelty by a new area of application, as well as aims and means of analysis. Moreover, the majority of the metrics and the framework itself are novel and unique contributions that for the first time enable such a thorough, systematic and flexible analysis of different networks trained with different datasets and training paradigms. That is why, they are also an important contribution to the field that enables the model similarity-based evaluation above simple accuracy testing.
>
> >(W1) In particular, this paper could be significantly strengthened by providing an experiment to concretely explain the decrease in errors only IDM (Fig. 6b). The authors hypothesize that this decrease is due to less typical samples or noise from mislabelled data, but provide no experiments to test these hypotheses.
>
> Thank you for this insightful suggestion. As suggested, we decided to dig deeper into the reasons for mistakes at different epochs. To do this, we provided an additional analysis in new **Appendix H** to test our hypothesis. First, we computed the cosine similarity values obtained for templates extracted for correctly and incorrectly classified images in different epochs for an example network - MobileNetV2. The results show that incorrectly classified samples significantly diverge from the network’s perception of a given class, especially in the later epochs of training (supporting ours and the reviewer’s intuition). We also performed a manual analysis of the incorrectly classified images for 3 epochs (5, 50, 299). The results further support our hypothesis that the mistakes in the later epochs are due to less typical and difficult samples (e.g. a dog hidden in a semi-transparent box, a very small, barely visible target object), mislabeling issues (e.g. wok - frying pan) or co-ocurence of objects (so mistakes that are “not fully mistakes”).
>
> TBC in the next comment...

---

> ### Author Response · Authors · 2024-11-24
>
> >(W2) A closer inspection of the cyclical nature of the bumps would be interesting.
>
> Thank you for this, another interesting suggestion. To inspect these bumps more closely, we took an example network - EfficientNetV2Bo (used in our additional experiments with CNNs) - and plotted its SAI(NCSM, SCSM) curves in the same plots as training loss and learning rate (we created a new **Appendix I**). These plots show that most probably these bumps are caused by the learning dynamics (bumps are also visible in the loss plot) and the the changes of the learning rate scheduler (the bumps occur when the learning rate scheduler makes the learning rate first constant and then utilizes learning rate decay with abrupt learning rate values changing). These abrupt changes can cause the loss and the SAI curve to temporarily spike as the optimizer adjusts. It is supported also by the fact that the SAI values stabilize s when learning slows down and the changes in the learning rate become less abrupt
>
> >(W3) The difference in curves between mean weights similarity for ResNet18 and MobileNetV2 and the other models is surprising and poorly explained (Fig 1a).
>
> Our understanding is that CNNs differ in similarity metric scores from transformer-type and hybrid networks, and we acknowledged it in our paper. To delve deeper into the differences between the networks, due to the limited number of pages in the main body of the text, we have added an extended analysis of this phenomenon in the Appendix G. In order to further prove our finding, we included additional experiments with two other CNN networks - DenseNet and EfficientNe that align with our findings that such a curvature is characteristic for CNNs.
>
> >(W4) I was not able to find a place where the authors clearly stated whether the experiments were conducted with training data or test data. Assuming the results are using test data, the results are missing the same plots for the training data. Are some of the phenomena we see due to overfitting on the training data?
>
> We provide our training-testing curves in **Appendix B.2.** While there are some indications of overfitting (for ViT model, as it also scores the lowest in the testing dataset), overfitting should not play major role in the experiments we performed, as ViT model is also our model example of “good” and “bad” performing network in **Appendix E**. Also in the newly added **Appendix J**, we present yet another way for measuring the network-perceived similarity (based on templates). The results show that the class similarity matrices obtained for two splits of the dataset (train, test) are highly similar (close to 1), and high similarly of the matrices compared to the weights-based class similarity matrix further shows NOT a significant impact of overfitting on these measurements, however at the same time there is a potential to use the subtle differences to detect overfitting based on similarity measures (which is an additional plus).
>
> >(W5) Please provide a detailed description of how WordNet similarity is computed in the appendix.
>
> We included such description in Appendix B.1 (semantic similarity is computed for each pair of the classes in the dataset as an inverse of the distance of the shortest path connecting them in the WordNet linguistic taxonomy (path connecting two nodes in the WordNet taxonomy tree)).
>
> >(W6) Minor weaknesses in writing
>
> We reviewed our paper one more time and fixed the typos, sentences that were hard to read and other issues.
>
> TBC in the next comment...

---

> ### Author Response · Authors · 2024-11-24
>
> >(Q1) I am not sure of the value of semantic similarity comparisons using WordNet, considering language based similarity and visual similarity should have significant differences. Can the authors speak more about the value of this experiment and how it can be interpreted?
>
> Thank you for raising this point. While semantic and visual similarity are not identical, they are not entirely separated. It is because semantic similarity (and the structure of the WordNet semantic tree) often stems from shared functionality, appearance, common ancestry, evolutionary traits, and shared areas of application or context. Comparing learned visual with semantic similarities allows us to evaluate how well models capture these meaningful semantic connections and relationships between categories. Moreover, imperfect alignment between semantic and visual relationships can also be valuable, as it provides additional insights into how models perceive and process information, highlighting potential areas for improvement or uncovering new perspectives on class relationships.
>
> >(Q2) I am also surprised that the authors do not measure similarity over each sample which by my understanding would be a direct and functional method of measuring similarity. Why is this not included? Wouldn't this metric allow you to inspect which samples specifically deviate from the model's perception of similarity.
>
> We performed this type of experiment in **Appendix H**. In there, we provide a table visualizing how average similarity between training data and correctly labeled images is increasing, while decreasing for incorrectly labeled images at the same time during the training, for both training and testing datasets. We also provide some examples of misclassified images during the course of training, and provide explanations for the reason of misclassification that also align with the development of models’ internal representations of similarities.

---

> > ### Comment · Area_Chair_8Mnd · 2024-11-25
> >
> > Dear Reviewer,
> >
> > The authors have provided their responses. Could you please review them and share your feedback?
> >
> > Thank you!

---

> ### Author Response · Authors · 2024-11-25
> **additional response**
>
> Regarding this:
>
> > Additionally, I also find some significant weaknesses in writing style. Line 053 - "This makes similarity barely explored to this day" is too strong of a statement. Line 202-215 - Dissimilarity Metric (DM) is not precisely defined please provide an equation. There is a tendency to anthropomorphize models, for example the word "strive" is used as if the model has a goal (Line 419, 503, 516). Also, "stick to it" (Line 505).
>
> According to the suggestion, we de-anthropomorphized models by limiting the use of such phrases and also some strong statements. We also defined better the DM metric in the additional appendix (and also highlighted that the presented WordNet-based version of it is our own idea there). This has been included in new Appendix L.

---

> ### Comment · Reviewer_7tX6 · 2024-11-25
>
> (Appendix H) - How is the errors-only IDM curve impacted by number of samples? I assume the number of errors decreases over time such that the remaining errors are more dissimilar.  Could you include this information somehow?
>
> (Appendix I) - This section is clear, thanks for including it.
>
> (Appendix G) - Also clear.
>
> (Appendix L) - This is a helpful addition.
>
> Minor Notes
> 1) Quote directions are wrong Line 332
> 2)  "with visible ’bumps’ in the curve (we analyze why do they occurin Appendix I)" This new text reads quite clunky, please improve how your incorporate these references in the final version.
> 3) Line 1574: The results supprot our hypothesis (typo).
> 4) Weight Similarity Index and Weight**s** Similarity Index are both used.
> 5)  Typos: Line 1390, Line 1756 (Self-sup -> self-sup), Line 1761 (Section -> section), Line 1797 (TCSM -> TNCSM)
>
> This is not a comprehensive list of issues, just what I noticed...
>
> Overall, the new additions strengthen the work. Provided the authors address my question about Appendix H, I would improve my rating to a 6. Please also review the text once more to check for grammar errors and flow. Currently the writing issues detracts from the presentation of the work.

---

> > ### Author Response · Authors · 2024-11-26
> >
> > We would like to thank you once more for your help! We also truly appreciate your willingness to raise your score. To provide the answer to the reviewer’s comment, we generated some additional results, on which we comment below. If there is anything more we can do to further improve our paper in the eyes of the reviewer or we can answer any further questions, we will be more than happy to do it. Given the many advantages highlighted by the reviewers, we believe that our paper has the potential to make a meaningful contribution to the community and spark further interest in this important area of research.
> >
> > > Comment: (Appendix H) - How is the errors-only IDM curve impacted by number of samples? I assume the number of errors decreases over time such that the remaining errors are more dissimilar. Could you include this information somehow?
> >
> > We agree with the reviewer that the number of mistakes has an impact on the errors-only IDM. However, it is an advantage compared to the standard IDM, because it allows us to better inspect the quality of predictions for the cases in which a given network makes mistakes.  As suggested by the reviewer, we included a plot presenting the number of testing mistakes (at a given epoch) and a plot presenting the histograms of cosine similarity values between the incorrectly classified samples and their templates for 4 example epochs in Appendix H. The first plot shows that the number of mistakes decreases in a logarithmic fashion (which of course is expected, as the network learns its task). We also agree that as the number of errors decreases, the remaining errors caused by the less typical/difficult etc. samples become more prominent within the set of samples for which the network made mistakes (this can be seen in the histogram, in which it is visible, that the errors with templates that are the most similar to their ground truth labels are eliminated during the course of training). That is why this slight drop occurs, but overall (which can be seen as a still high value of IDM) mistakes are still driven by the similarity perception of the network, which aligns with our earlier findings. We added this discussion to the Appendix.
> >
> > When it comes to the minor notes, thank you for providing us with these examples! As we performed a lot of additional experiments during the rebuttal (that required a lot of additional code) and added a lot of text to the paper, we did manage to check the text as carefully as we wanted to, therefore we are grateful for this help. We have corrected the mistakes in the paper and also some other minor mistakes we were able to find.

---

> > > ### Comment · Reviewer_7tX6 · 2024-11-27
> > >
> > > Thanks for the update, I have increased my rating.

---

> > > > ### Author Response · Authors · 2024-11-27
> > > >
> > > > We would like to thank the reviewer for their engagement and support! We would also like to report that we used the additional day for the paper modifications to thoroughly read and edit our paper to improve its presentation, as the reviewer suggested. We focused on:
> > > > * fixing the typos/grammatical errors,
> > > > * improving the punctuation,
> > > > * shortening the long sentences,
> > > > * readability issues,
> > > > * fixing some remaining incorrect quotation marks,
> > > > * refining the RQs a little bit for a better clarity,
> > > > * improving the flow of the sentences (by using more adequate words and rebuilding the sentences).
> > > >
> > > > We hope that now the writing issues do not detract the presentation and quality of our work and satisfy the standards of the reviewer. Once again, thank you for your invaluable contribution to help us improve our paper!

---

### Official Review · Reviewer_VJyB · 2024-11-02

**Soundness:** 2
**Presentation:** 3
**Contribution:** 1
**Rating:** 5
**Confidence:** 4

**Summary:**

This paper develops a framework for evaluating how semantic similarity perception emerges in learned visual representations. The motivation is to draw parallels and insights in to the emergence of similarity perception in humans. The authors primarily evaluate the task of supervised object recognition and compare the effects of architecture on similarity emergence. The main methodological contributions are new numerical, semantic similarity metrics, which reduces the need for difficult to scale qualitative measures. The primary finding is that over time models learn to make fewer and better mistakes. This learning is initially quick, then starts to refine and stabilize, with CNNs generally developing semantic similarity perception faster than ViTs.

**Strengths:**

The work at a high level is generally clear. The construction of the network, confusion, and semantic similarity matrices which are used to plot different curves during training seem technically reasonable. The goal of understanding developing qualitative measures of semantic similarity, particularly in the context of human cognitive science, is valuable from my perspective and is motivated well in the introduction. I agree that many approaches comparing to humans are very data-dependent, limiting their applicability, and think the use of WordNet is an interesting approach.

**Weaknesses:**

My biggest concern with this work is the focus on supervised object recognition training. Both from the perspective of computer vision and deep learning-based models of human vision, there are many other more powerful training objectives for visual learning. Self-supervised approaches, in particular, learn strong semantically meaningful representations (like DINO) and have been shown to produce strong candidate models of the visual processing (SimCLR for example is used in Margalit et al. 2024). While these models could additionally be trained downstream on a supervised objective, the proposed method would not be able to measure earlier effects during training where the bulk of similarity perception is likely to emerge. Relying on object classifiers limits the applicability of the proposed method to more powerful representation learning frameworks, especially for during training analyses.

In addition, regardless of the task, the training results presented using the proposed metrics do not appear particularly significant in my view. From the training loss and accuracy alone, it should be clear that the models rapidly get better at similarity and then stabilize. For similar reasons, it is unsurprising that CNNs learn similarity more quickly than ViTs, which are known to need large amounts of data to perform well on these small supervised tasks.

Finally, the connection to human cognition presented is weak, in my opinion. The only measure grounded in human studies is the use of WordNet for the semantic similarity measure. While this is a clever idea, there are no direct links to cognitive science results beyond this. In addition, WordNet is not grounded in work visual perception, being primarily language. In a study looking at representational similarity in classifiers, I believe it would be stronger to make connections to datasets and models that use human judgements or interpretations on images, or to instead evaluate vision models that more explicitly learn text relations, such as CLIP.

Minor: there are many metrics and acronyms presented which can be a bit confusing to keep track of.

**Questions:**

Can you elaborate on the choice of training objectives and models? In addition, do you see any use for your metrics on larger pre-trainined models, or the exploration of factors other than architecture such as model size?

Can you explain what insights your metrics contribute beyond what standard training metrics such as loss and accuracy yield?

Do you view your work as primarily trying to contribute new techniques for building cognitive science models? Can you elaborate if so? In your opinion, does your work bring insights into representation learning more broadly?

---

> ### Author Response · Authors · 2024-11-24
>
> We would like to thank the reviewer for an in-depth analysis of the work and providing us with many useful comments and suggestions that helped us improve our work. We are glad that the reviewer acknowledges the clarity of our work, the technically reasonable formulation of our methods, and the importance of using semantic similarity and WordNet to enable research in cognitive research as a data-independent and applicable alternative to human polls.
>
> Below, we answer the specific weaknesses/questions. We did our best to incorporate all the changes requested by the reviewer to improve our paper. We marked all the changes in pink in the revised version. If the changes and responses meet the reviewer’s expectations, we would kindly ask the reviewer to reconsider the given score. Thank you very much for your time and effort.
>
> >W1 Using only networks trained in a supervised manner. Relying on object classifiers limits the applicability of the proposed method to more powerful representation learning (e.g. networks such as DINO) frameworks and is not possible for their analysis during training.
>
> Thank you for this comment! We focused our research on supervised objectives and object classifiers, because such networks are not explicitly forced (or helped via providing necessary annotations) to perceive similarity well (therefore learning similarity is not trivial for them). Using classifier’s weights where possible is a significant advantage, as it does not require extracting features from large datasets to obtain meaningful class templates and subsequently - class similarity matrices. Nevertheless, using such extracted templates IS an option and can be treated as an extension of our framework for networks without such a classifier. Such an extension enables the similarity examination of networks such as DINO (and other networks trained for representation learning). To show this possibility, we discussed this aspect in new **Appendix J** and provided some additional experiments there. First, we took an example network with a classifier (MobileNetV2) and generated template-based class similarity matrices for its training and testing set (cosine similarity between them close to 1 shows that they are highly similar). We also compared them with the weights-based class similarity matrix showing that also these matrices are highly similar for different epochs (over 0.9 cosine similarity), therefore this template-based variant can be used as an approximation of the weights-based similarity (we did not use this method initially in our study, as the focus of our framework was on the computational efficiency of our methods). Next, we use this method to obtain the network class similarity matrix for DINOv2. The matrix presents a highly similar pattern to the one in WordNet-obtained matrix and the ones obtained for object recognition networks trained on mini-ImageNet. The numerical result for alignment between the CSM and semantic similarity (SAI(NCSM, SCSM)) is 0.85, therefore very similar to the traditionally trained networks. The results also show the possibility of using our frameworks for networks trained with different training paradigms.
>
> TBC in the next comment...

---

> ### Author Response · Authors · 2024-11-24
>
> >W2 The training results presented using the proposed metrics do not appear particularly significant in my view. From the training loss and accuracy alone, it should be clear that the models rapidly get better at similarity and then stabilize. For similar reasons, it is unsurprising that CNNs learn similarity more quickly than ViTs.
>
> Thank you for your feedback. We respectfully disagree that the proposed metrics do not provide any additional insights. While accuracy and loss do inform about the performance gains of the models, in the traditional form (overall accuracy, categorical cross entropy used for loss) they both take into consideration ONLY the binary information of the image’s label (therefore it is only the binary success: whether the network answered correctly or not). What they do not take into consideration are the inter-class relationships, e.g. expressed via similarity between them. They do not show whether the network makes mistakes between very similar (reasonable) classes (e.g. two breeds of dogs) or very dissimilar classes (e.g. a dog and a hammer). Our framework and metrics are specifically designed to fill this gap, and show how models structure inter-class relationships and how these evolve during training. E.g. via Network and WordNet based Inverse Dissimilarity Metric, our framework shows whether the mistakes are close to the ground truth according to the Network’s perception of similarity or semantic similarity. Other metrics show how the network structures its knowledge via examining the relationships between classes (whether they match the mistakes being made or semantic similarity). While matching the semantic similarity should be natural due to the correlated structure of visual data, it is NOT that trivial because the networks trained with supervised procedures are NOT provided with similarity information so whether they discover these similarities well is an indicator of a good operation and can EXPLAIN good accuracy or decreases in loss. The fact that the results align with expectations, such as CNNs learning similarity faster than ViTs, is actually a strength of the framework. It validates and contextualizes previously observed differences from a novel perspective and adds a new surface of understanding, providing fresh insights into the dynamics of different architectures. To sum up, accuracy and loss allow for simple, binary evaluation, while our framework and metrics - for more nuanced evaluations of network behavior.
>
> >W3 The connection to human cognition presented is weak, in my opinion. WordNet is not grounded in visual perception, being primarily language. In a study looking at representational similarity in classifiers, I believe it would be stronger to make connections to datasets and models that use human judgements or interpretations on images, or evaluate vision models that more explicitly learn text relations, such as CLIP.
>
> We agree that we may have overemphasized the focus on human perception of similarity in our initial text. We chose semantic similarity to build our framework because, unlike human judgments obtained through large polls, semantic similarity data can be easily extracted from lexical structures such as WordNet and its similarity judgments are consistent. It enhances the consistency, generality, and practicality of the proposed framework and enables others to use it with different models and datasets. In our study, we focused on purely visual models trained on supervised approaches (with no similarity enforcement), because it should be more demanding for such models (with less information given) to learn these non-trivial similarity relationships in only visual data. We however agree that the evaluation of text-image models such as CLIP would be an interesting addition, therefore in **Appendix J**, besides DINOv2, we also analyzed CLIP (via the described template-based approach to class similarity matrix). The results show that its class similarity matrix also has a highly similar structure to the ones on DINOv2, traditional classifier-based networks and WordNet (with slightly more ‘noise’ outside of the main semantic groups stemming most probably from text relations). Its alignment with semantic similarity is 0.82, so also very similar to traditional networks. These results further show that our framework can be used for other training paradigms.
>
> TBC in the next comment...

---

> ### Author Response · Authors · 2024-11-24
>
> >W4 Many metrics and acronyms
>
> To improve this aspect, where possible (due to a strict page limit), we included some additional short interpretations of the acronyms, e.g. we modified the description of Fig. 2 to highlight the purpose and meaning of SAI(NCSM, SCSM): Mini-ImageNet: SAI(NCSM, SCSM) Curves (network-semantic similarity alignment).
>
> >Q1 Can you elaborate on the choice of training objectives and models? In addition, do you see any use for your metrics on larger pre-trainined models, or the exploration of factors other than architecture such as model size?
>
> As mentioned in W3, we focused on purely visual models trained on supervised approaches (with no similarity enforcement), because it should be more demanding for such models (with less information given) to learn these non-trivial similarity relationships in only visual data. Nevertheless, as shown in additional **Appendix J**, our framework can be used also for contrastive, similarity-based models such as DINOv2 or text-image models, such as CLIP, showing that our framework is highly flexible and can be used for the evaluation of different models with different training objectives. Also, the proposed numerical measures can be of course used for exploration of other factors than the ones presented in this study (e.g. as suggested by reviewer **Mwfz** for the analysis of adversarial robustness).
>
> >Q2 Can you explain what insights your metrics contribute beyond what standard training metrics such as loss and accuracy yield?
>
> Please, see the answer to W2.
>
> >Q3 Do you view your work as primarily trying to contribute new techniques for building cognitive science models? Can you elaborate if so? In your opinion, does your work bring insights into representation learning more broadly?
>
> We believe that due to a highly flexible nature of our framework it can have a broader impact on areas of representation learning, model evaluation and explainability. First, it provides easy to interpret numerical measures to evaluate models from a similarity-based perspective (providing more informative and deeper insights than accuracy or loss). As shown via the additional experiment with CLIP and DINOv2 (Appendix J), it can be used not only with models trained with supervised procedures, but also with other networks and training paradigms focused e.g. on representation learning. The framework (with different similarity matrices given as reference - e.g. reflecting human judgements gathered via polls - instead of semantic similarity) can be also used to evaluate cognitive science models. To sum up, our framework is highly flexible and can be used with numerous models, datasets, tasks and paradigms, and can be easily adjusted to analyze many other aspects than the ones presented in the paper.

---

> > ### Comment · Area_Chair_8Mnd · 2024-11-25
> >
> > Dear Reviewer,
> >
> > The authors have provided their responses. Could you please review them and share your feedback?
> >
> > Thank you!

---

> > > ### Comment · Reviewer_VJyB · 2024-11-26
> > >
> > > I thank the authors for their responses and updates to the paper. Given the new additions and sentiment from other reviews that the results are interesting, I am willing to raise my score to a 5. I still, however, have concerns about the supervised/category label centric nature among other things, so I will not raise it further.

---

> > > > ### Author Response · Authors · 2024-11-26
> > > >
> > > > We understand your concerns regarding the supervised/category label-centric nature of our approach. We would like to elaborate on why we chose to focus on supervised schemes. Our focus was driven by their central role in real-world and industrial applications, as these approaches are still *ubiquitously* used to create applications in these areas. While representation learning paradigms are undoubtedly promising and show great potential, they currently lack the technological maturity and industrial adoption of supervised methods. That is why we chose to prioritize these tasks that are still *invaluable* for practical applications and have transformed computer vision (also none of the remaining reviewers raised the issue that testing the supervised networks from the similarity perspective is less important than networks trained with the focus on similarity perception development). Nevertheless, we do agree with the reviewer that text-image models and models trained with different objectives also significantly change the landscape of computer vision (and are *highly important*) and they focus on the problem that is also at the heart of our paper - similarity. While we have aimed to address these aspects in our revisions (and added the results for DINOv2 and CLIP - therefore we showed that our proposed tool itself can be useful for other training regimes), we recognize that there may still be room for improvement. That is why, we would like to ask the reviewer: What could we do to make our paper more convincing for the reviewer?

---

### Official Review · Reviewer_Mwfz · 2024-11-04

**Soundness:** 4
**Presentation:** 3
**Contribution:** 4
**Rating:** 8
**Confidence:** 4

**Summary:**

This paper describes a group of methods for evaluating similarity between classes in DNNs, called Deep Similarity Inspector. The authors evaluate the evolution of these metrics over training for a variety of network types and identify 3 stages of similarity over training (surge, refinement, stabilization).  They note differences in this evolution between CNN, ViT, and hybrid models, showing that CNNs peak quickly and refine, while ViTs develop similarity more slowly, and remain stable, but note that all models follow this 3 stage process. The paper compares these networks similarity to that of WORDNET similarity as a proxy for human semantic understanding.

**Strengths:**

Originality: DSI is an original contribution as far as I am aware.

Quality: The quality is generally good. The authors offer multiple measures of similarity, rather than just one for evaluating similarity in networks. They also evaluate multiple network types (CNN, ViT, and hybrid)

Clarity: The paper is overall clear in the purpose and execution of the experiments reported.

Significance: This is an important contribution to explainable AI. In addition, though not discussed by the authors, I believe these metrics could offer a unique metric to evaluate models during training (i.e. nothing that a model has moved into similarity stage 3 could be a signal for training nearing completion, or variations in the stages could inform hyper parameter tuning).

**Weaknesses:**

While the proposed formulation of similarity proposed is novel to my knowledge, the authors fail to compare their method to very similar ways of measuring similarity including methods like adversarial examples.

A major goal of the paper appears to be based on aligning networks with human psychology, yet WordNet is the only source of measuring human similarity perception. I feel there needs to be additional methods of comparing this to human perception, given the focus on human semantic alignment. On the other hand, I believe the similarity metrics themselves have merit outside the context of human alignment, especially in explainability, and as a general tool for evaluating model training. Given the lack of human-comparison in the results, it was confusing to read an abstract and introduction so focused on human perception, especially when I believe there are other, potentially stronger applications.

While the clarity of the experiments are good, the explanations of the individual metrics would benefit from intuitive explanations in addition to the mathematical descriptions provided. This is especially important given that the authors are introducing these concepts for the first time in this paper.

It is unclear how robust this method is to contexts outside of vision such as audio or even robotics. I don’t see why this method couldn’t be applied in another modality, but this discussion is completely absent from the paper.

The paper does not discuss the computational cost of computing these metrics, limiting the understanding of how easily it can be incorporated into a training pipeline.

**Questions:**

What is the computational cost of computing the proposed metrics during training? Is this something that could be cheaply calculated alongside the loss graph to aid in everyday training?

Could this metric be formulated as part of a loss function to improve accuracy or robustness? Does higher dissimilarity correlate with higher adversarial or other types of robustness?

Is there another measure of human similarity that could be compared to in addition to wordnet? This would strengthen the paper significantly - I see this as the biggest weakness of the paper given the strong human motivation in the abstract and introduction. Alternatively, the introduction and abstract could be modified to discuss the importance of the contributions in the context of model evaluation rather than human similarity.

L83-86: Please phrase this differently - ‘the real questions are’ suggests that these questions posed by the authors are the only meaningful or important inquiries in our field. The previous approaches cited are also important and valuable. A simple rephrase such as  “We focus on the related questions of…” would fix this, and acknowledge there are a broad spectrum of important questions in machine learning, and the authors are most interested in focusing on those in this paper.

Relatedly, the authors say there are only 2 (Bial et al 2017, Huang et al 2021) works that examine whether the similarity perception of deep object recognition models aligns with human judgements. First, this paper does not evaluate human judgements directly, only via wordnet. Furthermore, there are many more studies that compare object recognition with humans and measure related metrics of similarity. Adversarial examples for example measure similarity perception through the effectiveness of perturbations, and many previous works have measured this in both humans and models (ie Elsayed, et al 2018 Adversarial Examples that Fool Both Computer Vision and Time-Limited Humans.).  I suggest the authors broaden their literature review to include this and other examples of the many ways that similarity is measured in humans and models, and more fairly situate their contribution in the context of the field.

---

> ### Author Response · Authors · 2024-11-24
>
> First of all, we would like to thank the reviewer for deeply analyzing our paper and providing us with many helpful comments. We are glad that the reviewer acknowledges the originality of our methods, good quality of our work, thorough evaluation, clarity and significance of our methods and experiments for explainable AI. We would also like to thank the reviewer for providing us with additional concrete strengths of our paper that we did not highlight enough, and which helped us to strengthen our paper.
>
> Below, we answer the specific weaknesses/questions. We did our best to incorporate all the changes requested by the reviewer to improve our paper. We marked all the changes in pink in the revised version. If the changes and responses meet the reviewer’s expectations, we would kindly ask the reviewer to reconsider the given score. Thank you very much for your time and effort.
>
> >W1 Comparison with using adversarial attacks for measuring similarity
>
> Thank you for this suggestion! Using adversarial attacks can be an alternative to measure mistakes-based similarity. Adversarial attacks were not used initially in our framework for measuring similarity perception of networks, as our focus was, as the reviewer mentioned later, computational efficiency  (to enable testing during training and after each epoch). Generating adversarial perturbations introduces significant costs, particularly with iterative attacks. Moreover, the attack choice is highly variable, introducing additional complexity. However, we agree with the reviewer that incorporating adversarial attacks is an interesting extension. To explore this, we conducted smaller-scale experiments using MobileNetV2 and SwinV2T (1 CNN, 1 ViT) and Projected Gradient Descent (PGD) attack and demonstrated that confusion between originally predicted and post-attack labels can be used to create alternative, confusion-based class similarity matrices (we showed some examples CSMs), which naturally are compatible with our framework (we included the experiment in **Appendix K**. Additionally, we analyzed the fooling rate (as an indicator of adversarial robustness) along with our Semantic Similarity Index Curve for Network Similarity and Semantic Similarity in one plot. The results show that in later epochs, when the Semantic Alignment Index (SAI) decreases, adversarial robustness also decreases, potentially due to overfitting. This finding supports the utility of SAI as an indicator of robustness and demonstrates potential future applications of our framework in the area of security.
>
> >W2 WordNet is the only source of measuring human similarity perception - it was confusing to read an abstract and introduction so focused on human perception.
>
> Please, see the answer to Q3.
>
> >W3 Lack of discussion about using the methods outside of vision such as audio or even robotics. I don’t see why this method couldn’t be applied in another modality.
>
> Yes, our framework can possibly find application in domains outside of computer vision, as similarity principles (not in the exact, but similar form) persist in different domains, such as language, audio and practical application domains, such as robotics, autonomous vehicles etc. Thank you for this suggestion! We added the discussion on possible extended use of our framework in **Conclusion**.
>
> >W4 No discussion on the computational cost of computing these metrics & Q1 What is the computational cost of computing the proposed metrics during training? Is this something that could be cheaply calculated alongside the loss graph to aid in everyday training?
>
> Yes, all metrics generated with our framework can be cheaply calculated alongside the loss graph to aid in everyday training, which we believe is a significant advantage of our framework. All its operations can be implemented as either simple mathematical operations or are computed based on already obtained data e.g. for loss or accuracy computation, which is done in standard procedures. We measured the time and included it in **Appendix A**, **Section A.2**: “The additional complexity of computing our new metrics accounted for approximately **0.29%** of epoch time computation for MobileNetV2, and it stayed consistent for other networks as well, making it an insignificant addition to the total training time.”. These results show that the overhead needed for computation of our methods is almost negligible.
>
> TBC in the next comment...

---

> ### Author Response · Authors · 2024-11-24
>
> >Q2 Could this metric be formulated as part of a loss function to improve accuracy or robustness? Does higher dissimilarity correlate with higher adversarial or other types of robustness?
>
> Thank you for the insight - although not stated in the paper, as it is outside of its scope, our goal was to provide with exactly such a metric for e.g. monitoring networks performance during training. In **Appendix E**, we are considering the possibility of using our metrics not only as a loss function (or part of it for regularization purposes), but also as an early indicator of networks’ performance or signs of its overfitting. We also intend to consider it as a supplementary indicator for helping with learning rate scheduling, early stopping or checkpointing (for e.g. pre-training phase in transfer learning applications). Also, as discussed broader in W1, our additional results presented in **Appendix K** show the possibility of using the proposed metrics as indicators of the adversarial robustness degradation (as the reviewer suggested, lower dissimilarity correlates with lower adversarial robustness in later phases of training).
>
> >Q3 Is there another measure of human similarity that could be compared to in addition to wordnet? The introduction and abstract could be modified to discuss the importance of the contributions in the context of model evaluation rather than human similarity.
>
> We agree with the reviewer that we may have overemphasized the focus on human perception of similarity in our initial text. As the reviewer points out, human similarity is broader than just semantic similarity. We specifically chose semantic similarity to build our framework upon it because, unlike human judgments obtained through large polls, semantic similarity data can be easily extracted from lexical structures such as WordNet for various classes and general-purpose datasets. This approach’s advantages are consistent similarity judgments and a fixed structure from which similarities are computed, which enhances the consistency, generality, and practicality of the proposed framework and enables others to use it for their models and datasets to evaluate networks. However, we acknowledge that for certain datasets, extensions incorporating human similarity judgments have been proposed, so we cited works such as [1] in our expanded **related work** section. To address the weakness and improve clarity, we have modified our **abstract**, **introduction**, and **related works** section to better emphasize that our use of semantic similarity is because of its advantages for computational modeling. We also highlighted the contributions of our framework in the context of model evaluation, rather than as a comprehensive measure of human similarity perception.
>
> [1] Brett D Roads and Bradley C Love. Enriching imagenet with human similarity judgments and psychological embeddings. In Proceedings of the ieee/cvf conference on computer vision and pattern recognition, pp. 3547–3557, 2021.
>
> >Q4 L83-86: Please phrase this differently
>
> We of course perceive the previous approaches cited as important and valuable and highly contributing to the field, which we also aim to do by providing a novel framework and insights. As suggested, we modified this phrase (“we focus on examining to what extent …”) to acknowledge there are a broad spectrum of important questions in machine learning, and that we are most interested in focusing on some particular aspects of this domain in our paper.
>
> >Q5 I suggest the authors broaden their literature review to include this and other examples of the many ways that similarity is measured in humans and models, and more fairly situate their contribution in the context of the field.
>
> Thank you for this suggestion, we modified our **related work** (and also mentioned this in the abstract and introduction) to highlight that similarity has many different notions, e.g. visual similarity, feature similarity, contextual similarity, adversarial similarity and semantic similarity. We cited different additional works that focus on these similarity types to better place our research and make the literature survey more comprehensive. We also emphasized that semantic similarity is one of the proxies of how people perceive similarities (as it is based on shared functionalities, appearance, evolutionary traits, or contextual usage) and highlighted why we decided to use this similarity notion to build our framework upon it (consistency, possibility to use lexical databases to compute it for many categories and datasets to make the framework generalizable and practical for future usages).

---

> > ### Comment · Area_Chair_8Mnd · 2024-11-25
> >
> > Dear Reviewer,
> >
> > The authors have provided their responses. Could you please review them and share your feedback?
> >
> > Thank you!

---

> > > ### Comment · Reviewer_Mwfz · 2024-11-26
> > > **Response to Rebuttal**
> > >
> > > I appreciate the author’s thorough and complete responses and many updates to the manuscript.
> > >
> > > W1) Adversarial Attacks: The additional results here are exciting, showing that Semantic Similarity Alignment goes down as fooling rate goes up. It would be interesting to compare SSA for adversarially-trained networks, or see the curve in Figure K1 for a model trained with SSA as a regularizer, but this could be future work. I agree with the authors that adversarial robustness is not practical to evaluate continuously during training, but given these results, perhaps SSA given its lower computational cost could serve as a proxy for adversarial robustness in some cases. The authors have strengthened the paper significantly by including these additional analyses.
> > >
> > > W2/Q3) Other Domains: This is now addressed with the author’s updates.
> > >
> > > W4) Computational cost: I appreciate the authors evaluating this for MobileNetV2. This result supports this metric as something that could be evaluated during training and/or used as a regularizer.  it would be helpful to see a table with this value for the ‘other networks’ mentioned, as this is a bit vague.
> > >
> > > Q2) I thank the authors for making this clear now.
> > >
> > > Q3) I appreciate the authors including this in the update abstract, intro, and related works.
> > >
> > > Q4) This change greatly improves the tone of the paper.
> > >
> > > Q5) This additional references better situate the paper in terms of related work.
> > >
> > > The authors have well addressed my weaknesses and questions, and the paper is much stronger in my opinion. I appreciate the author’s good faith participation in the review process and significant efforts in incorporating all of the reviewer’s suggestions, addressing questions & weaknesses. I believe this is a good paper that well passes the acceptance threshold. I strongly recommend acceptance and raise my score to an 8.

---

> > > > ### Author Response · Authors · 2024-11-27
> > > >
> > > > We would like to thank the reviewer for their engagement in the reviewing process and the discussion. We are truly grateful for the time invested in reviewing our paper and providing all the constructive suggestions, especially the one about the adversarial robustness! We also think that the additional results are exciting, and that is why it is a perfect inspiration for our future work. We are flattered that the reviewer perceives our paper as good and well passing the acceptance threshold. Thank you for your kindness, support and strongly recommending the acceptance of our paper!

---

> ### Author Response · Authors · 2024-11-27
>
> Having expressed our sincere gratitude in the above comment, we would like to address the reviewer's remaining concern:
>
> > W4) Computational cost: I appreciate the authors evaluating this for MobileNetV2. This result supports this metric as something that could be evaluated during training and/or used as a regularizer. it would be helpful to see a table with this value for the ‘other networks’ mentioned, as this is a bit vague.
>
> To address this issue, we added a table with the results for 3 more networks to **Appendix A**. The results further support the fact that our metrics could be computed during training (which is actually implemented by default in the implementation of Deep Similarity Inspector provided by us) and/or used as a regularizer. Also, as suggested by other reviewers, we did our best to fix the text issues in the paper. We hope that these changes improve the presentation and quality of our work, and along with the additional results satisfy the reviewer’s high standards!

---

> > ### Comment · Area_Chair_8Mnd · 2024-11-30
> >
> > Dear Reviewer,
> >
> > The authors have provided their responses. Could you please review them and share your feedback?
> >
> > Thank you!

---

> > > ### Comment · Reviewer_Mwfz · 2024-12-02
> > > **Response to Rebuttal Part 2**
> > >
> > > Yes, these additional results further strengthen the paper, which is clear, well-executed with good control experiments and thorough. It makes a valuable contribution to the field.
> > >
> > > While the paper is very strong, I reserve a score of 10 for work that I believe is exceptional and transformative to the field. While this paper is highly commendable, it falls slightly short of that threshold for me. Therefore, I maintain my score of 8, reflecting my assessment of this as a very strong paper that will be of interest to the ICLR community.
> > >
> > > I appreciate the authors' effort in refining the manuscript and am excited about this paper - I believe this work will have a meaningful impact.

---

> > > > ### Author Response · Authors · 2024-12-02
> > > >
> > > > We would like to thank the reviewer one more time for all their help, support and kind words! We did what we could to refine the paper and take full advantage of the work of the reviewer and other reviewers! We are very grateful for acknowledging that! It means a lot to us that the reviewer gave us such a high score and thinks that our paper is very strong.

---

### Author Response · Authors · 2024-11-25

Dear AC and Reviewers, thank you for time and effort spent to provide us with many useful suggestions, and the AC for being active during the rebuttal. We are glad that the reviewers see the merits of our work:
* good motivation of our work [VJyB],
* originality and contributions of our DSI framework and metrics [Mwfz] and the technical correctness of them [VJyB]
* interesting question selection, experimental design and results [7tX6, 295U]
* discovering interesting properties of similarity perception development during training and networks’ mistakes [7tX6, 295U]
* acknowledging that using semantic similarity and WordNet (our reference) is interesting and more practical compared to human polls [VJyB]
* clarity [Mwfz, VJyB]

We did our best to include all suggested changes/additional content. We added 6 new appendices (with main paper references) and significantly modified abstract, introduction and related work. We marked all changes in pink in the paper (section title for larger fragm.). Below, we summarize the largest changes:
* we revised the paper's beginning (abstract, introduction, related work) to de-emphasize using human judgments as a reference, instead highlighting semantic similarity's validity, importance, and practical applications in model evaluation and explainable AI (the merit also according to the reviewer [VJyB]).
* we broaden our related work with references provided by the reviewers and us to show that similarity has different notions, why semantic similarity is more adequate than human judgments for our framework (but to highlight that for some limited cases, such analyses are possible), and to better place our work in the related literature.
* as suggested by [7tX6] and [295U], we performed additional experiments with 2 more CNNs to check whether the different shape of the Weights Similarity Index Curves for MobileNetV2 and ResNet18 is because of them being outliers - additional results support our initial hypothesis that this behavior is just characteristic for CNNs.  -> **Appendix G**.
* as suggested by [7tX6], we investigated the decrease in errors only IDM. We measured via cosine similarity how distant are the correctly/incorrectly classified images from the network's category perception (the incorrectly classified samples are much distant, especially for later epochs). We also performed a manual analysis of incorrectly classified samples after epochs 5, 50, 299. The results further support our hypothesis that the mistakes in the later epochs are due to less typical and difficult samples, or even mislabeling issues.  -> **Appendix H**
* as suggested by [7tX6], we analyzed the bumps in SAI(NCSM, SCSM) curve. Additional results (train loss/learning rate and SAI plotted together) show that bumps occur during dynamic learning phases, and after the learning rate changes, causing temporary spikes as the optimizer adjusts. The SAI curve stabilizes as learning slows and rate changes become smoother. Noise from techniques like dropout can also contribute to instabilities.  -> **Appendix I**
* the reviewer [VJyB] pointed out that our methods cannot be applied to networks w/o classifiers (e.g., contrastive loss/text-image). In **Appendix J**, we demonstrate that by generating CSMs using extracted features it is possible (with additional cost, so where possible it is better to use our method). We compared the Templates-based CSMs (train/test split) for an example classifier with each other and with its weights-based CSM, finding that Templates-based CSMs can serve as an approximation. We showed qualitative and quantitative results for CLIP and DINOv2 - despite focusing on representations, their similarity perception is very similar to the one of standard models, further supporting that similarity is natural and vital for learning of traditional models.
* as suggested by [Mwfz], we performed experiments to show that adversarial attacks can extend our framework (to generate confusion-based class similarity matrices). It requires a computational overhead due to perturbation generation, so by default it is not included in our framework (focus on simplicity, efficiency). We also examined whether the semantic similarity alignment can serve as an indicator of adversarial robustness. The results for 1 CNN & 1 ViT suggest that it is possible. -> **Appendix K**
* as suggested by [Mwfz], we measured the computation time of our methods: “The additional complexity of computing our new metrics accounted for approximately **0.29%** of epoch time computation for MobileNetV2” -> **Section A.2 in App. A**
*  reviewers [VJyB] and [295U] suggested that using many metrics and acronyms can be confusing, so we added Appendix L with a dictionary of acronyms and with broader descriptions (according to [7tX6]'s suggestion).

We answer all weaknesses/questions of the reviewers under their reviews. If the changes and responses meet the expectations, we would kindly ask the reviewers to reconsider their scores.

---

> ### Author Response · Authors · 2024-12-02
> **Dear Reviewers**
>
> Dear Reviewers,
>
> We would like to sincerely thank you for engaging in an exceptionally interesting discussion with us (with 38 comments in total, with this becoming the 39th one)! We appreciate all of your comments and support. You helped us significantly improve our paper and inspired us to dig deeper into the analyzed topics and craft additional experiments with some fascinating outcomes! We are greateful for all the interactions. It is last day that you, reviewers, can leave the comments. Therefore, if you have any remaining concerns and suggestions on how to improve our paper, please, do not hesitate to leave a comment for us, and we will do what we can to meet your expectations and standards. Thank you in advance!

---

### Meta-Review · Area_Chair_8Mnd · 2024-12-15

**Metareview:**

The paper proposed Deep Similarity Inspector (DSI) framework. It reveals how deep vision networks like CNNs and ViTs develop and refine similarity perception during training. Results highlight distinct dynamics and the phenomenon of mistake refinement.

The paper is generally well-written, the topic is well-motivated, and the insights on similarity for supervised learning models are interesting. The introduced DSI is novel and useful for explainable AI.

The paper receives mixed scores of 8, 6, 5 and 5. The key issues were identified by two reviewers, which led to the paper rejection. The authors are encouraged to incorporate this feedback into the revised version and resubmit it to the future conferences.

[limited problem setting to study semantic similarity on supervised learning] Both reviewers VJyB and 295U initially expressed concerns about using WordNet or supervised learning methods to study semantic similarity. For example, Review VJyB highlighted that the authors primarily evaluate the task of supervised object recognition and compare the effects of architecture on similarity emergence, while reviewer 295U raised the concern of using WordNet as the criteria to evaluate semantic similarity; i.e. class label not always being a good proxy for semantic similarity and the issues with the WordNet hierarchy.

Later on, during the internal discussions among AC and reviewers, reviewer 295U further raised the concern about the results on unsupervised approach DINOv2 in Appendix J. The insights obtained from the results are limited and known beforehand; i.e. DINOv2 learns good representations that can be used for transfer learning via linear probing.

[writing clarity and reorganization of text are needed] After the rebuttal, the reviewer 295U raised several issues on writing and clarity. The concerns are summarized here: The related work section is poorly structured, missing key references on human similarity judgment and neural network comparison methods, and lacks integration. While new experiments in the Appendix explore broader directions (e.g., detection, self-supervised methods), they need better alignment with the main paper for coherence. The title and introduction suggest a broad exploration of similarity, but the focus is narrowly on semantic similarity, and some analyses (e.g., matrix structure comparison in F1) are unclear or insufficiently explained.

[missing key experiments for important results] After the rebuttal, the reviewer 295U further raised the issue of missing key experiments for the main results. Specifically, key findings, like semantic similarity emergence, lack comprehensive results for datasets like ImageNet and COCO.

**Additional Comments On Reviewer Discussion:**

The paper receives mixed scores of 8, 6, 5 and 5.

After the rebuttal, the following issues were identified. AC initiated the internal discussions with reviewers. One reviewer raised further concerns on writing clarity and missing key experiments.

[limited problem setting to study semantic similarity on supervised learning] Both reviewers VJyB and 295U initially expressed concerns about using WordNet or supervised learning methods to study semantic similarity. For example, Review VJyB highlighted that the authors primarily evaluate the task of supervised object recognition and compare the effects of architecture on similarity emergence, while reviewer 295U raised the concern of using WordNet as the criteria to evaluate semantic similarity; i.e. class label not always being a good proxy for semantic similarity and the issues with the WordNet hierarchy.

Later on, during the internal discussions among AC and reviewers, reviewer 295U further raised the concern about the results on unsupervised approach DINOv2 in Appendix J. The insights obtained from the results are limited and known beforehand; i.e. DINOv2 learns good representations that can be used for transfer learning via linear probing.

[writing clarity and reorganization of text are needed] During the internal discussions among AC and reviewers, the reviewer 295U raised several issues on writing and clarity. The concerns are summarized here: The related work section is poorly structured, missing key references on human similarity judgment and neural network comparison methods, and lacks integration. While new experiments in the Appendix explore broader directions (e.g., detection, self-supervised methods), they need better alignment with the main paper for coherence. The title and introduction suggest a broad exploration of similarity, but the focus is narrowly on semantic similarity, and some analyses (e.g., matrix structure comparison in F1) are unclear or insufficiently explained.

[missing key experiments for important results] During the internal discussions among AC and reviewers, the reviewer 295U further raised the issue of missing key experiments for the main results. Specifically, key findings, like semantic similarity emergence, lack comprehensive results for datasets like ImageNet and COCO.

---

### Decision · Program_Chairs · 2025-01-22

Reject